# ADMM for Structured Fractional Minimization

**Ganzhao Yuan**
Peng Cheng Laboratory, China
`yuangzh@pcl.ac.cn`

## Abstract

This paper considers a class of structured fractional minimization problems. The numerator consists of a differentiable function, a simple nonconvex nonsmooth function, a concave nonsmooth function, and a convex nonsmooth function composed with a linear operator. The denominator is a continuous function that is either weakly convex or has a weakly convex square root. These problems are prevalent in various important applications in machine learning and data science. Existing methods, primarily based on subgradient methods and smoothing proximal gradient methods, often suffer from slow convergence and numerical stability issues. In this paper, we introduce FADMM, the first Alternating Direction Method of Multipliers tailored for this class of problems. FADMM decouples the original problem into linearized proximal subproblems, featuring two variants: one using Dinkelbach's parametric method (FADMM-D) and the other using the quadratic transform method (FADMM-Q). By introducing a novel Lyapunov function, we establish that FADMM converges to $\epsilon$-approximate critical points of the problem within an oracle complexity of $\mathcal{O}(1/\epsilon^3)$. Extensive experiments on synthetic and real-world datasets, including sparse Fisher discriminant analysis, robust Sharpe ratio minimization, and robust sparse recovery, demonstrate the effectiveness of our approach. [1]

## 1 Introduction

This paper focuses on the following class of nonconvex and nonsmooth fractional minimization problem (where '$\triangleq$' denotes definition):

$$\min_{\mathbf{x}} F(\mathbf{x}) \triangleq \frac{u(\mathbf{x})}{d(\mathbf{x})}, \text{ where } u(\mathbf{x}) \triangleq f(\mathbf{x}) + \delta(\mathbf{x}) - g(\mathbf{x}) + h(\mathbf{A}\mathbf{x}). \tag{1}$$

Here, $\mathbf{x} \in \mathbb{R}^n$ and $\mathbf{A} \in \mathbb{R}^{m \times n}$. We impose the following assumptions on Problem (1). (*i*) The function $f(\mathbf{x})$ is differentiable and possibly nonconvex. (*ii*) The function $\delta(\mathbf{x})$ is possibly nonconvex, nonsmooth, and non-Lipschitz. (*iii*) Both functions $g(\mathbf{x})$ and $h(\mathbf{x})$ are convex and possibly nonsmooth. (*iv*) Both functions $\delta(\mathbf{x})$ and $h(\mathbf{y})$ are simple, such that their proximal operators can be computed efficiently and exactly. (*v*) The function $d(\mathbf{x})$ is Lipschitz continuous, and either $d(\mathbf{x})$ itself or its square root, $d(\mathbf{x})^{1/2}$, is weakly convex. (*vi*) To ensure Problem (1) is well-defined, we assume that all functions $g(\mathbf{x})$, $\delta(\mathbf{x})$, $h(\mathbf{y})$, and $d(\mathbf{x})$ are proper and lower semicontinuous, $u(\mathbf{x}) \geq 0$, and $d(\mathbf{x}) \geq 0$ for all $\mathbf{x}$ and $\mathbf{y}$.

Problem (1) serves as a fundamental optimization framework in various machine learning and data science models, such as sparse Fisher discriminant analysis (Bishop & Nasrabadi, 2006), (robust) Sharpe ratio maximization (Chen et al., 2011), robust sparse recovery (Yuan, 2023; Yang & Zhang, 2011), limited-angle CT reconstruction (Wang et al., 2021), AUC maximization (Wang et al., 2022), and signal-to-noise ratio maximization (Shen & Yu, 2018a;b). An alternative formulation of Problem (1) can be obtained by replacing the maximization with the minimization, as explored in the fractional optimization literature (Stancu-Minasian, 2012; Schaible, 1995). Although these two formulations are generally distinct, the corresponding algorithmic developments can readily adapt to either formulation.

---

[1]Future versions of this paper can be found at `https://arxiv.org/abs/2411.07496`.

## 1.1 MOTIVATING APPLICATIONS

Many models in machine learning and data science can be formulated as Problem (1). We present the sparse Fisher discriminant analysis application below, with additional applications, including robust Sharpe ratio maximization and robust sparse recovery, detailed in Appendix B.

• **Sparse Fisher Discriminant Analysis (Sparse FDA)**. Given observations from two distinct classes, let $\boldsymbol{\mu}_{(i)} \in \mathbb{R}^n$ and $\boldsymbol{\Sigma}_{(i)} \in \mathbb{R}^{n \times n}$ represent the mean vector and covariance matrix of class $i$ ($i = 1 \text{ or } 2$), respectively. Classical FDA (Bishop & Nasrabadi, 2006; Xu & Li, 2020) aims to find an orthogonal subspace $\mathbf{X} \in \Omega$ with $\Omega \triangleq \{\mathbf{X} \mid \mathbf{X}^\mathsf{T}\mathbf{X} = \mathbf{I}_r\}$, that maximizes the between-class variance relative to the within-class variance. This leads to the following optimization problem: $\min_{\mathbf{X} \in \Omega} \operatorname{tr}(\mathbf{X}^\mathsf{T}(\boldsymbol{\Sigma}_{(1)} + \boldsymbol{\Sigma}_{(2)})\mathbf{X})/\operatorname{tr}(\mathbf{X}^\mathsf{T}((\boldsymbol{\mu}_{(1)} - \boldsymbol{\mu}_{(2)})(\boldsymbol{\mu}_{(1)} - \boldsymbol{\mu}_{(2)})^\mathsf{T})\mathbf{X})$. Inducing sparsity in the solution helps mitigate overfitting and enhances the interpretability of the model in high-dimensional data analysis (Journée et al., 2010). We consider the following Difference-of-Convex (DC) model (Bi et al., 2014; Bi & Pan, 2016; Gotoh et al., 2018) for learning sparse orthogonal loadings for FDA:

$$\min_{\mathbf{X} \in \mathbb{R}^{n \times r}} \frac{\operatorname{tr}(\mathbf{X}^\mathsf{T}\mathbf{C}\mathbf{X}) + \rho(\|\mathbf{X}\|_1 - \|\mathbf{X}\|_{[k]})}{\operatorname{tr}(\mathbf{X}^\mathsf{T}\mathbf{D}\mathbf{X})}, \text{ s. t. } \mathbf{X} \in \Omega, \tag{2}$$

where $\mathbf{D} \triangleq (\boldsymbol{\mu}_{(1)} - \boldsymbol{\mu}_{(2)})(\boldsymbol{\mu}_{(1)} - \boldsymbol{\mu}_{(2)})^\mathsf{T}$, $\mathbf{C} = \boldsymbol{\Sigma}_{(1)} + \boldsymbol{\Sigma}_{(2)}$, and $\|\mathbf{X}\|_{[k]}$ is the $\ell_1$ norm of the $k$ largest (in magnitude) elements of the matrix $\mathbf{X}$. Problem (2) exhibits a beneficial exact penalty property (Bi et al., 2014; Bi & Pan, 2016), such that $\|\mathbf{X}\|_{[k]}$ closely approximates $\|\mathbf{X}\|_1$ when $\rho$ exceeds a certain threshold, thereby leading to a solution $\mathbf{X}$ with $k$-sparsity. We define $\iota_\Omega(\mathbf{X})$ as the indicator function of the set $\Omega$. Problem (2) coincides with Problem (1) with $\mathbf{x} = \operatorname{vec}(\mathbf{X})$, $f(\mathbf{x}) = \operatorname{tr}(\mathbf{X}^\mathsf{T}\mathbf{C}\mathbf{X})$, $\delta(\mathbf{x}) = \iota_\Omega(\mathbf{X})$, $g(\mathbf{x}) = \rho\|\mathbf{X}\|_{[k]}$, $\mathbf{A} = \mathbf{I}$, $h(\mathbf{A}\mathbf{x}) = \rho\|\mathbf{X}\|_1$, and $d(\mathbf{x}) = \operatorname{tr}(\mathbf{X}^\mathsf{T}\mathbf{D}\mathbf{X})$. Importantly, both $d(\mathbf{x})$ and $d(\mathbf{x})^{1/2}$ are $W_d$-weakly convex with $W_d = 0$.

## 1.2 CONTRIBUTIONS AND ORGANIZATIONS

The contributions of this paper are threefold. (***i***) We propose FADMM, a new ADMM tailored for nonsmooth composite fractional minimization problems. This method includes two specialized variants: FADMM based on Dinkelbach's parametric method (FADMM-D) and FADMM based on the quadratic transform method (FADMM-Q). (***ii***) We establish that both FADMM-D and FADMM-Q algorithms converge to an $\epsilon$-critical point with a computational complexity of $\mathcal{O}(1/\epsilon^3)$. This is the first report of iteration complexity results for estimating approximate stationary points for this class of fractional programs. (***iii***) We conducted experiments on sparse FDA, robust Sharpe ratio maximization, and robust sparse recovery to demonstrate the effectiveness of our approach.

The rest of the paper is organized as follows: Section 2 reviews related work. Section 3 presents technical preliminaries. Section 4 details the proposed algorithm. Section 5 discusses global convergence. Section 6 addresses iteration complexity. Section 7 provides some experiment results, and Section 8 concludes the paper.

## 2 RELATED WORK

We review some nonconvex optimization algorithms that are related to solve the fractional program in Problem (1).

• **Algorithms in Limited Scenarios**. Existing fractional minimization algorithms primarily address a special instance of Problem (1) that $\min_{\mathbf{x}} F(\mathbf{x}) \triangleq u(\mathbf{x})/d(\mathbf{x})$, where $u(\mathbf{x}) \triangleq f(\mathbf{x}) + \delta(\mathbf{x})$. (***i***) Dinkelbach's Parametric Algorithm (DPA) (Dinkelbach, 1967) is a classical approach. The fractional program has an optimal solution $\bar{\mathbf{x}}$ if and only if $\bar{\mathbf{x}}$ is an optimal solution to the problem $\min_{\mathbf{x}} u(\mathbf{x}) - \bar{\lambda}d(\mathbf{x})$, where $\bar{\lambda} = F(\bar{\mathbf{x}})$. Since the optimal value $\bar{\lambda}$ is generally unknown, iterative methods are used. DPA generates a sequence $\{\mathbf{x}^t\}$ as: $\mathbf{x}^{t+1} = \arg\min_{\mathbf{x}} u(\mathbf{x}) - \lambda^t d(\mathbf{x})$, with $\lambda^t$ updated as $\lambda^t = F(\mathbf{x}^t)$. (***ii***) The Quadratic Transform Algorithm (QTA) (Zhou & Yang, 2014; Shen & Yu, 2018a;b) reformulates the problem as: $\min_{\mathbf{x}} -d(\mathbf{x})/u(\mathbf{x}) \Leftrightarrow \min_{\mathbf{x}, \alpha} \alpha^2 u(\mathbf{x}) - 2\alpha d(\mathbf{x})^{1/2}$. QTA generates a sequence $\{\mathbf{x}^t\}$ as: $\mathbf{x}^{t+1} = \arg\min_{\mathbf{x}}(\alpha^t)^2 u(\mathbf{x}) - 2\alpha^t d(\mathbf{x})^{1/2}$, with $\alpha^t$ updated as $\alpha^t = d(\mathbf{x}^t)^{1/2} \cdot u(\mathbf{x}^t)^{-1}$. This method is particularly suited for solving multiple-ratio fractional

programs. (***iii***) Linearized variants of DPA and QTA (Li & Zhang, 2022; Boţ et al., 2023a) address the high computational cost of solving nonconvex subproblems in DPA and QTA. They employ full splitting paradigms and achieve fast convergence by performing a single iteration of splitting algorithms at each step, efficiently avoiding inner loops to solve complex nonconvex problems. The proposed ADMM algorithm is built on linearized DPA and QTA to solve the subproblems.

● **General Algorithms for Solving Problem (1)**. (***i***) Subgradient Projection Methods (SPM) (Li et al., 2021) provide a simple and intuitive approach to solving Problem (1). SPM iteratively updates the solution by moving along the negative subgradient direction and projecting onto the feasible set: $\mathbf{x}^{t+1} = \mathcal{P}_\Omega(\mathbf{x}^t - \eta^t \mathbf{g}^t)$, where $\mathbf{g}^t \in \partial F(\mathbf{x}^t)$, $\Omega$ is the constraint set, and $\eta^t$ is a diminishing step size. However, due to the non-uniqueness of $\mathbf{g}^t$, these methods often exhibit slower convergence and numerical instability. (***ii***) Smoothing Proximal Gradient Methods (SPGM) (Beck & Rosset, 2023; Yuan, 2024a; Bian & Chen, 2020; Böhm & Wright, 2021) combine gradient-based optimization with smoothing techniques to handle nonsmooth terms in the objective function. By approximating nonsmooth components with smooth surrogates, SPGM enables more efficient updates via the proximal gradient method, achieving convergence for complex nonsmooth problems. Notably, the proposed FADMM algorithm reduces to SPGM when the multiplier is set to zero in all iterations. (***iii***) Full Splitting Algorithm (FSA) (Boţ et al., 2023b) applies a smoothing technique by introducing a strongly concave term into the dual maximization problem, effectively framing the method as a primal-dual approach. However, the convergence analysis of FSA relies on the Kurdyka-Lojasiewicz (KL) inequality of the problem, and no iteration complexity results are provided. Overall, our proposed FADMM algorithm demonstrates faster convergence and superior numerical stability compared to SPM, SPGM, and FSA.

● **Other Fractional Minimization Algorithms**. (***i***) Charnes-Cooper transform algorithm converts an original linear-fractional programming problem to a standard linear programming problem (Charnes & Cooper, 1962). Using the transformation $\mathbf{y} = \frac{\mathbf{x}}{d(\mathbf{x})}$, $t = \frac{1}{d(\mathbf{x})}$, Problem (1) can be reformulated as: $\min_{t,\mathbf{y}} tu(\mathbf{y}/t)$, $s.t. \ td(\mathbf{y}/t) = 1$. (***ii***) Coordinate descent algorithms (Yuan, 2023) iteratively solve one-dimensional subproblems globally and are guaranteed to converge to stronger coordinate-wise stationary points for a specific class of fractional programs. (***iii***) Inertial proximal block coordinate methods (Boţ et al., 2023a), based on the quadratic transform, have been proposed to address a class of nonsmooth sum-of-ratios minimization problems.

● **ADMM for Nonconvex Optimization**. The Alternating Direction Method of Multipliers (ADMM) is a powerful optimization technique that addresses complex problems by breaking them down into simpler, more manageable subproblems, which are then solved iteratively to achieve convergence. The standard ADMM was first introduced in (Gabay & Mercier, 1976), with complexity analysis for convex settings conducted in (He & Yuan, 2012; Monteiro & Svaiter, 2013). Motivated by research on the convergence analysis of nonconvex ADMM (Li & Pong, 2015; Hong et al., 2016; Boţ et al., 2019; Boţ & Nguyen, 2020; Yuan, 2024b), we propose applying ADMM to solve structured fractional minimization problems. To the best of our knowledge, this is the first instance of ADMM being applied to fractional programs. Our goal is to investigate both the theoretical iteration complexity required to reach an approximate stationary point and the empirical performance of the proposed method.

## 3 TECHNICAL PRELIMINARIES

This section presents technical preliminaries on basic assumptions, stationary points, and Nesterov's smoothing techniques. Notations, additional technical preliminaries, and relevant lemmas are provided in Appendix Section A.

### 3.1 BASIC ASSUMPTIONS AND STATIONARY POINTS

We impose the following assumptions on Problem (1) throughout this paper.

**Assumption 3.1.** *There exists a universal positive constant $\bar{\mathbf{x}}$ such that $\|\mathbf{x}\| \leq \bar{\mathbf{x}}$ for all $\mathbf{x} \in \text{dom}(F)$.*

**Assumption 3.2.** *The function $f(\cdot)$ is $L_f$-smooth such that $\|\nabla f(\mathbf{x}) - \nabla f(\mathbf{x}')\| \leq L_f \|\mathbf{x} - \mathbf{x}'\|$ holds for all $\mathbf{x}, \mathbf{x}' \in \mathbb{R}^n$. This implies that: $|f(\mathbf{x}) - f(\mathbf{x}') - \langle \nabla f(\mathbf{x}'), \mathbf{x} - \mathbf{x}' \rangle| \leq \frac{L_f}{2} \|\mathbf{x} - \mathbf{x}'\|_2^2$ (cf. Lemma 1.2.3 in (Nesterov, 2003)).*

**Assumption 3.3.** *Let $\mu > 0$, $\mathbf{x}' \in \mathbb{R}^n$, and $\mathbf{y}' \in \mathbb{R}^m$. Both proximal operators, $\mathrm{Prox}(\mathbf{y}'; h, \mu) \triangleq \arg\min_{\mathbf{y}} \frac{1}{2\mu} \|\mathbf{y} - \mathbf{y}'\|_2^2 + h(\mathbf{y})$ and $\mathrm{Prox}(\mathbf{x}'; \delta, \mu) \triangleq \arg\min_{\mathbf{x}} \frac{1}{2\mu} \|\mathbf{x} - \mathbf{x}'\|_2^2 + \delta(\mathbf{x})$, can be computed efficiently and exactly.*

**Assumption 3.4.** *The function $d(\mathbf{x})$ is $C_d$-Lipschitz continuous with $C_d \geq 0$, and meets one of the following conditions for some $W_d \geq 0$: (a) $d(\mathbf{x})$ is $W_d$-weakly convex. (b) $\sqrt{d(\mathbf{x})}$ is $W_d$-weakly convex.*

**Remark 3.5.** *(i) Assumption 3.1 holds by setting $\delta(\mathbf{x}) = \iota_\Omega(\mathbf{x})$, where $\Omega$ is a compact set. This follows from the fact that if $\mathbf{x} \in \mathrm{dom}(F) \triangleq \{\mathbf{x} : F(\mathbf{x}) < +\infty\}$, then $\mathbf{x}$ is feasible, ensuring that $\|\mathbf{x}^t\| \leq \bar{\mathbf{x}}$ for some $\bar{\mathbf{x}} > 0$. (ii) Assumption 3.2 is commonly used in the convergence analysis of nonconvex algorithms. (iii) Assumption 3.3 is mild and is satisfied by our applications. Appendix G details the computation of proximal operators.*

We now introduce the definition of stationary points for Problem (1). A straightforward option is the *Fréchet stationary point* (Rockafellar & Wets., 2009; Mordukhovich, 2006). Recall that a solution $\ddot{\mathbf{x}}$ is a Fréchet stationary point of Problem (1) if: $\mathbf{0} \in \widehat{\partial} F(\ddot{\mathbf{x}}) = \widehat{\partial}((f + \delta - g + h \circ A)/d)(\ddot{\mathbf{x}})$. However, computing a Fréchet stationary point is challenging for general nonconvex nonsmooth programs. Following the work of (Li et al., 2022b; Li & Zhang, 2022; Boţ et al., 2023a;b; Yuan, 2023), we adopt a weaker notion of optimality, namely critical points (or limiting lifted stationary points), defined as follows:

**Definition 3.6.** (Critical Point) A solution $\dot{\mathbf{x}} \in \mathrm{dom}(F)$ is a critical point of Problem (1) if: $\mathbf{0} \in \partial\delta(\dot{\mathbf{x}}) + \nabla f(\dot{\mathbf{x}}) - \partial g(\dot{\mathbf{x}}) + \mathbf{A}^\mathsf{T} \partial h(\mathbf{A}\dot{\mathbf{x}}) - F(\dot{\mathbf{x}})\partial d(\dot{\mathbf{x}})$.

**Remark 3.7.** *Using Lemma A.1 in Appendix A.2, we obtain that $\widehat{\partial} F(\mathbf{x}) \in g(\mathbf{x})^{-2}\{\partial\delta(\mathbf{x}) + \nabla f(\mathbf{x}) - \partial g(\mathbf{x}) + \mathbf{A}^\mathsf{T}\partial h(\mathbf{A}\mathbf{x}) - F(\mathbf{x})\partial d(\mathbf{x})\}$ for any $\mathbf{x}$. According to Definition 3.6, $\mathbf{0} \in \widehat{\partial} F(\ddot{\mathbf{x}})$ implies that $\ddot{\mathbf{x}}$ is a critical point of Problem 1, while the converse is generally not true. However, under certain mild conditions discussed in (Boţ et al., 2023b;a), Definition 3.6 aligns with the standard Fréchet stationary point that $\mathbf{0} \in \widehat{\partial} F(\dot{\mathbf{x}})$.*

## 3.2 Nesterov's Smoothing Technique

The nonsmooth nature of the function $h(\mathbf{y})$ presents challenges for the algorithm design and theoretical analysis. To address this, we approximate $h(\mathbf{y})$ with a smooth function $h_\mu(\mathbf{y})$ using Nesterov's smoothing technique (Nesterov, 2003; 2013; Devolder et al., 2012), which relies on the conjugate function of $h(\mathbf{y})$. We introduce the following useful definition in this context.

**Definition 3.8.** *For a proper, convex, and lower semicontinuous function $h(\mathbf{y}) : \mathbb{R}^m \mapsto \mathbb{R}$, the Nesterov's smoothing function for $h(\mathbf{y})$ with a parameter $\mu \in (0, \infty)$ is defined as: $h_\mu(\mathbf{y}) = \max_{\mathbf{v}} \langle \mathbf{y}, \mathbf{v} \rangle - h^*(\mathbf{v}) - \frac{\mu}{2}\|\mathbf{v}\|_2^2$.*

We outline some key properties of Nesterov's smoothing function.

**Lemma 3.9.** *(Proof in Appendix C.1) Assume that $h(\mathbf{y})$ is $C_h$-Lipschitz continuous. We let $\mu > 0$, and $0 < \mu_2 \leq \mu_1$. We have the following results:*

(a) *The function $h_\mu(\mathbf{y})$ is $(1/\mu)$-smooth and convex, with its gradient given by $\nabla h_\mu(\mathbf{y}) = \arg\max_{\mathbf{v}} \left\{ \langle \mathbf{y}, \mathbf{v} \rangle - h^*(\mathbf{v}) - \frac{\mu}{2}\|\mathbf{v}\|_2^2 \right\}$, it holds that $\nabla h_\mu(\mathbf{y}) = \frac{1}{\mu}(\mathbf{y} - \mathrm{Prox}(\mathbf{y}; h, \mu))$.*
(b) $0 < h(\mathbf{y}) - h_\mu(\mathbf{y}) \leq \frac{1}{2}C_h^2\mu$.
(c) *The function $h_\mu(\mathbf{y})$ is $C_h$-Lipschitz continuous.*
(d) $h_\mu(\mathbf{y}) = \frac{1}{2\mu}\|\mathbf{y} - \dot{\mathbf{y}}\|_2^2 + h(\dot{\mathbf{y}})$, *where $\dot{\mathbf{y}} = \mathrm{Prox}(\mathbf{y}; h, \mu)$.*
(e) $0 \leq h_{\mu_2}(\mathbf{y}) - h_{\mu_1}(\mathbf{y}) \leq \frac{1}{2}C_h^2(\mu_1 - \mu_2)$.
(f) $\|\nabla h_{\mu_2}(\mathbf{y}) - \nabla h_{\mu_1}(\mathbf{y})\| \leq (\frac{\mu_1}{\mu_2} - 1)C_h$.

**Lemma 3.10.** *(Proof in Section C.2) Assume that $h(\mathbf{y})$ is $C_h$-Lipschitz continuous. Consider $\bar{\mathbf{y}} = \arg\min_{\mathbf{y}} h_\mu(\mathbf{y}) + \frac{1}{2}\beta\|\mathbf{y} - \mathbf{b}\|_2^2 \triangleq \mathrm{Prox}(\mathbf{b}; h_\mu, 1/\beta)$, where $\mathbf{b} \in \mathbb{R}^m$, and $\beta, \mu > 0$. We have:*

(a) $\bar{\mathbf{y}} = \frac{\check{\mathbf{y}} + \beta\mu\mathbf{b}}{1 + \beta\mu}$, *where $\check{\mathbf{y}} \triangleq \mathrm{Prox}(\mathbf{b}; h, \mu + 1/\beta)$.*
(b) $\beta(\mathbf{b} - \bar{\mathbf{y}}) \in \partial h(\check{\mathbf{y}})$.
(c) $\|\bar{\mathbf{y}} - \check{\mathbf{y}}\| \leq \mu C_h$.

**Remark 3.11.** *(i) Lemmas 3.9 and 3.10 can be derived using standard convex analysis and play an essential role in the analysis of the proposed* FADMM *algorithm. (ii) Nesterov's smoothing function (Nesterov, 2003) is essentially equivalent to the Moreau envelope smoothing function (Böhm & Wright, 2021; Beck, 2017), as demonstrated in Lemma 3.9(d). (iii) Lemma 3.10 demonstrates how to compute the proximal operator* $\mathrm{Prox}(\mathbf{b}; h_\mu, 1/\beta)$ *using* $\mathrm{Prox}(\mathbf{b}; h, \mu + 1/\beta)$, *where* $\beta > 0$ *and* $\mathbf{b} \in \mathbb{R}^m$ *are given parameters.*

## 4 THE PROPOSED FADMM ALGORITHM

This section presents the proposed FADMM Algorithm for solving Problem (1), featuring two variants: one based on Dinkelbach's parametric method (FADMM-D) (Dinkelbach, 1967) and the other on the quadratic transform method (FADMM-Q) (Zhou & Yang, 2014; Shen & Yu, 2018a). Notably, FADMM-D and FADMM-Q target different problem structures: FADMM-D is designed for Assumption 3.4(a), while FADMM-Q is suited for Assumption 3.4(b), with potential extensions to multi-ratio fractional programs (Boţ et al., 2023a).

We first introduce a new variable $\mathbf{y} \in \mathbb{R}^m$ and reformulate Problem (1) as: $\min_{\mathbf{x}, \mathbf{y}}\{f(\mathbf{x}) + \delta(\mathbf{x}) - g(\mathbf{x}) + h_\mu(\mathbf{y})\}/d(\mathbf{x})$, $\mathbf{A}\mathbf{x} = \mathbf{y}$, where $h_\mu(\mathbf{y})$ is the Nesterov's smoothing function of $h(\mathbf{y})$, with $\mu \to 0$ as the smoothing parameter. Notably, similar smoothing techniques have been used in the design of augmented Lagrangian methods (Zeng et al., 2022), ADMM (Li et al., 2022a; Yuan, 2025; 2024b), and minimax optimization (Zhang et al., 2020). From this problem, we define two functions, referred to as modified augmented Lagrangian functions, as follows:

$$\mathcal{L}(\mathbf{x}, \mathbf{y}; \mathbf{z}; \beta, \mu) \triangleq \tfrac{\mathcal{U}(\mathbf{x}, \mathbf{y}; \mathbf{z}; \beta, \mu)}{d(\mathbf{x})}, \tag{3}$$

$$\mathcal{K}(\alpha, \mathbf{x}, \mathbf{y}; \mathbf{z}; \beta, \mu) \triangleq -2\alpha\sqrt{d(\mathbf{x})} + \alpha^2 \mathcal{U}(\mathbf{x}, \mathbf{y}; \mathbf{z}; \beta, \mu), \tag{4}$$

where

$$\mathcal{U}(\mathbf{x}, \mathbf{y}; \mathbf{z}; \beta, \mu) \triangleq \underbrace{f(\mathbf{x}) + \langle \mathbf{A}\mathbf{x} - \mathbf{y}, \mathbf{z}\rangle + \tfrac{\beta}{2}\|\mathbf{A}\mathbf{x} - \mathbf{y}\|_2^2}_{\triangleq\, \mathcal{S}(\mathbf{x}, \mathbf{y}; \mathbf{z}; \beta)} + \delta(\mathbf{x}) - g(\mathbf{x}) + h_\mu(\mathbf{y}). \tag{5}$$

Here, $\beta$ is the penalty parameter and $\mathbf{z}$ is the dual variable for the linear constraint. In brief, FADMM updates the primal variables sequentially, keeping the others fixed, and updates the dual variables via gradient ascent on the dual problem. It iteratively generates a sequence $\{\mathbf{x}^t, \mathbf{y}^t, \mathbf{z}^t, \lambda^t, \beta^t, \mu^t\}_{t=0}^\infty$ or $\{\alpha^t, \mathbf{x}^t, \mathbf{y}^t, \mathbf{z}^t, \beta^t, \mu^t\}_{t=0}^\infty$, where $\beta^t = \beta^0(1 + \xi t^p)$, $\mu^t = \tfrac{\chi}{\beta^t}$, and $\{\beta^0, \xi, p, \chi\}$ are fixed constants.

**Majorization Minimization (MM).** MM is an effective optimization strategy to minimize complex functions and is widely used to develop practical optimization algorithms (Mairal, 2013; Razaviyayn et al., 2013). This technique iteratively constructs a majorization function that upper-bounds the objective, enabling efficient optimization and gradual reduction of the objective function. We define $s(\mathbf{x}) \triangleq \mathcal{S}(\mathbf{x}, \mathbf{y}^t, \mathbf{z}^t; \beta^t)$, where $t$ is known from context. Given that $s(\mathbf{x})$ is $(L_f + \beta^t\|\mathbf{A}\|_2^2)$-smooth, $g(\mathbf{x})$ is convex, and both $d(\mathbf{x})$ or $\sqrt{d(\mathbf{x})}$ are $W_d$-weakly convex, we construct the majorization functions for the four functions as follows.

**(a)** $s(\mathbf{x}) \leq \mathcal{U}^t(\mathbf{x}; \mathbf{x}^t) \triangleq s(\mathbf{x}^t) + \langle \mathbf{x} - \mathbf{x}^t, \nabla s(\mathbf{x}^t)\rangle + \tfrac{1}{2}(L_f + \beta^t\|\mathbf{A}\|_2^2)\|\mathbf{x} - \mathbf{x}^t\|_2^2$.

**(b)** $-g(\mathbf{x}) \leq \mathcal{R}(\mathbf{x}; \mathbf{x}^t) \triangleq -g(\mathbf{x}^t) - \langle \mathbf{x} - \mathbf{x}^t, \boldsymbol{\xi}\rangle, \forall \boldsymbol{\xi} \in \partial g(\mathbf{x}^t)$.

**(c)** $-d(\mathbf{x}) \leq \dot{\mathcal{V}}(\mathbf{x}; \mathbf{x}^t) \triangleq -d(\mathbf{x}^t) - \langle \mathbf{x} - \mathbf{x}^t, \boldsymbol{\xi}\rangle + \tfrac{W_d}{2}\|\mathbf{x} - \mathbf{x}^t\|_2^2, \forall \boldsymbol{\xi} \in \partial d(\mathbf{x}^t)$.

**(d)** $-\sqrt{d(\mathbf{x})} \leq \ddot{\mathcal{V}}(\mathbf{x}; \mathbf{x}^t) \triangleq -\sqrt{d(\mathbf{x}^t)} - \langle \mathbf{x} - \mathbf{x}^t, \boldsymbol{\xi}\rangle + \tfrac{W_d}{2}\|\mathbf{x} - \mathbf{x}^t\|_2^2, \forall \boldsymbol{\xi} \in \partial\sqrt{d(\mathbf{x}^t)}$.

• FADMM-D **Algorithm**. Based on Equation (3), FADMM-D alternately updates the primal variables $\{\mathbf{x}, \mathbf{y}\}$ and the dual variable $\mathbf{z}$. (*i*) To update the variable $\mathbf{x}$, we approximately solve Dinkelbach's parametric subproblem as follows: $\mathbf{x}^{t+1} \approx \arg\min_{\mathbf{x}} \dot{\mathcal{W}}^t(\mathbf{x}) \triangleq s(\mathbf{x}) + \delta(\mathbf{x}) - g(\mathbf{x}) - \lambda^t d(\mathbf{x})$, where $\lambda^t = \mathcal{L}(\mathbf{x}^t, \mathbf{y}^t; \mathbf{z}^t; \beta^t, \mu^t)$. However, this problem is challenging in general. We apply MM methods and consider the following problem:

$$\min_{\mathbf{x}} \dot{\mathcal{M}}^t(\mathbf{x}; \mathbf{x}^t, \lambda^t) \triangleq \delta(\mathbf{x}) + \mathcal{U}^t(\mathbf{x}; \mathbf{x}^t) + \mathcal{R}(\mathbf{x}; \mathbf{x}^t) + \lambda^t \dot{\mathcal{V}}(\mathbf{x}; \mathbf{x}^t) + \tfrac{\theta-1}{2}\ell(\beta^t)\|\mathbf{x} - \mathbf{x}^t\|_2^2, \quad (6)$$

where $\ell(\beta^t) \triangleq L_f + \beta^t\|\mathbf{A}\|_2^2 + \lambda^t W_d$, and $\theta > 1$. One can verify that $\dot{\mathcal{M}}^t(\mathbf{x}; \mathbf{x}^t, \lambda^t)$ is a majorization function, satisfying $\dot{\mathcal{W}}^t(\mathbf{x}) \leq \dot{\mathcal{M}}^t(\mathbf{x}; \mathbf{x}^t, \lambda^t)$ and $\dot{\mathcal{W}}^t(\mathbf{x}^t) = \dot{\mathcal{M}}^t(\mathbf{x}^t; \mathbf{x}^t, \lambda^t)$ for all $\mathbf{x}$ and

---

**Algorithm 1: FADMM: The Proposed ADMM using Dinkelbach's Parametric Method or the Quadratic Transform Method for Solving Problem (1).**

---

**(S0)** Initialize $\{\mathbf{x}^0, \mathbf{y}^0, \mathbf{z}^0\}$.
**(S1)** Choose $\xi \in (0, \infty)$, $\theta \in (1, \infty)$, $p \in (0, 1)$, and $\chi \in (2\sqrt{1+\xi}, \infty)$.
**(S2)** Choose $\beta^0$ large enough such that $\beta^0 > \underline{\mathrm{v}}/(\underline{\mathrm{Fd}})$, satisfying Assumption 5.7.
**for** $t$ from 0 to $T$ **do**

  **(S3)** $\beta^t = \beta^0(1 + \xi t^p)$, $\mu^t = \chi/\beta^t$.
  **(S4)** Solve the $\mathbf{x}$-subproblem using FADMM-D or FADMM-Q:
  **if** FADMM-D **then**
   | Set $\lambda^t = \mathcal{U}(\mathbf{x}^t, \mathbf{y}^t; \mathbf{z}^t; \beta^t, \mu^t)/d(\mathbf{x}^t)$, and $\mathbf{x}^{t+1} \in \arg\min_\mathbf{x} \dot{\mathcal{M}}^t(\mathbf{x}; \mathbf{x}^t, \lambda^t)$.
  **end**
  **if** FADMM-Q **then**
   | Set $\alpha^{t+1} = \sqrt{d(\mathbf{x}^t)}/\mathcal{U}(\mathbf{x}^t, \mathbf{y}^t; \mathbf{z}^t; \beta^t, \mu^t)$, and $\mathbf{x}^{t+1} \in \arg\min_\mathbf{x} \ddot{\mathcal{M}}^t(\mathbf{x}; \mathbf{x}^t, \alpha^{t+1})$.
  **end**
  **(S5)** $\mathbf{y}^{t+1} = \arg\min_\mathbf{y} h_{\mu^t}(\mathbf{y}) + \frac{\beta^t}{2}\|\mathbf{y} - \mathbf{b}^t\|_2^2$, where $\mathbf{b}^t \triangleq \mathbf{y}^t - \nabla_\mathbf{y}\mathcal{S}(\mathbf{x}^{t+1}, \mathbf{y}^t; \mathbf{z}^t; \beta^t)/\beta^t$.
   It can be solved as $\mathbf{y}^{t+1} = \frac{\check{\mathbf{y}}^{t+1} + \beta^t\mu^t\mathbf{b}^t}{1+\beta^t\mu^t}$, where $\check{\mathbf{y}}^{t+1} \triangleq \mathrm{Prox}(\mathbf{b}^t; h, \mu^t + 1/\beta^t)$.
  **(S6)** $\mathbf{z}^{t+1} = \mathbf{z}^t + \beta^t(\mathbf{A}\mathbf{x}^{t+1} - \mathbf{y}^{t+1})$.
**end**

---

$\mathbf{x}^t$. Problem (6) reduces to the computation of a proximal operator for the function $\delta(\mathbf{x})$, yielding $\mathbf{x}^{t+1} \in \arg\min_\mathbf{x} \dot{\mathcal{M}}^t(\mathbf{x}; \mathbf{x}^t, \lambda^t) = \mathrm{Prox}(\mathbf{x}'; \delta, \theta\ell(\beta^t))$, where $\mathbf{x}' = \mathbf{x}^t - \mathbf{g}/(\theta\ell(\beta^t))$, and $\mathbf{g} \in \nabla s(\mathbf{x}^t) - \partial g(\mathbf{x}^t) - \lambda^t\partial d(\mathbf{x}^t)$. (**ii**) When minimizing the modified augmented Lagrangian function in Equation (3) over $\mathbf{y}$, the problem reduces to solving: $\mathbf{y}^{t+1} \in \arg\min_\mathbf{y} h_{\mu^t}(\mathbf{y}) + \frac{1}{2}\beta^t\|\mathbf{y} - \mathbf{b}^t\|_2^2$, where $\mathbf{b}^t \triangleq \mathbf{y}^t - \nabla_\mathbf{y}\mathcal{S}(\mathbf{x}^{t+1}, \mathbf{y}^t; \mathbf{z}^t; \beta^t)/\beta^t$. (**iii**) We adjust the dual variable $\mathbf{z}$ using the standard gradient ascent update rule in ADMM.

• **FADMM-Q Algorithm**. Based on Equation (4), FADMM-Q alternates between updating the primal variables $\{\alpha, \mathbf{x}, \mathbf{y}\}$ and the dual variable $\mathbf{z}$. (**i**) To update the variable $\alpha$, we set the gradient of $\mathcal{K}(\alpha, \mathbf{x}, \mathbf{y}; \mathbf{z}; \beta)$ w.r.t. $\alpha$ to zero, resulting in the update rule: $\alpha^{t+1} = \sqrt{d(\mathbf{x}^t)}/\mathcal{U}(\mathbf{x}^t, \mathbf{y}^t; \mathbf{z}^t; \beta^t, \mu^t)$. (**ii**) To update the variable $\mathbf{x}$, we approximately solve the following problem: $\mathbf{x}^{t+1} \approx \arg\min_\mathbf{x} \ddot{\mathcal{W}}^t(\mathbf{x}) \triangleq s(\mathbf{x}) + \delta(\mathbf{x}) - g(\mathbf{x}) - \frac{2}{\alpha^{t+1}}\sqrt{d(\mathbf{x})}$. To tackle this challenging problem, we employ MM methods and formulate the following problem:

$$\min_\mathbf{x} \ddot{\mathcal{M}}^t(\mathbf{x}; \mathbf{x}^t, \alpha^{t+1}) \triangleq \delta(\mathbf{x}) + \mathcal{U}^t(\mathbf{x}; \mathbf{x}^t) + \mathcal{R}(\mathbf{x}; \mathbf{x}^t) + \frac{2}{\alpha^{t+1}}\ddot{\mathcal{V}}(\mathbf{x}; \mathbf{x}^t) + \frac{\theta-1}{2}\ell(\beta^t)\|\mathbf{x} - \mathbf{x}^t\|_2^2, \quad (7)$$

where $\ell(\beta^t) \triangleq L_f + \beta^t\|\mathbf{A}\|_2^2 + \frac{2}{\alpha^{t+1}}W_d$, and $\theta > 1$. One can show that $\ddot{\mathcal{W}}^t(\mathbf{x}) \leq \ddot{\mathcal{M}}^t(\mathbf{x}; \mathbf{x}^t, \alpha^{t+1})$ and $\ddot{\mathcal{W}}^t(\mathbf{x}^t) \leq \ddot{\mathcal{M}}^t(\mathbf{x}^t; \mathbf{x}^t, \alpha^{t+1})$ for all $\mathbf{x}$ and $\mathbf{x}^t$. Problem (7) can be efficiently and effectively solved, as it reduces to the computation of a proximal operator for the function $\delta(\mathbf{x})$, yielding $\mathbf{x}^{t+1} \in \arg\min_\mathbf{x} \ddot{\mathcal{M}}^t(\mathbf{x}; \mathbf{x}^t, \alpha^{t+1}) = \mathrm{Prox}(\mathbf{x}'; \delta, \theta\ell(\beta^t))$, where $\mathbf{x}' = \mathbf{x}^t - \mathbf{g}/(\theta\ell(\beta^t))$ and $\mathbf{g} \in \nabla s(\mathbf{x}^t) - \partial g(\mathbf{x}^t) - \frac{2}{\alpha^{t+1}}\partial\sqrt{d(\mathbf{x}^t)}$. (**iii**) We use the same strategy as in FADMM-D to update the primal variable $\mathbf{y}$ and the dual variable $\mathbf{z}$.

We summarize FADMM-D and FADMM-Q in Algorithm 1, and provide the following remarks.

**Remark 4.1.** (**i**) *The $\mathbf{y}$-subproblem in Step (S5) of Algorithm 1 can be solved by invoking Lemma 3.10. (**ii**) The introduction of the strongly convex term $\frac{\theta-1}{2}\ell(\beta^t)\|\mathbf{x} - \mathbf{x}^t\|_2^2$ with $\theta > 1$ as in Problems (6) and (7) is crucial to our analysis. (**iii**) By minimizing $\mathcal{K}(\alpha, \mathbf{x}, \mathbf{y}; \mathbf{z}; \beta, \mu)$ and setting its gradient w.r.t. $\alpha$ to zero, we obtain $\alpha^* = \mathcal{U}(\mathbf{x}, \mathbf{y}; \mathbf{z}; \beta, \mu)/d(\mathbf{x})$, which leads to $\mathcal{L}(\mathbf{x}, \mathbf{y}; \mathbf{z}; \beta, \mu) = -1/\mathcal{K}(\alpha^*, \mathbf{x}, \mathbf{y}; \mathbf{z}; \beta, \mu)$. Thus, Formulations (3) and (4) are equivalent in a certain sense.*

## 5 GLOBAL CONVERGENCE

This section establishes the global convergence of both FADMM-D and FADMM-Q. We begin with an initial theoretical analysis applicable to both algorithms, followed by a detailed, separate analysis for each.

## 5.1 INITIAL THEORETICAL ANALYSIS

First, we impose the following condition on Algorithm 1.

**Assumption 5.1.** *Let* $\{\mathbf{x}^t\}_{t=0}^{\infty}$ *be generated by Algorithm 1. For all t, there exist constants* $\{\underline{d}, \overline{d}\}$ *such that* $0 < \underline{d} \le d(\mathbf{x}^t) \le \overline{d}$*, and constants* $\{\underline{F}, \overline{F}\}$ *such that* $0 < \underline{F} \le F(\mathbf{x}^t) \le \overline{F}$*.*

**Remark 5.2.** *(i) The existence of the upper bounds* $\overline{d}$ *and* $\overline{F}$ *is guaranteed by the boundedness of* $\mathbf{x}$ *and the continuity of the functions* $d(\mathbf{x})$ *and* $F(\mathbf{x})$ *within their respective effective domains. (ii) The lower bound condition* $\underline{d} > 0$ *is mild and widely utilized in the literature (Li & Zhang, 2022; Yuan, 2023; Boţ et al., 2023b). (iii) The lower bound condition* $\underline{F} > 0$ *is reasonable; otherwise, it suffices to solve the non-fractional problem:* $\min_{\mathbf{x}} u(\mathbf{x})$*.*

Second, we provide first-order optimality conditions for the solution $\mathbf{y}^{t+1}$.

**Lemma 5.3.** *(Proof in Appendix D.1,* First-Order Optimality Conditions) For all $t \ge 0$, we have: $\mathbf{z}^{t+1} = \nabla h_{\mu^t}(\mathbf{y}^{t+1}) \in \partial h(\check{\mathbf{y}}^{t+1})$.

Third, using the subsequent lemma, we establish an upper bound for the term $\|\mathbf{z}^{t+1} - \mathbf{z}^t\|_2^2$.

**Lemma 5.4.** *(Proof in Section D.2,* Controlling Dual using Primal) For all $t \ge 1$, we have: $\|\mathbf{z}^{t+1} - \mathbf{z}^t\|_2^2 \le 2\frac{(\beta^t)^2}{\chi^2}\|\mathbf{y}^{t+1} - \mathbf{y}^t\|_2^2 + 2C_h^2(\frac{6}{t} - \frac{6}{t+1})$.

Fourth, we show that the solution $\{\mathbf{x}^t, \mathbf{y}^t, \mathbf{z}^t\}$ is always bounded for all $t \ge 0$.

**Lemma 5.5.** *(Proof in Appendix D.3) Let* $t \ge 0$*. There exists universal constants* $\{\overline{x}, \overline{y}, \overline{z}\}$ *such that* $\|\mathbf{x}^t\| \le \overline{x}$*,* $\|\mathbf{z}^t\| \le \overline{z}$*, and* $\|\mathbf{y}^t\| \le \overline{y}$*.*

Fifth, the subsequent lemma establishes bounds for the term $\mathcal{U}(\mathbf{x}^t, \mathbf{y}^t, \mathbf{z}^t, \beta^t)$.

**Lemma 5.6.** *(Proof in Appendix D.4) For all* $t \ge 1$*, we have:* $\underline{F} \cdot \underline{d} - \underline{v}/\beta^t \le \mathcal{U}(\mathbf{x}^t, \mathbf{y}^t; \mathbf{z}^t; \beta^t, \mu^t) \le \overline{F} \cdot \overline{d} + \overline{v}/\beta^t$*, where* $\underline{v} \triangleq 8\overline{z}^2 + \frac{1}{2}\chi\overline{z}^2$ *and* $\overline{v} \triangleq 24\overline{z}^2$*.*

Given Lemma 5.6, we make the following additional assumption.

**Assumption 5.7.** *Assume* $\Delta \triangleq \beta^0 - \underline{v}/(\underline{F} \cdot \underline{d}) > 0$*.*

**Remark 5.8.** *(i) By Assumption 5.7, we have* $\beta^t \ge \beta^0 > \underline{v}/(\underline{F} \cdot \underline{d})$*, ensuring* $\mathcal{U}(\mathbf{x}^t, \mathbf{y}^t; \mathbf{z}^t; \beta^t, \mu^t) > 0$ *and* $\lambda^t > 0$ *for both* FADMM-D *and* FADMM-Q *for all* $t \ge 1$*. These inequalities are crucial to our analysis (see Inequalities (31), (38)). (ii) Assumption 5.7 is automatically satisfied when t is sufficiently large due to increasing penalty update rules. In practice,* $\beta^0 = 1$ *can be used.*

Finally, we demonstrate some critical properties for the parameters $\{\beta^t, \lambda^t, \alpha^t, \ell(\beta^t)\}$.

**Lemma 5.9.** *(Proof in Appendix D.5) Let* $t \ge 1$*. For both* FADMM-D *and* FADMM-Q*, we have:*

**(a)** $\beta^t \le \beta^{t+1} \le (1 + \xi)\beta^t$*.*
**(b)** *There exist positive constants* $\{\overline{\lambda}, \underline{\lambda}\}$ *such that* $\underline{\lambda} \le \lambda^t \le \overline{\lambda}$*.*
**(c)** *There exist positive constants* $\{\underline{\alpha}, \overline{\alpha}\}$ *such that* $\underline{\alpha} \le \alpha^t \le \overline{\alpha}$*.*
**(d)** *There exist positive constants* $\{\underline{\ell}, \overline{\ell}\}$ *such that* $\beta^t\underline{\ell} \le \ell(\beta^t) \le \beta^t\overline{\ell}$*.*

## 5.2 ANALYSIS FOR FADMM-D

This subsection provides the convergence analysis of FADMM-D.

We define $\varepsilon_x \triangleq \underline{\ell}(\theta - 1)/(2\overline{d}) > 0$, $\varepsilon_y \triangleq \{1 - 4(1 + \xi)/\chi^2\}/(2\overline{d}) > 0$, and $\varepsilon_z \triangleq \xi/(2\overline{d}) > 0$. We define the following sequence associated with a specific potential function:

$$\mathbb{P}^t \triangleq \underbrace{\mathcal{L}(\mathbf{x}^t, \mathbf{y}^t; \mathbf{z}^t; \beta^t, \mu^t)}_{\triangleq \mathbb{L}^t} + \underbrace{12(1 + \xi)C_h^2/(\beta^0\underline{d}t)}_{\triangleq \mathbb{T}^t} + \underbrace{C_h^2\mu^t/(2\underline{d})}_{\triangleq \mathbb{U}^t} .$$

We first present two useful lemmas regarding the decrease of the variables $\{\mathbf{x}\}$ and $\{\mathbf{y}, \mathbf{z}, \beta, \mu\}$.

**Lemma 5.10.** *(Proof in Section E.1,* Decrease on the Function $\mathcal{L}(\mathbf{x}, \mathbf{y}; \mathbf{z}; \beta, \mu)$ w.r.t. $\mathbf{x}$*) For all* $t \ge 1$*, we have:* $\varepsilon_x\beta^t\|\mathbf{x}^{t+1} - \mathbf{x}^t\|_2^2 + \mathcal{L}(\mathbf{x}^{t+1}, \mathbf{y}^t; \mathbf{z}^t; \beta^t, \mu^t) \le \mathcal{L}(\mathbf{x}^t, \mathbf{y}^t; \mathbf{z}^t; \beta^t, \mu^t)$*.*

**Lemma 5.11.** *(Proof in Section E.2,* Decrease on the Function $\mathcal{L}(\mathbf{x}, \mathbf{y}; \mathbf{z}; \beta, \mu)$ *w.r.t.* $\{\mathbf{y}, \mathbf{z}, \beta, \mu\}$*)* For all $t \geq 1$, we have: $\varepsilon_y \beta^t \|\mathbf{y}^{t+1} - \mathbf{y}^t\|_2^2 + \varepsilon_z \beta^t \|\mathbf{A}\mathbf{x}^{t+1} - \mathbf{y}^{t+1}\|_2^2 + \mathcal{L}(\mathbf{x}^{t+1}, \mathbf{y}^{t+1}; \mathbf{z}^{t+1}; \beta^{t+1}, \mu^{t+1}) - \mathcal{L}(\mathbf{x}^{t+1}, \mathbf{y}^t; \mathbf{z}^t; \beta^t, \mu^t) \leq \mathbb{U}^t + \mathbb{T}^t - \mathbb{U}^{t+1} - \mathbb{T}^{t+1}$.

The following lemma demonstrates a decrease property on a potential function.

**Lemma 5.12.** *(Proof in Section E.3,* Decrease on a Potential Function*)* We let $t \geq 1$. We define $\mathcal{E}^t \triangleq \beta^t \{\|\mathbf{x}^{t+1} - \mathbf{x}^t\|_2^2 + \|\mathbf{y}^{t+1} - \mathbf{y}^t\|_2^2 + \|\mathbf{A}\mathbf{x}^{t+1} - \mathbf{y}^{t+1}\|_2^2\}$. We have:

 (a) There exists a univeral positive constant $\underline{\mathbb{P}}$ such that $\mathbb{P}^t \geq \underline{\mathbb{P}}$.
 (b) It holds that $\min(\varepsilon_x, \varepsilon_y, \varepsilon_z)\mathcal{E}^t \leq \mathbb{P}^t - \mathbb{P}^{t+1}$.

The following theorem establishes the global convergence of FADMM-D.

**Theorem 5.13.** *(Proof in Section E.4,* Global Convergence*)* We let $t \geq 1$. We define $\mathcal{E}_+^t \triangleq \beta^t \{\|\mathbf{x}^{t+1} - \mathbf{x}^t\| + \|\mathbf{y}^{t+1} - \mathbf{y}^t\| + \|\mathbf{A}\mathbf{x}^{t+1} - \mathbf{y}^{t+1}\|\}$. We have: $\frac{1}{T} \sum_{t=1}^T \mathcal{E}_+^t \leq \mathcal{O}(T^{(p-1)/2})$. In other words, there exists an index $\bar{t}$ with $1 \leq \bar{t} \leq T$ such that $\mathcal{E}_+^{\bar{t}} \leq \mathcal{O}(T^{(p-1)/2})$.

**Remark 5.14.** *(i) With the choice* $p \in (0, 1)$*, Theorem (5.13) implies that* $\mathcal{E}_+^t$ *converges to 0 in the ergodic sense. (ii) The convergence* $\mathcal{E}_+^t \to 0$ *is significantly stronger than the convergence* $\|\mathbf{x}^{t+1} - \mathbf{x}^t\| + \|\mathbf{y}^{t+1} - \mathbf{y}^t\| + \|\mathbf{A}\mathbf{x}^{t+1} - \mathbf{y}^{t+1}\| \to 0$ *as* $\{\beta^t\}_{t=1}^\infty$ *is increasing.*

### 5.3 ANALYSIS FOR FADMM-Q

This subsection presents the convergence analysis of FADMM-Q.

We define $\varepsilon_x \triangleq \frac{1}{2}\underline{\alpha}^2 \ell(\theta - 1) > 0$, $\varepsilon_y \triangleq \frac{1}{2}\underline{\alpha}^2 \{1 - 4(1 + \xi)/(\chi^2)\}$, and $\varepsilon_z \triangleq \frac{1}{2}\xi\underline{\alpha}^2 > 0$. We define the following sequence associated with a specific potential function:

$$\mathbb{P}^t \triangleq \underbrace{\mathcal{K}(\alpha^t, \mathbf{x}^t, \mathbf{y}^t; \mathbf{z}^t; \beta^t, \mu^t)}_{\triangleq \mathbb{K}^t} + \underbrace{12\overline{\alpha}^2(1 + \xi)C_h^2/(\beta^0 t)}_{\triangleq \mathbb{T}^t} + \underbrace{\tfrac{1}{2}\overline{\alpha}^2 C_h^2 \mu^t}_{\triangleq \mathbb{U}^t}.$$

The following two lemmas establish the decrease of the variables $\{\lambda, \mathbf{x}\}$ and $\{\mathbf{y}, \mathbf{z}, \beta, \mu\}$.

**Lemma 5.15.** *(Proof in Section E.5,* Decrease on the Function $\mathcal{K}(\lambda, \mathbf{x}, \mathbf{y}; \mathbf{z}; \beta, \mu)$ *w.r.t.* $\lambda$ and $\mathbf{x}$*)* For all $t \geq 1$, we have: $\mathcal{K}(\lambda^{t+1}, \mathbf{x}^{t+1}, \mathbf{y}^t; \mathbf{z}^t; \beta^t, \mu^t) + \varepsilon_x \beta^t \|\mathbf{x}^{t+1} - \mathbf{x}^t\|_2^2 \leq \mathcal{K}(\lambda^t, \mathbf{x}^t, \mathbf{y}^t; \mathbf{z}^t; \beta^t, \mu^t)$.

**Lemma 5.16.** *(Proof in Section E.6,* Decrease on the Function $\mathcal{K}(\lambda, \mathbf{x}, \mathbf{y}; \mathbf{z}; \beta, \mu)$ *w.r.t.* $\{\mathbf{y}, \mathbf{z}, \beta, \mu\}$*)* For all $t \geq 1$, we have: $\varepsilon_y \beta^t \|\mathbf{y}^{t+1} - \mathbf{y}^t\|_2^2 + \varepsilon_z \beta^t \|\mathbf{A}\mathbf{x}^{t+1} - \mathbf{y}^{t+1}\|_2^2 + \mathcal{K}(\lambda^{t+1}, \mathbf{x}^{t+1}, \mathbf{y}^{t+1}; \mathbf{z}^{t+1}; \beta^{t+1}, \mu^{t+1}) - \mathcal{K}(\lambda^{t+1}, \mathbf{x}^{t+1}, \mathbf{y}^t; \mathbf{z}^t; \beta^t, \mu^t) \leq \mathbb{U}^t + \mathbb{T}^t - \mathbb{U}^{t+1} - \mathbb{T}^{t+1}$.

The following lemma shows a decrease property on a potential function.

**Lemma 5.17.** *(Proof in Section E.7,* Decrease on a Potential Function*)* We let $t \geq 1$. We define $\mathcal{E}^t \triangleq \beta^t \{\|\mathbf{x}^{t+1} - \mathbf{x}^t\|_2^2 + \|\mathbf{y}^{t+1} - \mathbf{y}^t\|_2^2 + \|\mathbf{A}\mathbf{x}^{t+1} - \mathbf{y}^{t+1}\|_2^2\}$. We have:

 (a) There exists a univeral positive constant $\underline{\mathbb{P}}$ such that $\mathbb{P}^t \geq \underline{\mathbb{P}}$.
 (b) It holds that $\min(\varepsilon_x, \varepsilon_y, \varepsilon_z)\mathcal{E}^t \leq \mathbb{P}^t - \mathbb{P}^{t+1}$.

The following theorem establishes the global convergence of FADMM-Q.

**Theorem 5.18.** *(Proof in Section E.8,* Global Convergence*)* We let $t \geq 1$. We define $\mathcal{E}_+^t \triangleq \beta^t \{\|\mathbf{x}^{t+1} - \mathbf{x}^t\| + \|\mathbf{y}^{t+1} - \mathbf{y}^t\| + \|\mathbf{A}\mathbf{x}^{t+1} - \mathbf{y}^{t+1}\|\}$. We have: $\frac{1}{T} \sum_{t=1}^T \mathcal{E}_+^t \leq \mathcal{O}(T^{(p-1)/2})$. In other words, there exists an index $\bar{t}$ with $1 \leq \bar{t} \leq T$ such that $\mathcal{E}_+^{\bar{t}} \leq \mathcal{O}(T^{(p-1)/2})$.

**Remark 5.19.** *Theorem 5.18 is analogous to Theorem 5.13, with* $\mathcal{E}_+^{\bar{t}}$ *converging to 0 in the ergodic sense.*

## 6 ITERATION COMPLEXITY

This section examines the iteration complexity of FADMM for converging to critical points.

First, we introduce the notion of approximate critical points for the problem (1), which will play an important role in our analysis.

**Definition 6.1.** *($\epsilon$-Critical Point) We define* $\mathrm{Crit}(\mathbf{x}^+, \mathbf{x}, \mathbf{y}^+, \mathbf{y}, \mathbf{z}^+, \mathbf{z}) \triangleq \|\mathbf{x}^+ - \mathbf{x}\| + \|\mathbf{y}^+ - \mathbf{y}\| + \|\mathbf{z}^+ - \mathbf{z}\| + \|\mathbf{A}\mathbf{x}^+ - \mathbf{y}^+\| + \|\partial h(\mathbf{y}^+) - \mathbf{z}^+\| + \|\partial \delta(\mathbf{x}^+) + \nabla f(\mathbf{x}^+) - \partial g(\mathbf{x}) + \mathbf{A}^\mathsf{T}\mathbf{z}^+ - \varphi(\mathbf{x}, \mathbf{y})\partial d(\mathbf{x})\|$, *and* $\varphi(\mathbf{x}, \mathbf{y}) = \{f(\mathbf{x}) + \delta(\mathbf{x}) - g(\mathbf{x}) + h(\mathbf{y})\}/d(\mathbf{x})$. *A solution* $(\bar{\mathbf{x}}^+, \bar{\mathbf{x}}, \bar{\mathbf{y}}^+, \bar{\mathbf{y}}, \bar{\mathbf{z}}^+, \bar{\mathbf{z}})$ *is a critical point of Problem (1) if:*

$$\mathrm{Crit}(\bar{\mathbf{x}}^+, \bar{\mathbf{x}}, \bar{\mathbf{y}}^+, \bar{\mathbf{y}}, \bar{\mathbf{z}}^+, \bar{\mathbf{z}}) \leq \epsilon.$$

**Remark 6.2.** *(i) If* $\epsilon = 0$, *Definition 6.1 simplifies to the (exact) critical point as described in Definition 3.6. (ii) The study in (Boţ et al., 2023b) introduces a notation of approximate limiting subdifferential to define the $\epsilon$-critical point, whereas we simply employ consecutive iterations for its definition.*

Finally, we establish the iteration complexity of FADMM as follows.

**Theorem 6.3.** *(Proof in Section F.1, Iteration Complexity for Both* FADMM-D *and* FADMM-Q*) We define* $\mathcal{W}^t \triangleq \{\mathbf{x}^{t+1}, \mathbf{x}^t, \breve{\mathbf{y}}^{t+1}, \mathbf{y}^t, \mathbf{z}^{t+1}, \mathbf{z}^t\}$. *Let the sequence* $\{\mathcal{W}^t\}_{t=0}^T$ *be generated by* FADMM-D *or* FADMM-Q*. If* $p \in (0, 1)$, *we have:*

$$\mathrm{Crit}(\mathcal{W}^t) \leq \mathcal{O}(T^{-p}) + \mathcal{O}(T^{(p-1)/2}).$$

*In particular, with the choice* $p = 1/3$, *we have* $\mathrm{Crit}(\mathcal{W}^t) \leq \mathcal{O}(T^{-1/3})$. *In other words, there exists* $1 \leq \bar{t} \leq T$ *such that:* $\mathrm{Crit}(\mathcal{W}^{\bar{t}}) \leq \epsilon$, *provided that* $T \geq \mathcal{O}(\frac{1}{\epsilon^3})$.

**Remark 6.4.** *(i) To our knowledge, Theorem 6.3 is the first complexity result for ADMM applied to this class of fractional programs, and it matches the iteration bound of smoothing proximal gradient methods (Beck & Rosset, 2023; Böhm & Wright, 2021). (ii) The point* $\{\mathbf{x}^{t+1}, \mathbf{x}^t, \breve{\mathbf{y}}^{t+1}, \mathbf{y}^t, \mathbf{z}^{t+1}, \mathbf{z}^t\}$ *rather than the point* $\{\mathbf{x}^{t+1}, \mathbf{x}^t, \mathbf{y}^{t+1}, \mathbf{y}^t, \mathbf{z}^{t+1}, \mathbf{z}^t\}$ *serves as an approximate critical point of Problem (1) in Theorem 6.3.*

## 7 EXPERIMENTS

This section evaluates the effectiveness of FADMM-D and FADMM-Q on sparse FDA. Additional experiments on robust Sharpe ratio minimization, and robust sparse recovery, please refer to Appendix Section I.

▶ **Compared Methods**. We compare FADMM-D and FADMM-Q with three state-of-the-art general-purpose algorithms that solve Problem (1): (***i***) the Subgradient Projection Method (SPM) (Li et al., 2021), (***ii***) the Smoothing Proximal Gradient Method (SPGM) (Beck & Rosset, 2023; Yuan, 2024a; Bian & Chen, 2020; Böhm & Wright, 2021), and (***iii***) the Full Splitting Algorithm (FSA) (Boţ et al., 2023b). For FADMM-D and FADMM-Q, if we fix $\mathbf{z}^t = \mathbf{0}$ and $\mu^t = 0$ for all $t$ in Algorithm 1, they respectively reduce to two SPGM variants: SPGM-D and SPGM-Q. For FSA, we adapt the algorithm from (Boţ et al., 2023b) to our notation to address Problem (1). Implementation details of FSA are provided in Appendix Section H. We examine two fixed small step sizes, $\gamma \in (10^{-3}, 10^{-4})$, leading to two variants: FSA-I and FSA-II.

▶ **Experimental Settings**. For all SPGM and FADMM, we consider the default parameter settings $(\xi, \theta, p, \chi) = (1/2, 1.01, 1/3, 2\sqrt{1+\xi} + 10^{-14})$. For SPM, we use the default diminishing step size $\eta^t = 1/\beta^t$, where $\beta^t$ is the same penalty parameter as in SPGM and FADMM. For all algorithms, we initialize their solutions drawn from a standard Gaussian distribution. All methods are implemented in MATLAB on an Intel 2.6 GHz CPU with 64 GB RAM. We incorporate a set of 8 datasets into our experiments, comprising both randomly generated and publicly available real-world data. Appendix Section I describes how to generate the data used in the experiments. We compare the objective values for all methods after running $t$ seconds with $t = 20$. The corresponding MATLAB code is available on the author's research webpage.

▶ **Experimental Results on Sparse FDA**. We consider solving Problem (2) using the following parameters $r = 20$, $k = 0.1 \times n \times r$, and $\rho \in \{10, 100, 1000, 10000\}$. According to the exact penalty theory (Bi et al., 2014; Bi & Pan, 2016), a reasonable value for $\beta^t$ is expected to be at least larger than $\rho$. We set $\beta^0 = 100\rho$, which appears to work well. The experimental results for $\rho \in \{10, 1000\}$ are presented in Figures 1 and 2, while the results for $\rho \in \{100, 10000\}$ are provided in Appendix Section I. Based on these results, we draw the following conclusions. (***i***) SPM tends to be less efficient in comparison to other methods. This is primarily because, in the case of a sparse solution, the subdifferential set of the objective function is large and provides a poor approximation of the

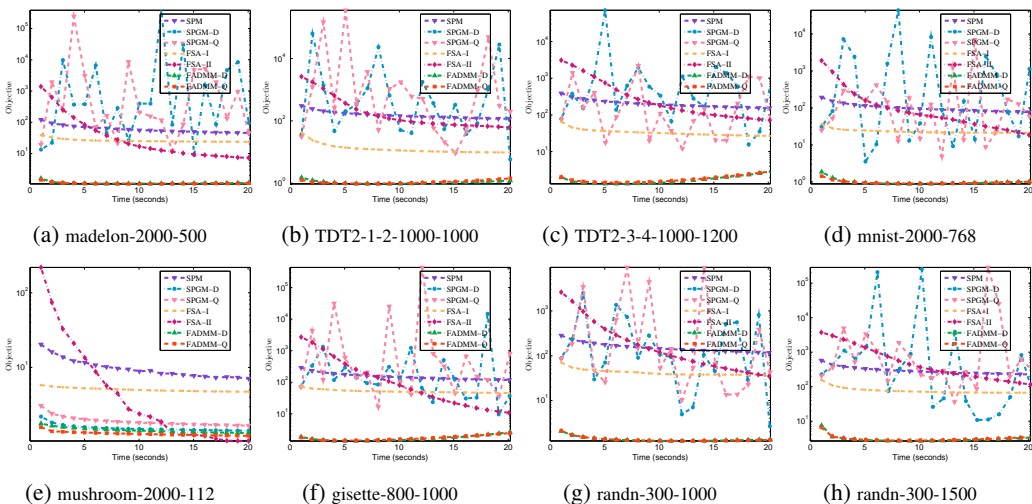

Figure 1: Results on sparse FDA on different datasets with $\rho = 10$.

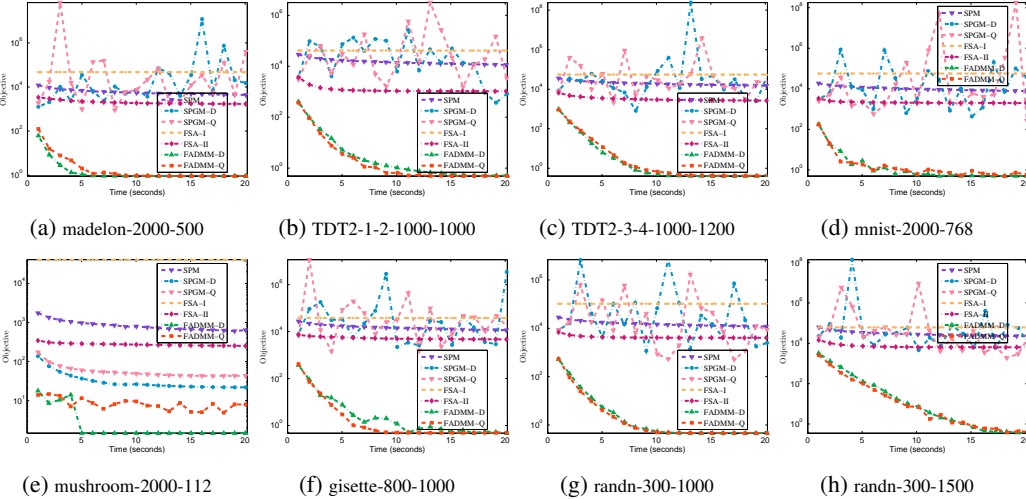

Figure 2: Results on sparse FDA on different datasets with $\rho = 1000$.

(negative) descent direction. (**ii**) SPGM-D and SPGM-Q, utilizing a variable smoothing strategy, generally demonstrates better performance than SPM. (**iii**) The proposed FADMM-D and FADMM-Q generally exhibit similar performance, both achieving the lowest objective function values among all the methods examined. This supports the widely accepted view that primal-dual methods are generally more robust and faster than primal-only methods. (**iv**) The proposed FADMM-D and FADMM-Q still outperform FSA, which uses a sufficiently small step size to ensure convergence.

## 8 CONCLUSIONS

In this paper, we introduce FADMM, the first ADMM algorithm designed to solve general structured fractional minimization problems. Our approach integrates Nesterov's smoothing technique (equivalent to the Moreau envelope smoothing technique) into the algorithm's updates to guarantee convergence. We present two specific variants of FADMM: one using Dinkelbach's parametric method (FADMM-D) and the other using the quadratic transform method (FADMM-Q). Additionally, we establish the iteration complexity of FADMM for convergence to approximate critical points. Finally, we validate the effectiveness of our methods through experimental results.

## ACKNOWLEDGMENTS

This work was supported by NSFC (12271278, 61772570), and Guangdong Natural Science Funds for Distinguished Young Scholar (2018B030306025).

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

# Appendix

The appendix is organized as follows.

Appendix A presents notation, technical preliminaries, and relevant lemmas.

Appendix B provides additional motivating applications.

Appendix C contains proofs for Section 3.

Appendix D contains proofs for Section 4.

Appendix E contains proofs for Section 5.

Appendix F contains proofs for Section 6.

Appendix G explains the computation of the proximal operator.

Appendix I demonstrates additional experimental details and results.

## A    NOTATIONS, TECHNICAL PRELIMINARIES, AND RELEVANT LEMMAS

### A.1    NOTATIONS

In this paper, lowercase boldface letters signify vectors, while uppercase letters denote real-valued matrices. The following notations are utilized throughout this paper.

- $[n]$: $\{1, 2, ..., n\}$
- $\|\mathbf{x}\|$: Euclidean norm: $\|\mathbf{x}\| = \|\mathbf{x}\|_2 = \sqrt{\langle \mathbf{x}, \mathbf{x} \rangle}$
- $\mathbf{X}^\mathsf{T}$ : the transpose of the matrix $\mathbf{X}$
- $\mathrm{vec}(\mathbf{X})$ : the vector formed by stacking the column vectors of $\mathbf{X}$ with $\mathrm{vec}(\mathbf{X}) \in \mathbb{R}^{nr \times 1}$
- $\mathrm{mat}(\mathbf{x})$ : convert $\mathbf{x} \in \mathbb{R}^{nr \times 1}$ into a matrix with $\mathrm{mat}(\mathrm{vec}(\mathbf{X})) = \mathbf{X}$ with $\mathrm{mat}(\mathbf{x}) \in \mathbb{R}^{n \times r}$
- $\mathbf{0}_{n,r}$ : a zero matrix of size $n \times r$; the subscript is omitted sometimes
- $\mathbf{I}_r$: identity matrix with $\mathbf{I}_r \in \mathbb{R}^{r \times r}$
- $\mathbf{X} \succeq \mathbf{0}$ : matrix $\mathbf{X}$ is symmetric positive semidefinite
- $\|\mathbf{X}\|_\mathsf{F}$ : Frobenius norm: $(\sum_{ij} \mathbf{X}_{ij}^2)^{1/2}$
- $\partial f(\mathbf{x})$ : limiting subdifferential of $f(\mathbf{x})$ at $\mathbf{x}$
- $\iota_\Omega(\mathbf{x})$ : the indicator function of a set $\Omega$ with $\iota_\Omega(\mathbf{x}) = 0$ if $\mathbf{x} \in \Omega$ and otherwise $+\infty$
- $\mathrm{tr}(\mathbf{A})$ : Sum of the elements on the main diagonal $\mathbf{A}$ with $\mathrm{tr}(\mathbf{A}) = \sum_i \mathbf{A}_{i,i}$
- $\langle \mathbf{X}, \mathbf{Y} \rangle$ : Euclidean inner product, i.e., $\langle \mathbf{X}, \mathbf{Y} \rangle = \sum_{ij} \mathbf{X}_{ij} \mathbf{Y}_{ij}$
- $\|\mathbf{X}\|$ : Operator/Spectral norm: the largest singular value of $\mathbf{X}$
- $\mathrm{dist}(\Xi, \Xi')$ : the distance between two sets with $\mathrm{dist}(\Xi, \Xi') \triangleq \inf_{\mathbf{x} \in \Xi, \mathbf{x}' \in \Xi'} \|\mathbf{x} - \mathbf{x}'\|$
- $\|\partial F(\mathbf{x})\|$: $\|\partial F(\mathbf{x})\| = \inf_{\mathbf{z} \in \partial F(\mathbf{x})} \|\mathbf{z}\|_\mathsf{F} = \mathrm{dist}(\mathbf{0}, \partial F(\mathbf{x}))$
- $\|\mathbf{X}\|_{[k]}$: $\ell_1$ norm of the $k$ largest (in magnitude) elements of the matrix $\mathbf{X}$
- $\mathbf{x}_i$: the $i$-th element of vector $\mathbf{x}$
- $\mathbf{X}_{i,j}$ or $\mathbf{X}_{ij}$ : the $(i^{\mathrm{th}}, j^{\mathrm{th}})$ element of matrix $\mathbf{X}$
- $\mathcal{P}_\Omega(\mathbf{X}')$ : Orthogonal projection of $\mathbf{X}'$ with $\mathcal{P}_\Omega(\mathbf{X}') = \arg\min_{\mathbf{X} \in \Omega} \|\mathbf{X}' - \mathbf{X}\|_\mathsf{F}^2$
- $\dot{\partial}\varphi(\mathbf{x}) \pm \ddot{\partial}\varphi(\mathbf{x})$: standard Minkowski addition (or subtraction) between sets $\dot{\partial}\varphi(\mathbf{x})$ and $\ddot{\partial}\varphi(\mathbf{x})$

### A.2    TECHNICAL PRELIMINARIES ON NONSMOOTH NONCONVEX OPTIMIZATION

We present various techniques in convex analysis, nonsmooth analysis, and nonconvex analysis (Mordukhovich et al., 2006; Mordukhovich, 2006; Rockafellar & Wets., 2009; Bertsekas, 2015), encompassing conjugate functions, weakly convex functions, the Fréchet subdifferential, limiting subdifferential, and rules for sum and quotient in the Fréchet subdifferential context.

For any extended real-valued (not necessarily convex) function $f : \mathbb{R}^n \to (-\infty, +\infty]$, we denote by $\mathrm{dom}(f) := \{\mathbf{x} \in \mathbb{R}^n : f(\mathbf{x}) < +\infty\}$ its *effective domain*. The function $f(\mathbf{x})$ is *proper* if $\mathrm{dom}(f) \neq \emptyset$. The function $f(\mathbf{x})$ is lower-semicontinuous at some point $\dot{\mathbf{x}} \in \mathbb{R}^n$ if $\liminf_{\mathbf{x} \to \dot{\mathbf{x}}} f(\mathbf{x}) \geq f(\dot{\mathbf{x}})$.

• **Conjugate Functions**. For a proper, convex, lower semicontinuous function $h(\mathbf{y}) : \mathbb{R}^m \mapsto \mathbb{R}$, we denote the *(Fenchel) conjugate function* of $h(\mathbf{y})$ as $h^*(\mathbf{y}) \triangleq \sup_{\mathbf{v} \in \mathrm{dom}(h)} \{\mathbf{y}^\mathsf{T}\mathbf{v} - h(\mathbf{v})\}$, and it follows that $h^{**}(\mathbf{y}) = h(\mathbf{y}) = \sup_{\mathbf{v} \in \mathrm{dom}(h^*)} \{\mathbf{y}^\mathsf{T}\mathbf{v} - h^*(\mathbf{v})\}$, where $h^{**}(\mathbf{y})$ is called the biconjugate function. For any $\mu > 0$, we have that $(\mu h)^*(\mathbf{y}) = \sup_{\mathbf{x} \in \mathrm{dom}(h)} \{\langle \mathbf{y}, \mathbf{x} \rangle - \mu h(\mathbf{x})\} = \mu h^*(\frac{\mathbf{y}}{\mu})$. For any $\mathbf{x}, \mathbf{y} \in \mathbb{R}^n$, the following statements are equivalent (see (Rockafellar & Wets., 2009), Proposition 11.3): $\langle \mathbf{x}, \mathbf{y} \rangle = h(\mathbf{x}) + h^*(\mathbf{y}) \Leftrightarrow \mathbf{y} \in \partial h(\mathbf{x}) \Leftrightarrow \mathbf{x} \in \partial h^*(\mathbf{y})$. The conjugate of the support function of a closed convex set $\Omega$ is its indicator function, i.e., $h(\mathbf{y}) = \sup_{\mathbf{v} \in \Omega} \langle \mathbf{v}, \mathbf{y} \rangle$, and $h^*(\mathbf{v}) = \iota_\Omega(\mathbf{v})$. Typical nonsmooth functions for $h(\mathbf{y})$ include $\{\|\mathbf{y}\|_1, \|\max(0, \mathbf{y})\|_1, \|\mathbf{y}\|_\infty\}$, with their respective conjugate functions $h^*(\mathbf{y})$ being $\{\iota_{[-1,1]^m}(\mathbf{y}), \iota_{[0,1]^m}(\mathbf{y}), \iota_{\|\mathbf{y}\|_1 \leq 1}(\mathbf{y})\}$.

• **Weakly Convex Functions**. The function $d(\mathbf{x})$ is weakly convex if a constant $W_d \geq 0$ exists, making $d(\mathbf{x}) + \frac{W_d}{2}\|\mathbf{x}\|_2^2$ convex, with the smallest such $W_d$ known as the modulus of weak convexity. Weakly convex functions constitute a diverse class of functions which covers convex functions, differentiable functions whose gradient is Lipschitz continuous, as well as compositions of convex, Lipschitz-continuous functions with $C^1$-smooth mappings that have Lipschitz continuous Jacobians (Drusvyatskiy & Paquette, 2019).

• **Fréchet Subdifferential and Limiting (Fréchet) Subdifferential**. The *Fréchet subdifferential* of $F$ at $\ddot{\mathbf{x}} \in \mathrm{dom}(F)$, denoted as $\widehat{\partial}F(\ddot{\mathbf{x}})$, is defined as

$$\widehat{\partial}F(\ddot{\mathbf{x}}) \triangleq \left\{ \mathbf{v} \in \mathbb{R}^n : \lim_{\mathbf{x} \to \ddot{\mathbf{x}}} \inf_{\mathbf{x} \neq \ddot{\mathbf{x}}} \frac{F(\mathbf{x}) - F(\ddot{\mathbf{x}}) - \langle \mathbf{v}, \mathbf{x} - \ddot{\mathbf{x}} \rangle}{\|\mathbf{x} - \ddot{\mathbf{x}}\|} \geq 0 \right\}.$$

The *limiting subdifferential* of $F(\mathbf{x})$ at $\dot{\mathbf{x}} \in \mathrm{dom}(F)$, denoted as $\partial F(\dot{\mathbf{x}})$, is defined as

$$\partial F(\dot{\mathbf{x}}) \triangleq \left\{ \mathbf{v} \in \mathbb{R}^n : \exists \mathbf{x}^k \to \dot{\mathbf{x}}, F(\mathbf{x}^k) \to F(\dot{\mathbf{x}}), \mathbf{v}^k \in \widehat{\partial}F(\mathbf{x}^k) \to \mathbf{v}, \forall k \right\}.$$

It is straightforward to verify that $\widehat{\partial}F(\mathbf{x}) \subseteq \partial F(\mathbf{x})$, $\widehat{\partial}(\alpha F)(x) = \alpha\widehat{\partial}F(x)$ and $\partial(\alpha F)(x) = \alpha\partial F(x)$ hold for any $x \in \mathrm{dom}(F)$ and $\alpha > 0$. Additionally, if $F(\cdot)$ is differentiable at $\mathbf{x}$, then $\widehat{\partial}F(\mathbf{x}) = \partial F(\mathbf{x}) = \{\nabla F(\mathbf{x})\}$ with $\nabla F(\mathbf{x})$ being the gradient of $F(\cdot)$ at $\mathbf{x}$; when $F(\cdot)$ is convex, $\widehat{\partial}F(\mathbf{x})$ and $\partial F(\mathbf{x})$ reduce to the classical subdifferential for convex functions, i.e., $\widehat{\partial}F(\mathbf{x}) = \partial F(\mathbf{x}) = \{\mathbf{v} \in \mathbf{R}^n : F(\mathbf{z}) - F(\mathbf{x}) - \langle \mathbf{v}, \mathbf{z} - \mathbf{x} \rangle \geq 0, \forall \mathbf{z} \in \mathbb{R}^n\}$.

• **Sum and Quotient Rules for the Fréchet Subdifferential**. First, we examine sum rules for the Fréchet subdifferential. Let $\varphi_1, \varphi_2 : \mathbb{R}^n \to (-\infty, +\infty]$ be proper, closed functions, and let $\mathbf{x} \in \mathrm{dom}(\varphi_1) \cap \mathrm{dom}(\varphi_2)$. Then, $\widehat{\partial}\varphi_1(\mathbf{x}) + \widehat{\partial}\varphi_2(\mathbf{x}) \subseteq \widehat{\partial}(\varphi_1 + \varphi_2)(\mathbf{x})$, where equality holds if $\varphi_1$ or $\varphi_2$ is differentiable at $x$ (See Corollary 10.9 in (Rockafellar & Wets., 2009)). Moreover, $\partial(\varphi_1 + \varphi_2)(\mathbf{x}) \subseteq \partial\varphi_1(\mathbf{x}) + \partial\varphi_2(\mathbf{x})$ holds when $\varphi_1$ or $\varphi_2$ is locally Lipschitz continuous at $\mathbf{x}$, and it holds with equality when $\varphi_1$ or $\varphi_2$ is continuously differentiable at $\mathbf{x}$ (See Exercise 10.10 in (Rockafellar & Wets., 2009)).

We review the quotient rules for the Fréchet subdifferential. The concept of calmness (Rockafellar & Wets., 2009) plays an important role in our analysis. A proper function $g : \mathbb{R}^n \to (-\infty, +\infty]$ is said to be *calm* at $\mathbf{x} \in \mathrm{dom}(g)$ if there exist $\varepsilon > 0$ and $\kappa > 0$ such that $|g(\mathbf{x}) - g(\mathbf{x}')| \leq \kappa\|\mathbf{x} - \mathbf{x}'\|$ for all $\mathbf{x}' \in \mathcal{B}(\mathbf{x}, \varepsilon) \triangleq \{\mathbf{z} \in \mathbb{R}^n : \|\mathbf{z} - \mathbf{x}\| < \varepsilon\}$. Many convex functions, including Lipschitz continuous functions, satisfy the calmness condition.

We define $\varphi : \mathbb{R}^n \mapsto (-\infty, +\infty]$ at $\mathbf{x} \in \mathbb{R}^n$ as:

$$\varphi(\mathbf{x}) := \begin{cases} \frac{\varphi_1(\mathbf{x})}{\varphi_2(\mathbf{x})}, & \text{if } \mathbf{x} \in \mathrm{dom}(\varphi_1) \text{ and } \varphi_2(\mathbf{x}) \neq 0, \\ +\infty, & \text{else.} \end{cases}$$

The following lemma concerns the quotient rules for the Fréchet subdifferential.

**Lemma A.1.** *Let $\varphi_1 : \mathbb{R}^n \to (-\infty, +\infty]$ and $\varphi_2 : \mathbb{R}^n \to \mathbb{R}$ be two functions which are finite at $\mathbf{x}$ with $\varphi_2(\mathbf{x}) > 0$. We denote $\alpha_1 := \varphi_1(\mathbf{x})$ and $\alpha_2 := \varphi_2(\mathbf{x})$. Suppose that $\varphi_1$ is closed and continuous at $\mathbf{x}$ relative to $\mathrm{dom}(\varphi_1)$, that $\varphi_2$ is calm at $\mathbf{x}$. We have the following results:*

**(a)** *It holds that: $\widehat{\partial}\varphi(\mathbf{x}) = \frac{1}{\alpha_2^2}\{\widehat{\partial}(\alpha_2\varphi_1 - \alpha_1\varphi_2)(\mathbf{x})\}$.*

**(b)** *If $\varphi_2(\mathbf{x})$ is differentiable at $\mathbf{x}$, then $\widehat{\partial}\varphi(\mathbf{x}) = \frac{1}{\alpha_2^2}\{\alpha_2\widehat{\partial}\varphi_1(\mathbf{x}) - \alpha_1\nabla\varphi_2(\mathbf{x})\}$.*

**(c)** *If $\alpha_1 \geq 0$ and $\varphi_2(\mathbf{x})$ is convex, then $\widehat{\partial}\varphi(\mathbf{x}) \subseteq \frac{1}{\alpha_2^2}\{\widehat{\partial}(\alpha_2\varphi_1)(\mathbf{x}) - \alpha_1\widehat{\partial}\varphi_2(\mathbf{x})\} \subseteq \frac{1}{\alpha_2}\{\partial(\varphi_1)(\mathbf{x}) - \frac{\alpha_1}{\alpha_2}\partial\varphi_2(\mathbf{x})\}$.*

*Proof.* Refer to Proposition 2.2 in (Li & Zhang, 2022) and Lemma 2.1 in (Boţ et al., 2023a). We omit the proofs for conciseness.

$\square$

## A.3 RELEVANT LEMMAS

We present some useful lemmas that will be used subsequently.

**Lemma A.2.** (**Extended Moreau Decomposition**) Assume $h(\mathbf{y})$ is convex. For any $\mu > 0$ and $\mathbf{b} \in \mathbb{R}^m$, we define $\mathrm{Prox}(\mathbf{b}; h, \mu) \triangleq \arg\min_{\mathbf{y}} h(\mathbf{y}) + \frac{1}{2\mu}\|\mathbf{b} - \mathbf{y}\|_2^2$. It holds that: $\mathbf{b} = \mathrm{Prox}(\mathbf{b}; h, \mu) + \mu\mathrm{Prox}(\frac{\mathbf{b}}{\mu}; h^*, \frac{1}{\mu})$.

*Proof.* The conclusion of this lemma can be found in (Nesterov, 2013; Parikh et al., 2014). For completeness, we present the proof.

We define $\bar{\mathbf{y}} \triangleq \mathrm{Prox}(\mathbf{b}; h, \mu)$. We derive:

$$
\begin{aligned}
\mathbf{b} - \bar{\mathbf{y}} \in \partial(\mu h)(\bar{\mathbf{y}}) \quad &\overset{①}{\Leftrightarrow} \quad \bar{\mathbf{y}} \in \partial((\mu h)^*)(\mathbf{b} - \bar{\mathbf{y}}) \\
&\Leftrightarrow \quad \mathbf{b} - (\mathbf{b} - \bar{\mathbf{y}}) \in \partial((\mu h)^*)(\mathbf{b} - \bar{\mathbf{y}}) \\
&\Leftrightarrow \quad \mathbf{b} - \bar{\mathbf{y}} = \mathrm{Prox}(\mathbf{b}; (\mu h)^*, 1) \\
&\overset{②}{\Leftrightarrow} \quad \mathbf{b} = \mathrm{Prox}(\mathbf{b}; h, \mu) + \mu\mathrm{Prox}(\tfrac{\mathbf{b}}{\mu}; h^*, \tfrac{1}{\mu}),
\end{aligned}
$$

where step ① uses the definition of the conjugate function, and the property of the subdifferential that $\mathbf{v} \in \partial h(\mathbf{y}) \Leftrightarrow \mathbf{y} \in \partial h^*(\mathbf{v})$; step ② uses the following derivations that: $\arg\min_{\mathbf{y}}(\mu h)^*(\mathbf{y}) + \frac{1}{2}\|\mathbf{y} - \mathbf{b}\|_2^2 = \arg\min_{\mathbf{y}} \mu h^*(\mathbf{y}/\mu) + \frac{1}{2}\|\mathbf{y} - \mathbf{b}\|_2^2 = \mu\arg\min_{\mathbf{y}'} h^*(\mathbf{y}') + \frac{\mu}{2}\|\mathbf{y}' - \mathbf{b}/\mu\|_2^2 = \mu\mathrm{Prox}(\frac{\mathbf{b}}{\mu}; h^*, \frac{1}{\mu})$.

$\square$

**Lemma A.3.** *Let $\beta^t \triangleq \beta^0(1 + \xi t^p)$ and $\mu^t \propto \frac{1}{\beta^t}$, where $\beta^0 \geq 0$ and $\xi \in (0, 1)$. For any integer $t \geq 1$, we have: $(\frac{\mu^{t-1}}{\mu^t} - 1)^2 \leq \frac{6}{t} - \frac{6}{t+1}$.*

*Proof.* We define $h(t) \triangleq t^p$, where $p \in (0, 1)$.

Given the concavity of $h(t)$, it follows that: $h(y) - h(x) \leq \langle y - x, \nabla h(x)\rangle$. Letting $x = t - 1$ and $y = t$, for all $t > 1$, we have:

$$t^p - (t-1)^p \leq p(t-1)^{p-1}. \tag{8}$$

**Part (a)**. When $t = 1$, we have: $(\frac{\mu^{t-1}}{\mu^t} - 1)^2 - (\frac{6}{t} - \frac{6}{t+1}) = (\frac{\beta^1}{\beta^0} - 1)^2 - 3 = \xi^2 - 3 \leq -2 < 0$.

**Part (b)**. When $t \geq 2$, we derive:

$$
\begin{aligned}
(\tfrac{\mu^{t-1}}{\mu^t} - 1)^2 \quad &\overset{①}{=} \quad (\tfrac{\beta^t}{\beta^{t-1}} - 1)^2 \quad \overset{②}{=} \quad (\tfrac{1+\xi t^p}{1+\xi(t-1)^p} - 1)^2 \quad \overset{③}{\leq} \quad (\tfrac{t^p}{(t-1)^p} - 1)^2 \\
&= \quad (\tfrac{t^p - (t-1)^p}{(t-1)^p})^2 \quad \overset{④}{\leq} \quad ((t-1)^{p-1-p})^2 = \tfrac{1}{(t-1)^2} \quad \overset{⑤}{\leq} \quad \tfrac{6}{t} - \tfrac{6}{t+1},
\end{aligned}
$$

where step ① uses $\mu^t \propto \frac{1}{\beta^t}$; step ② uses $\beta^t = \beta^0(1 + \xi t^p)$; step ③ uses $1 < \frac{1+\xi t^p}{1+\xi(t-1)^p} \leq \frac{\xi t^p}{\xi(t-1)^p}$; step ④ uses Inequality (8) and $p \leq 1$; step ⑤ uses $\frac{1}{(t-1)^2} \leq \frac{6}{t} - \frac{6}{t+1}$ for any integer $t \geq 2$.

$\square$

**Lemma A.4.** *For all $t \geq 1$, $p \in (0, 1)$. It holds that:* $-1 \leq (t+1)^p - t^p - 2p(t+1)^{p-1} \leq 0$.

*Proof.* We assume $t \geq 1$ and $p \in (0, 1)$.

First, consider the function $h(t) = t^p$. We have $\nabla h(t) = pt^{p-1}$ and $\nabla^2 h(t) = p(p-1)t^{p-2} < 0$. Therefore, $h(t)$ is concave. For all $x > 0$ and $y > 0$, we have: $h(x) - h(y) \geq \langle x - y, \nabla h(x) \rangle$. Letting $x = t$ and $y = t + 1$, we have:

$$t^p - (t+1)^p \geq -pt^{p-1} \tag{9}$$

Letting $x = t + 1$ and $y = t$, we have:

$$(t+1)^p - t^p \geq p(t+1)^{p-1} \tag{10}$$

Second, we shown that $t^{p-1} \leq 2(t+1)^{p-1}$. Given $t \geq 1$, we have: $\frac{t+1}{t} \leq 2$, leading to $(\frac{t+1}{t})^{p-1} \geq 2^{p-1} \geq \frac{1}{2}$. We obtain:

$$t^{p-1} \leq 2(t+1)^{p-1}. \tag{11}$$

**Part (a).** We now prove the upper bound. We derive: $(t+1)^p - t^p - 2p(t+1)^{(p-1)} \overset{①}{\leq} pt^{p-1} - 2p(t+1)^{(p-1)} \overset{②}{\leq} 0$, where step ① uses Inequality (9); step ② uses Inequality (11).

**Part (b).** We now focus on the lower bound. We obtain: $(t+1)^p - t^p - 2p(t+1)^{p-1} \overset{①}{\geq} p(t+1)^{p-1} - 2p(t+1)^{p-1} = -p(t+1)^{p-1} \overset{②}{\geq} -p \overset{③}{\geq} -1$, where step ① uses Inequality (10); step ② uses $(t+1)^{p-1} \leq 1$; step ③ uses $p \leq 1$.

$\square$

**Lemma A.5.** *For all $t \geq 1$, $p \in (0, 1)$. It holds that:* $(1-p)T^{1-p} \leq \sum_{t=1}^{T} \frac{1}{t^p} \leq \frac{T^{1-p}}{1-p}$.

*Proof.* We let $t \geq 1$, $p \in (0, 1)$, and $q \in (0, 1)$. We define $h(t) \triangleq \frac{1}{q}(t+1)^q - \frac{1}{q} - qt^q$.

First, we prove that $h(1) = \frac{1}{q}2^q - \frac{1}{q} - q \geq 0$. We let $f(q) = 2^q - 1 - q^2$. We have $\nabla f(q) = \ln(2)2^q - 2q$ and $\nabla^2 f(q) = \ln(2)^2 2^q - 2 \leq \ln(2)^2 2 - 2 < 0$. Therefore, $f(q)$ is concave. The global minimum lies in the boundary point. We have $h(1) \geq \frac{1}{q}\min(f(1), f(0)) = \frac{1}{q}\min(2^0 - 1 - 0^2, 2^1 - 1 - 1^2) = \frac{0}{q} = 0$. Therefore, we have:

$$h(1) = \frac{1}{q}2^q - \frac{1}{q} - q \geq 0. \tag{12}$$

Second, we prove that $h(t) \geq 0$. We have: $\nabla h(t) = (t+1)^{q-1} - qt^{q-1} \geq t^{q-1} - qt^{q-1} = (1-q)t^{q-1} \geq 0$. Therefore, the function $h(t)$ is increasing. We further obtain:

$$h(t) \triangleq \frac{1}{q}(t+1)^q - \frac{1}{q} - qt^q \geq h(1) \overset{①}{\geq} 0, \tag{13}$$

where step ① uses Inequality (12).

Third, we define $h(t) = t^{-p}$. Using the integral test for convergence [2], we have:

$$\int_{1}^{T+1} h(x)dx \leq \sum_{t=1}^{T} h(t) \leq h(1) + \int_{1}^{T} h(x)dx.$$

---

[2] https://en.wikipedia.org/wiki/Integral_test_for_convergence

**Part (a)**. We define $g(x) = \frac{1}{1-p}x^{1-p}$. We derive: $\sum_{t=1}^{T} t^{-p} \leq g(1) + \int_1^T x^{-p}dx \overset{①}{=} 1 + g(T) - g(1) = 1 + \frac{1}{1-p}(T)^{1-p} - \frac{1}{1-p} \overset{②}{=} \frac{T^{(1-p)}-p}{1-p} < \frac{T^{(1-p)}}{1-p}$, where step ① uses $\nabla g(x) = x^{-p}$; step ② uses Inequality (13) with $q = 1 - p$ and $t = T$.

**Part (b)**. We derive: $\sum_{t=1}^{T} t^{-p} \geq \sum_{t=1}^{T} \int_t^{t+1} x^{-p}dx = \int_1^{T+1} x^{-p}dx \overset{①}{\geq} g(T+1) - g(1) = \frac{1}{1-p}(T+1)^{1-p} - \frac{1}{1-p} \overset{②}{\geq} (1-p)T^{1-p}$, where step ① uses $\nabla g(x) = x^{-p}$; step ② uses Inequality (13) with $q = 1 - p$ and $t = T$.

$\square$

## B  ADDITIONAL MOTIVATING APPLICATIONS

• **Robust Sharpe Ratio Maximization (Robust SRM)**. Recall that the standard SRM, which operates without data uncertainty and is commom in finance, can be formulated as: $\max_{\mathbf{x} \in \Omega} \frac{\mathbf{d}^\top \mathbf{x} - r}{\mathbf{x}^\top \mathbf{C}\mathbf{x}}$, where $\mathbf{C} \succeq 0$ is the risk covariance matrix, $\mathbf{d} \in \mathbb{R}^n$ are the expected returns, $r \in \mathbb{R}$ is the risk-free rate, and $\Omega \triangleq \{\mathbf{x} \mid \mathbf{x} \geq \mathbf{0}, \mathbf{x}^\top \mathbf{1} = 1\}$ ensures valid portfolio weights (see (Chen et al., 2011; Boţ et al., 2023b)). In contrast, the robust SRM, designed to handle scenario data uncertainty, is defined by: $\min_{\mathbf{x} \in \Omega} \frac{\max_{i=1}^{m} \{\mathbf{b}_i - (\mathbf{D}\mathbf{x})_i\}}{\max_{i=1}^{p} \mathbf{x}^\top \mathbf{C}_{(i)}\mathbf{x}}$, where each $\mathbf{C}_{(i)} \in \mathbb{R}^{n \times n} \succeq \mathbf{0}$, $\mathbf{b} \in \mathbb{R}^m$, and $\mathbf{D} \in \mathbb{R}^{m \times n}$. The corresponding equivalent optimization problem is formulated as:

$$\min_{\mathbf{x} \in \mathbb{R}^n} \frac{\max(0, \max(\mathbf{b} - \mathbf{D}\mathbf{x}))}{\max_{i=1}^{p} \mathbf{x}^\top \mathbf{C}_{(i)}\mathbf{x}}, \text{s. t. } \mathbf{x} \in \Omega. \tag{14}$$

We use $\max(0, \max(\mathbf{b} - \mathbf{D}\mathbf{x}))$ instead of simply $\max(\mathbf{b} - \mathbf{D}\mathbf{x})$ to explicitly enforce nonnegativity in the numerator for all $\mathbf{x} \in \Omega$. Problem (14) corresponds to Problem (1) with $f(\mathbf{x}) = g(\mathbf{x}) = 0$, $\delta(\mathbf{x}) = \iota_\Omega(\mathbf{x})$, $\mathbf{A} = -\mathbf{D}$, $h(\mathbf{y}) = \max(0, \max(\mathbf{y} + \mathbf{b}))$, and $d(\mathbf{x}) = \max_{i=1}^{p} \mathbf{x}^\top \mathbf{C}_{(i)}\mathbf{x}$. Notably, both $d(\mathbf{x})$ and $d(\mathbf{x})^{1/2}$ are $W_d$-weakly convex with $W_d = 0$.

• **Robust Sparse Recovery**. It is a signal processing technique, which can effectively acquire and reconstruct the signal by finding the solution of the underdetermined linear system. Given a design matrix $\mathbf{A} \in \mathbb{R}^{m \times n}$ and an observation vector $\mathbf{b} \in \mathbb{R}^m$, robust sparse recovery can be formulated as the following fractional optimization problem (Li & Zhang, 2022; Yang & Zhang, 2011; Yuan, 2023):

$$\min_{\mathbf{x} \in \mathbb{R}^n} \frac{\rho_1 \|\mathbf{A}\mathbf{x} - \mathbf{b}\|_1 + \rho_2 \|\mathbf{x}\|_1}{\|\mathbf{x}\|_{[k]}}, \text{s. t. } \mathbf{x} \in \Omega, \tag{15}$$

where $\Omega \triangleq \{\mathbf{x} \mid \|\mathbf{x}\|_\infty \leq \rho_0\}$, and $\rho_0, \rho_1, \rho_2$ are positive constants provided by the users. Problem (15) coincides with Problem (1) with $f(\mathbf{x}) = g(\mathbf{x}) = 0$, $\delta(\mathbf{x}) = \iota_\Omega(\mathbf{x}) + \rho_2 \|\mathbf{x}\|_1$, $h(\mathbf{y}) = \|\mathbf{y} - \mathbf{b}\|_1$, and $d(\mathbf{x}) = \|\mathbf{x}\|_{[k]}$. Importantly, $d(\mathbf{x})$ is $W_d$-weakly convex with $W_d = 0$.

## C  PROOFS FOR SECTION 3

### C.1  PROOF OF LEMMA 3.9

*Proof.* The results of this lemma can be derived using standard convex analysis. Some of these results are well-established and scattered throughout the literature (Nesterov, 2003; Yurtsever et al., 2018; Silveti-Falls et al., 2020; Nesterov, 2013). For completeness, we provide the full proofs of the lemma.

For any $\mathbf{y} \in \mathbb{R}^,$, we define $h_\mu(\mathbf{y}) \triangleq \max_\mathbf{v} \langle \mathbf{y}, \mathbf{v} \rangle - h^*(\mathbf{v}) - \frac{\mu}{2}\|\mathbf{v}\|_2^2$. Since $\mu > 0$ and $\frac{\mu}{2}\|\mathbf{v}\|_2^2$ is $\mu$-strongly convex, the maximization problem has a unique solution and thus the subgradient set is a single set (Nesterov, 2003), i.e., $\partial h_\mu(\mathbf{y}) = \nabla h_\mu(\mathbf{y}) = \arg\max_\mathbf{v} \{\langle \mathbf{y}, \mathbf{v} \rangle - h^*(\mathbf{v}) - \frac{\mu}{2}\|\mathbf{v}\|_2^2\}$.

**Part (a)**. We now prove that $h_\mu(\mathbf{y})$ is $(1/\mu)$-smooth. For any $\mathbf{y}_1, \mathbf{y}_2 \in \mathbb{R}^m$, we define $\mathbf{v}_1 = \arg\max_{\mathbf{v}}\{\langle \mathbf{y}_1, \mathbf{v}\rangle - h^*(\mathbf{v}) - \frac{\mu}{2}\|\mathbf{v}\|_2^2\}$, and $\mathbf{v}_2 = \arg\max_{\mathbf{v}}\{\langle \mathbf{y}_2, \mathbf{v}\rangle - h^*(\mathbf{v}) - \frac{\mu}{2}\|\mathbf{v}\|_2^2\}$. We have:

$$
\begin{aligned}
\|\mathbf{y}_1 - \mathbf{y}_2\|\|\mathbf{v}_1 - \mathbf{v}_2\| &\overset{①}{\geq} \langle \mathbf{y}_1 - \mathbf{y}_2, \mathbf{v}_1 - \mathbf{v}_2\rangle \\
&\overset{②}{=} \mu\langle \mathbf{v}_1 - \mathbf{v}_2, \mathbf{v}_1 - \mathbf{v}_2\rangle + \langle \partial h^*(\mathbf{v}_1) - \partial h^*(\mathbf{v}_2), \mathbf{v}_1 - \mathbf{v}_2\rangle \\
&\overset{③}{\geq} \mu\|\mathbf{v}_2 - \mathbf{v}_1\|_2^2 + 0,
\end{aligned}
$$

where step ① uses the Cauchy-Schwarz Inequality; step ② uses the optimality of $\mathbf{v}_1$ that $\mathbf{y}_1 - \partial h^*(\mathbf{v}_1) - \mu\mathbf{v}_1 = \mathbf{0}$ and the optimality of $\mathbf{v}_2$ that $\mathbf{y}_2 - \partial h^*(\mathbf{v}_2) - \mu\mathbf{v}_2 = \mathbf{0}$; step ③ uses the monotonicity of subdifferentials for the convex function $h^*(\mathbf{v})$. Dividing both sides by $\|\mathbf{v}_1 - \mathbf{v}_2\|$, we have: $\frac{\|\mathbf{v}_2 - \mathbf{v}_1\|}{\|\mathbf{y}_1 - \mathbf{y}_2\|} \leq \frac{1}{\mu}$, which implies that the function $h_\mu(\mathbf{y})$ is $(1/\mu)$-smooth.

We now prove that the function $h_\mu(\mathbf{y})$ is convex. For any $\mathbf{y}_1, \mathbf{y}_2 \in \mathbb{R}^m$ and $\mu > 0$, we define $\mathbf{u}_1 = \arg\max_{\mathbf{u}}\{\langle \mathbf{y}_1, \mathbf{u}\rangle - h^*(\mathbf{u}) - \frac{\mu}{2}\|\mathbf{u}\|_2^2\}$, and $\mathbf{u}_2 = \arg\max_{\mathbf{u}}\{\langle \mathbf{y}_2, \mathbf{u}\rangle - h^*(\mathbf{u}) - \frac{\mu}{2}\|\mathbf{u}\|_2^2\}$. We have:

$$
\begin{aligned}
h_\mu(\mathbf{y}_1) - h_\mu(\mathbf{y}_2) &\overset{①}{=} h_\mu(\mathbf{y}_1) - \{\langle \mathbf{y}_2, \mathbf{u}_2\rangle - h^*(\mathbf{u}_2) - \frac{\mu}{2}\|\mathbf{u}_2\|_2^2\} \\
&\overset{②}{\leq} h_\mu(\mathbf{y}_1) - \{\langle \mathbf{y}_2, \mathbf{u}_1\rangle - h^*(\mathbf{u}_1) - \frac{\mu}{2}\|\mathbf{u}_1\|_2^2\} \\
&\overset{③}{=} \{\langle \mathbf{y}_1, \mathbf{u}_1\rangle - h^*(\mathbf{u}_1) - \frac{\mu}{2}\|\mathbf{u}_1\|_2^2\} - \{\langle \mathbf{y}_2, \mathbf{u}_1\rangle - h^*(\mathbf{u}_1) - \frac{\mu}{2}\|\mathbf{u}_1\|_2^2\} \\
&= \langle \mathbf{u}_1, \mathbf{y}_1 - \mathbf{y}_2\rangle,
\end{aligned}
$$

where step ① uses the definition of $h_\mu(\mathbf{y}_2)$ and $\mathbf{u}_2$; step ② uses the optimality of $\mathbf{u}_2$; step ③ uses the definition of $h_\mu(\mathbf{y}_1)$.

**Part (b)**. For any $\mathbf{y} \in \mathbb{R}^m$ and $\mu > 0$, we define $h_\mu(\mathbf{y}) \triangleq \max_{\mathbf{v}} \langle \mathbf{y}, \mathbf{v}\rangle - h^*(\mathbf{v}) - \frac{\mu}{2}\|\mathbf{v}\|_2^2$, $\mathbf{u}_1 \triangleq \arg\max_{\mathbf{u}}\{\langle \mathbf{y}, \mathbf{u}\rangle - h^*(\mathbf{u})\}$, and $\mathbf{u}_2 \triangleq \arg\max_{\mathbf{u}}\{\langle \mathbf{y}, \mathbf{u}\rangle - h^*(\mathbf{u}) - \frac{\mu}{2}\|\mathbf{u}\|_2^2\}$.

**b-i)**. We now prove that $0 < h(\mathbf{y}) - h_\mu(\mathbf{y})$. We have:

$$
\begin{aligned}
h_\mu(\mathbf{y}) &= \max_{\mathbf{v}}\{\mathbf{v}^\mathsf{T}\mathbf{y} - h^*(\mathbf{v}) - \frac{\mu}{2}\|\mathbf{v}\|_2^2\} \\
&\overset{①}{\leq} \max_{\mathbf{v}}\{\mathbf{v}^\mathsf{T}\mathbf{y} - h^*(\mathbf{v})\} + \max_{\mathbf{v}}\{-\frac{\mu}{2}\|\mathbf{v}\|_2^2\} \\
&\overset{②}{=} h(\mathbf{y}),
\end{aligned}
$$

where step ① uses a general property of the maximum function when applied to the sum of two functions; step ② uses the definition of $h(\mathbf{y}) = \max_{\mathbf{v}}\{\mathbf{v}^\mathsf{T}\mathbf{y} - h^*(\mathbf{v})\}$ and the fact that $\max_{\mathbf{v}}\{-\frac{\mu}{2}\|\mathbf{v}\|_2^2\} = 0$.

**b-ii)**. We now prove that $h(\mathbf{y}) - h_\mu(\mathbf{y}) \leq \frac{\mu}{2}C_h^2$. We have:

$$
\begin{aligned}
h(\mathbf{y}) - h_\mu(\mathbf{y}) &\overset{①}{=} \{\langle \mathbf{y}, \mathbf{u}_1\rangle - h^*(\mathbf{u}_1)\} - h_\mu(\mathbf{y}) \\
&\overset{②}{\leq} \{\langle \mathbf{y}, \mathbf{u}_1\rangle - h^*(\mathbf{u}_1)\} - \{\langle \mathbf{y}, \mathbf{u}_1\rangle - h^*(\mathbf{u}_1) - \frac{\mu}{2}\|\mathbf{u}_1\|_2^2\} \\
&= \frac{\mu}{2}\|\mathbf{u}_2\|_2^2 = \frac{\mu}{2}\|\nabla h_\mu(\mathbf{y})\|_2^2 \\
&\overset{③}{\leq} \frac{\mu}{2}C_h^2,
\end{aligned}
$$

where step ① uses the definition of $h(\mathbf{y})$: $h(\mathbf{y}) = \{\langle \mathbf{y}, \mathbf{u}_1\rangle - h^*(\mathbf{u}_1)\}$; step ② uses the definition of $h_\mu(\mathbf{y})$ and the optimality of $\mathbf{u}_2$: $h_\mu(\mathbf{y}) = \{\langle \mathbf{y}, \mathbf{u}_2\rangle - h^*(\mathbf{u}_2) - \frac{\mu}{2}\|\mathbf{u}_2\|_2^2\} \geq \{\langle \mathbf{y}, \mathbf{u}_1\rangle - h^*(\mathbf{u}_1) - \frac{\mu}{2}\|\mathbf{u}_1\|_2^2\}$; step ③ uses Claim (*b*) of this lemma.

**Part (c)**. We now prove that the function $h_\mu(\mathbf{y})$ is $C_h$-Lipschitz continuous. For any $\mathbf{y} \in \mathbb{R}^m$ and $\mu > 0$, we define $h_\mu(\mathbf{y}) \triangleq \max_\mathbf{v} \langle \mathbf{y}, \mathbf{v} \rangle - h^*(\mathbf{v}) - \frac{\mu}{2}\|\mathbf{v}\|_2^2$. We have:

$$
\begin{aligned}
\nabla h_\mu(\mathbf{y}) &\overset{\text{①}}{=} \arg\max_\mathbf{v}\{\langle \mathbf{y}, \mathbf{v}\rangle - h^*(\mathbf{v}) - \tfrac{\mu}{2}\|\mathbf{v}\|_2^2\} \\
&= \arg\min_\mathbf{v}\{h^*(\mathbf{v}) + \tfrac{\mu}{2}\|\mathbf{v} - \mathbf{y}/\mu\|_2^2\} \\
&\overset{\text{②}}{=} \operatorname{Prox}(\tfrac{\mathbf{y}}{\mu}; h^*, 1/\mu) \\
&\overset{\text{③}}{=} \tfrac{1}{\mu}(\mathbf{y} - \operatorname{Prox}(\mathbf{y}; h, \mu)) && (16) \\
&\overset{\text{④}}{\in} \partial h(\operatorname{Prox}(\mathbf{y}; h, \mu)), && (17)
\end{aligned}
$$

where step ① uses the fact that the function $h_\mu(\mathbf{y})$ is smooth and its gradient can be computed as: $\nabla h_\mu(\mathbf{y}) = \arg\max_\mathbf{v}\{\langle \mathbf{y}, \mathbf{v}\rangle - h^*(\mathbf{v}) - \frac{\mu}{2}\|\mathbf{v}\|_2^2\}$; step ② uses the definition of $\operatorname{Prox}(\mathbf{b}; h, \mu) \triangleq \arg\min_\mathbf{y} h(\mathbf{y}) + \frac{1}{2\mu}\|\mathbf{b} - \mathbf{y}\|_2^2$; step ③ uses the extended Moreau decomposition property as shown in Lemma A.2; step ④ uses the optimality of $\operatorname{Prox}(\mathbf{y}; h, \mu)$ that $\mathbf{0} \in \partial h(\operatorname{Prox}(\mathbf{y}; h, \mu)) + \frac{1}{\mu}(\operatorname{Prox}(\mathbf{y}; h, \mu) - \mathbf{y})$. Using Equation (17), we directly conclude that $\nabla h_\mu(\mathbf{y})$ is $C_h$-Lipschitz continuous with $\|\nabla h_\mu(\mathbf{y})\| \leq C_h$.

**Part (d)**. We show how to compute $h_\mu(\mathbf{y})$. For any $\mathbf{y} \in \mathbb{R}^m$ and $\mu > 0$, we define $h_\mu(\mathbf{y}) \triangleq \max_\mathbf{v} \langle \mathbf{y}, \mathbf{v}\rangle - h^*(\mathbf{v}) - \frac{\mu}{2}\|\mathbf{v}\|_2^2$, and $\bar{\mathbf{v}} = \arg\max_\mathbf{v}\{\langle \mathbf{y}, \mathbf{v}\rangle - h^*(\mathbf{v}) - \frac{\mu}{2}\|\mathbf{v}\|_2^2\}$. We have:

$$
\mathbf{y} - \mu\bar{\mathbf{v}} \in \partial h^*(\bar{\mathbf{v}}) \quad \overset{\text{①}}{\Leftrightarrow} \quad \langle \mathbf{y} - \mu\bar{\mathbf{v}}, \bar{\mathbf{v}}\rangle = h^*(\bar{\mathbf{v}}) + h(\mathbf{y} - \mu\bar{\mathbf{v}}). \tag{18}
$$

where step ① uses the equivalence relation: $\langle \tilde{\mathbf{x}}, \tilde{\mathbf{y}}\rangle = h(\tilde{\mathbf{x}}) + h^*(\tilde{\mathbf{y}}) \Leftrightarrow \tilde{\mathbf{y}} \in \partial h(\tilde{\mathbf{x}}) \Leftrightarrow \tilde{\mathbf{x}} \in \partial h^*(\tilde{\mathbf{y}})$ for all $\tilde{\mathbf{x}}$ and $\tilde{\mathbf{y}}$, as stated in Proposition 11.3 of (Rockafellar & Wets., 2009). Therefore, we have:

$$
\begin{aligned}
h_\mu(\mathbf{y}) &\overset{\text{①}}{=} \langle \mathbf{y}, \bar{\mathbf{v}}\rangle - h^*(\bar{\mathbf{v}}) - \tfrac{\mu}{2}\|\bar{\mathbf{v}}\|_2^2 \\
&\overset{\text{②}}{=} \langle \mathbf{y}, \bar{\mathbf{v}}\rangle - \langle \mathbf{y} - \mu\bar{\mathbf{v}}, \bar{\mathbf{v}}\rangle - h(\mathbf{y} - \mu\bar{\mathbf{v}}) - \tfrac{\mu}{2}\|\bar{\mathbf{v}}\|_2^2 \\
&\overset{\text{③}}{=} \langle \mathbf{y}, \bar{\mathbf{v}}\rangle - \langle \mathbf{y} - \mu\bar{\mathbf{v}}, \bar{\mathbf{v}}\rangle + h(\operatorname{Prox}(\mathbf{y}; h, \mu)) - \tfrac{\mu}{2}\|\bar{\mathbf{v}}\|_2^2 \\
&= h(\operatorname{Prox}(\mathbf{y}; h, \mu)) + \tfrac{\mu}{2}\|\tfrac{1}{\mu}(\mathbf{y} - \operatorname{Prox}(\mathbf{y}; h, \mu))\|_2^2,
\end{aligned}
$$

where step ① uses the definition of $h_\mu(\mathbf{y})$; step ② uses Equation (18) that $h^*(\bar{\mathbf{v}}) = \langle \mathbf{y} - \mu\bar{\mathbf{v}}, \bar{\mathbf{v}}\rangle - h(\mathbf{y} - \mu\bar{\mathbf{v}})$; step ③ uses $\bar{\mathbf{v}} = \frac{1}{\mu}(\mathbf{y} - \operatorname{Prox}(\mathbf{y}; h, \mu))$, as shown in Equation (16).

**Part (e)**. For any $\mathbf{y} \in \mathbb{R}^m$ and $\mu_1, \mu_2 > 0$ with $\mu_2 \leq \mu_1$, we define $\mathbf{u}_1 = \arg\max_\mathbf{u}\{\langle \mathbf{y}, \mathbf{u}\rangle - h^*(\mathbf{u}) - \frac{\mu_1}{2}\|\mathbf{u}\|_2^2\}$, and $\mathbf{u}_2 = \arg\max_\mathbf{u}\{\langle \mathbf{y}, \mathbf{u}\rangle - h^*(\mathbf{u}) - \frac{\mu_2}{2}\|\mathbf{u}\|_2^2\}$.

**e-i)**. We now prove that $0 \leq h_{\mu_2}(\mathbf{y}) - h_{\mu_1}(\mathbf{y})$ for all $0 < \mu_2 \leq \mu_1$. We have:

$$
\begin{aligned}
&h_{\mu_1}(\mathbf{y}) - h_{\mu_2}(\mathbf{y}) \\
&\overset{\text{①}}{=} \{\langle \mathbf{y}, \mathbf{u}_1\rangle - h^*(\mathbf{u}_1) - \tfrac{\mu_1}{2}\|\mathbf{u}_1\|_2^2\} - h_{\mu_2}(\mathbf{y}) \\
&\overset{\text{②}}{\leq} \{\langle \mathbf{y}, \mathbf{u}_1\rangle - h^*(\mathbf{u}_1) - \tfrac{\mu_1}{2}\|\mathbf{u}_1\|_2^2\} - \{\langle \mathbf{y}, \mathbf{u}_1\rangle - h^*(\mathbf{u}_1) - \tfrac{\mu_2}{2}\|\mathbf{u}_1\|_2^2\} \\
&= \tfrac{\mu_2 - \mu_1}{2}\|\mathbf{u}_1\|_2^2 \\
&\overset{\text{③}}{\leq} 0,
\end{aligned}
$$

where step ① uses the definition of $h_{\mu_1}(\mathbf{y})$: $h_{\mu_1}(\mathbf{y}) = \{\langle \mathbf{y}, \mathbf{u}_1\rangle - h^*(\mathbf{u}_1) - \frac{\mu_1}{2}\|\mathbf{u}_1\|_2^2\}$; step ② uses the definition of $h_{\mu_2}(\mathbf{y})$ and the optimality of $\mathbf{u}_2$: $h_{\mu_2}(\mathbf{y}) = \{\langle \mathbf{y}, \mathbf{u}_2\rangle - h^*(\mathbf{u}_2) - \frac{\mu_2}{2}\|\mathbf{u}_2\|_2^2\} \geq \{\langle \mathbf{y}, \mathbf{u}_1\rangle - h^*(\mathbf{u}_1) - \frac{\mu_2}{2}\|\mathbf{u}_1\|_2^2\}$; step ③ uses $\mu_2 \leq \mu_1$.

**e-ii).** We now prove that $h_{\mu_2}(\mathbf{y}) - h_{\mu_1}(\mathbf{y}) \leq \frac{\mu_1 - \mu_2}{2} C_h^2$ for all $0 < \mu_2 \leq \mu_1$. We have:

$$h_{\mu_2}(\mathbf{y}) - h_{\mu_1}(\mathbf{y})$$

$$\overset{①}{=} \{\langle \mathbf{y}, \mathbf{u}_2 \rangle - h^*(\mathbf{u}_2) - \tfrac{\mu_2}{2} \|\mathbf{u}_2\|_2^2\} - h_{\mu_1}(\mathbf{y})$$

$$\overset{②}{\leq} \{\langle \mathbf{y}, \mathbf{u}_2 \rangle - h^*(\mathbf{u}_2) - \tfrac{\mu_2}{2} \|\mathbf{u}_2\|_2^2\} - \{\langle \mathbf{y}, \mathbf{u}_2 \rangle - h^*(\mathbf{u}_2) - \tfrac{\mu_1}{2} \|\mathbf{u}_2\|_2^2\}$$

$$= \tfrac{\mu_1 - \mu_2}{2} \|\mathbf{u}_2\|_2^2 = \tfrac{\mu_1 - \mu_2}{2} \|\nabla h_{\mu_2}(\mathbf{y})\|_2^2$$

$$\overset{③}{\leq} \tfrac{\mu_1 - \mu_2}{2} C_h^2,$$

where step ① uses the definition of $h_{\mu_2}(\mathbf{y})$: $h_{\mu_2}(\mathbf{y}) = \{\langle \mathbf{y}, \mathbf{u}_2 \rangle - h^*(\mathbf{u}_2) - \tfrac{\mu_2}{2} \|\mathbf{u}_2\|_2^2\}$; step ② uses the definition of $h_{\mu_1}(\mathbf{y})$ and the optimality of $\mathbf{u}_1$: $h_{\mu_1}(\mathbf{y}) = \{\langle \mathbf{y}, \mathbf{u}_1 \rangle - h^*(\mathbf{u}_1) - \tfrac{\mu_1}{2} \|\mathbf{u}_1\|_2^2\} \geq \{\langle \mathbf{y}, \mathbf{u}_2 \rangle - h^*(\mathbf{u}_2) - \tfrac{\mu_1}{2} \|\mathbf{u}_2\|_2^2\}$; step ③ uses Claim (**b**) of this lemma.

**Part (f).** We now prove that $\|\nabla h_{\mu_2}(\mathbf{y}) - \nabla h_{\mu_1}(\mathbf{y})\| \leq (\frac{\mu_1}{\mu_2} - 1) C_h$ for all $0 < \mu_2 \leq \mu_1$. Using Equality (16), we have:

$$\nabla h_\mu(\mathbf{y}) = \tfrac{1}{\mu}(\mathbf{y} - \mathrm{Prox}(\mathbf{y}; h, \mu)).$$

We now examine the following mapping $\mathcal{H}(\upsilon) \triangleq \upsilon(\mathbf{y} - \mathrm{Prox}(\mathbf{y}; h, 1/\upsilon))$. We derive:

$$\lim_{\delta \to 0} \tfrac{\mathcal{H}(\upsilon + \delta) - \mathcal{H}(\upsilon)}{\delta} = \lim_{\delta \to 0} \tfrac{(\upsilon + \delta)(\mathbf{y} - \mathrm{Prox}(\mathbf{y}; h, 1/(\upsilon + \delta))) - \upsilon(\mathbf{y} - \mathrm{Prox}(\mathbf{y}; h, 1/\upsilon))}{\delta}$$

$$= \lim_{\delta \to 0} \tfrac{\delta \mathbf{y} - (\upsilon + \delta)\mathrm{Prox}(\mathbf{y}; h, 1/\upsilon) + \upsilon \mathrm{Prox}(\mathbf{y}; h, 1/\upsilon)}{\delta}$$

$$= \mathbf{y} - \mathrm{Prox}(\mathbf{y}; h, 1/\upsilon).$$

Therefore, the first-order derivative of the mapping $\mathcal{H}(\upsilon)$ *w.r.t.* $\upsilon$ always exists and can be computed as: $\nabla_\upsilon \mathcal{H}(\upsilon) = \mathbf{y} - \mathrm{Prox}(\mathbf{y}; h, 1/\upsilon)$, resulting in:

$$\forall \upsilon, \upsilon' > 0, \tfrac{\|\mathcal{H}(\upsilon) - \mathcal{H}(\upsilon')\|}{|\upsilon - \upsilon'|} \leq \|\mathbf{y} - \mathrm{Prox}(\mathbf{y}; h, 1/\upsilon)\|.$$

Letting $\upsilon = 1/\mu_1$ and $\upsilon' = 1/\mu_2$, we derive:

$$\begin{aligned}
\tfrac{\|\nabla h_{\mu_1}(\mathbf{y}) - \nabla h_{\mu_2}(\mathbf{y})\|}{|1/\mu_1 - 1/\mu_2|} &\leq \|\mathbf{y} - \mathrm{Prox}(\mathbf{y}; h, \mu_1)\| \\
&\overset{①}{\leq} \mu_1 \|\partial h(\mathrm{Prox}(\mathbf{y}; h, \mu_1))\| \\
&\overset{②}{\leq} \mu_1 C_h,
\end{aligned}$$

where step ① uses the optimality of $\mathrm{Prox}(\mathbf{y}; h, \mu_1)$ that $\mathbf{0} \in \partial h(\mathrm{Prox}(\mathbf{y}; h, \mu_1)) + \tfrac{1}{\mu_1}(\mathrm{Prox}(\mathbf{y}; h, \mu_1) - \mathbf{y})$ for all $\mu_1$; step ② uses the Lipschitz continuity of $h(\cdot)$. We further obtain:

$$\|\nabla h_{\mu_1}(\mathbf{x}) - \nabla h_{\mu_2}(\mathbf{x})\| \leq |\tfrac{1}{\mu_1} - \tfrac{1}{\mu_2}| \cdot \mu_1 C_h = (\tfrac{\mu_1}{\mu_2} - 1) C_h.$$

$\square$

## C.2 Proof of Lemma 3.10

*Proof.* Consider the strongly convex minimization problem: $\bar{\mathbf{y}} = \arg\min_{\mathbf{y}} h_\mu(\mathbf{y}) + \tfrac{1}{2}\beta \|\mathbf{y} - \mathbf{b}\|_2^2$, which can be equivalently repressed as:

$$(\bar{\mathbf{y}}, \bar{\mathbf{v}}) = \arg\min_{\mathbf{y}} \max_{\mathbf{v}} \{\mathbf{y}^\mathsf{T}\mathbf{v} - h^*(\mathbf{v}) - \tfrac{\mu}{2}\|\mathbf{v}\|_2^2 + \tfrac{\beta}{2}\|\mathbf{y} - \mathbf{b}\|_2^2\}.$$

Using the optimality of the variables $\{\bar{\mathbf{y}}, \bar{\mathbf{v}}\}$, we have:

$$\bar{\mathbf{y}} = \mathbf{b} - \tfrac{1}{\beta}\bar{\mathbf{v}}, \tag{19}$$

$$\mathbf{0} = -\partial h^*(\bar{\mathbf{v}}) - \mu\bar{\mathbf{v}} + \bar{\mathbf{y}}. \tag{20}$$

Plugging Equation (19) into Equation (20) to eliminate $\bar{\mathbf{y}}$ yields:

$$\mathbf{0} = -\partial h^*(\bar{\mathbf{v}}) - \mu\bar{\mathbf{v}} + \mathbf{b} - \tfrac{1}{\beta}\bar{\mathbf{v}}. \tag{21}$$

Second, we derive the following equalities:

$$\bar{\mathbf{v}} \overset{①}{=} \arg\max_{\mathbf{v}} -h^*(\mathbf{v}) + \langle \mathbf{v}, \mathbf{b}\rangle - \tfrac{\mu}{2}\|\mathbf{v}\|_2^2 - \tfrac{1}{2\beta}\|\mathbf{v}\|_2^2 \tag{22}$$

$$= \arg\min_{\mathbf{v}} h^*(\mathbf{v}) - \langle \mathbf{v}, \mathbf{b}\rangle + \tfrac{1}{2}(\mu + \tfrac{1}{\beta})\|\mathbf{v}\|_2^2$$

$$= \arg\min_{\mathbf{v}} h^*(\mathbf{v}) + \tfrac{1}{2}(\mu + \tfrac{1}{\beta})\|\mathbf{v} - \mathbf{b}/(\mu + \tfrac{1}{\beta})\|_2^2$$

$$\overset{②}{=} \mathrm{Prox}(\tfrac{\mathbf{b}}{\mu+1/\beta}); h^*, \tfrac{1}{\mu+1/\beta})$$

$$\overset{③}{=} \tfrac{1}{\mu+1/\beta} \cdot (\mathbf{b} - \mathrm{Prox}(\mathbf{b}; h, \mu + \tfrac{1}{\beta})) \tag{23}$$

$$\overset{④}{\in} \partial h(\mathrm{Prox}(\mathbf{b}; h, \mu + \tfrac{1}{\beta})), \tag{24}$$

where step ① uses the fact that Equation (21) is the necessary and sufficient first-order optimality condition for Problem (22); step ② uses the definition of $\mathrm{Prox}(\cdot; \cdot, \cdot)$; step ③ uses the extended Moreau decomposition that $\mathbf{a} = \mathrm{Prox}(\mathbf{a}; h, \mu) + \mu\,\mathrm{Prox}(\tfrac{\mathbf{a}}{\mu}; h^*, 1/\mu)$ for all $\mu > 0$ and $\mathbf{a}$, as shown in Lemma A.2; step ④ uses the following necessary and sufficient first-order optimality condition for $\mathrm{Prox}(\mathbf{b}; h, \mu + \tfrac{1}{\beta})$:

$$\tfrac{1}{\mu+1/\beta} \cdot \{\mathbf{b} - \mathrm{Prox}(\mathbf{b}; h, \mu + \tfrac{1}{\beta})\} \in \partial h(\mathrm{Prox}(\mathbf{b}; h, \mu + \tfrac{1}{\beta})).$$

**Part (a).** Combining Equation (23) with Equation (19) to eliminate $\bar{\mathbf{v}}$, we have:

$$\bar{\mathbf{y}} = \mathbf{b} - \tfrac{1}{\beta} \cdot \tfrac{1}{\mu+1/\beta} \cdot \{\mathbf{b} - \mathrm{Prox}(\mathbf{b}; h, \mu + 1/\beta)\}$$

$$= \mathbf{b} - \tfrac{1}{\mu\beta+1} \cdot \{\mathbf{b} - \mathrm{Prox}(\mathbf{b}; h, \mu + 1/\beta)\}.$$

**Part (b).** We define $\check{\mathbf{y}} \triangleq \mathrm{Prox}(\mathbf{b}; h, \mu + 1/\beta)$. We have:

$$\beta(\mathbf{b} - \bar{\mathbf{y}}) \overset{①}{=} \tfrac{\beta}{\mu\beta+1} \cdot \{\mathbf{b} - \mathrm{Prox}(\mathbf{b}; h, \mu + 1/\beta)\}$$

$$\overset{②}{=} \tfrac{1}{\mu+1/\beta} \cdot \{\mathbf{b} - \check{\mathbf{y}}\} \overset{③}{=} \bar{\mathbf{v}} \overset{④}{\in} \partial h(\check{\mathbf{y}}), \tag{25}$$

where step ① uses Claim (*a*) of this lemma; step ② uses the definition of $\check{\mathbf{y}}$; step ③ uses Equality (23); step ④ uses Equality (24) that $\bar{\mathbf{v}} \in \partial h(\check{\mathbf{y}})$.

**Part (c).** We now prove that $\|\bar{\mathbf{y}} - \check{\mathbf{y}}\| \leq \mu C_h$. We derive:

$$\|\bar{\mathbf{y}} - \check{\mathbf{y}}\| \overset{①}{=} \|\mathbf{b} - \tfrac{1}{\beta\mu+1} \cdot (\mathbf{b} - \check{\mathbf{y}}) - \check{\mathbf{y}}\|$$

$$= \tfrac{\beta\mu}{1+\beta\mu}\|\check{\mathbf{y}} - \mathbf{b}\|$$

$$\overset{②}{=} \tfrac{\beta\mu}{1+\beta\mu}(\mu + 1/\beta)\|\partial h(\check{\mathbf{y}})\|$$

$$\overset{③}{\leq} \tfrac{\beta\mu}{1+\beta\mu}(\mu + 1/\beta)C_h = \mu C_h,$$

where step ① uses Claim (*a*) of this lemma that $\bar{\mathbf{y}} = \mathbf{b} - \tfrac{1}{\beta\mu+1} \cdot \{\mathbf{b} - \check{\mathbf{y}}\}$; step ② uses Equality (25) that $\mathbf{b} - \check{\mathbf{y}} \in (\mu + 1/\beta)\partial h(\check{\mathbf{y}})$; step ③ uses the fact that $h(\mathbf{y})$ is $C_h$-Lipschitz continuous.

$\square$

# D  PROOFS FOR SECTION 4

## D.1  PROOF OF LEMMA 5.3

*Proof.* **Part (a)**. We now show that $\mathbf{z}^{t+1} = \nabla h_{\mu^t}(\mathbf{y}^{t+1})$. For any $t \geq 0$, we have:

$$\mathbf{0} \overset{①}{=} \nabla h_{\mu^t}(\mathbf{y}^{t+1}) + \beta^t(\mathbf{y}^{t+1} - \mathbf{y}^t) + \nabla_{\mathbf{y}} \mathcal{S}(\mathbf{x}^{t+1}, \mathbf{y}^t; \mathbf{z}^t; \beta^t)$$
$$\overset{②}{=} \nabla h_{\mu^t}(\mathbf{y}^{t+1}) + \beta^t(\mathbf{y}^{t+1} - \mathbf{y}^t) + \beta^t(\mathbf{y}^t - \mathbf{A}\mathbf{x}^{t+1}) - \mathbf{z}^t$$
$$\overset{③}{=} \nabla h_{\mu^t}(\mathbf{y}^{t+1}) - \mathbf{z}^{t+1},$$

where step ① uses the optimality condition for $\mathbf{y}^{t+1}$; step ② uses $\nabla_{\mathbf{y}} \mathcal{S}(\mathbf{x}^{t+1}, \mathbf{y}^t; \mathbf{z}^t; \beta^t) = \beta^t(\mathbf{y} - \mathbf{A}\mathbf{x}^{t+1}) - \mathbf{z}^t$; step ③ uses $\mathbf{z}^{t+1} - \mathbf{z}^t = \beta^t(\mathbf{A}\mathbf{x}^{t+1} - \mathbf{y}^{t+1})$.

**Part (b)**. We now show that $\nabla h_{\mu^t}(\mathbf{y}^{t+1}) \in \partial h(\breve{\mathbf{y}}^{t+1})$. For any $t \geq 0$, we have:

$$\partial h(\breve{\mathbf{y}}^{t+1}) \overset{①}{\ni} \beta^t(\mathbf{b}^t - \mathbf{y}^{t+1})$$
$$\overset{②}{=} \beta^t(\mathbf{A}\mathbf{x}^{t+1} + \mathbf{z}^t/\beta^t - \mathbf{y}^{t+1})$$
$$\overset{③}{=} \mathbf{z}^{t+1},$$

where step ① uses Lemma 3.10(b); step ② uses $\mathbf{b}^t = \mathbf{y}^t - \nabla_{\mathbf{y}} \mathcal{S}(\mathbf{x}^{t+1}, \mathbf{y}^t; z^t; \beta^t)/\beta^t = \mathbf{y}^t - [\beta^t(\mathbf{y}^t - \mathbf{A}\mathbf{x}^{t+1}) - \mathbf{z}^t]/\beta^t = \mathbf{A}\mathbf{x}^{t+1} + \mathbf{z}^t/\beta^t$; step ③ uses $\mathbf{z}^{t+1} - \mathbf{z}^t = \beta^t(\mathbf{A}\mathbf{x}^{t+1} - \mathbf{y}^{t+1})$. $\qquad \square$

## D.2  PROOF OF LEMMA 5.4

*Proof.* Since $t$ can be arbitrary, for any $t \geq 1$, we have from Lemma 5.3:

$$\mathbf{0} = \nabla h_{\mu^t}(\mathbf{y}^{t+1}) - \mathbf{z}^{t+1},$$
$$\mathbf{0} = \nabla h_{\mu^{t-1}}(\mathbf{y}^t) - \mathbf{z}^t.$$

Combining the two equalities above, we have, for any $t \geq 1$:

$$\mathbf{z}^{t+1} - \mathbf{z}^t = \nabla h_{\mu^t}(\mathbf{y}^{t+1}) - \nabla h_{\mu^{t-1}}(\mathbf{y}^t).$$

This further leads to the following inequalities:

$$\|\mathbf{z}^{t+1} - \mathbf{z}^t\|_2^2 = \|\nabla h_{\mu^t}(\mathbf{y}^{t+1}) - \nabla h_{\mu^{t-1}}(\mathbf{y}^t)\|_2^2$$
$$= \|\nabla h_{\mu^t}(\mathbf{y}^{t+1}) - \nabla h_{\mu^t}(\mathbf{y}^t) + \nabla h_{\mu^t}(\mathbf{y}^t) - \nabla h_{\mu^{t-1}}(\mathbf{y}^t)\|_2^2$$
$$\overset{①}{\leq} 2\|\nabla h_{\mu^t}(\mathbf{y}^{t+1}) - \nabla h_{\mu^t}(\mathbf{y}^t)\|_2^2 + 2\|\nabla h_{\mu^t}(\mathbf{y}^t) - \nabla h_{\mu^{t-1}}(\mathbf{y}^t)\|_2^2$$
$$\overset{②}{\leq} 2\|\tfrac{1}{\mu^t}(\mathbf{y}^{t+1} - \mathbf{y}^t)\|_2^2 + 2(\tfrac{\mu^{t-1}}{\mu^t} - 1)^2 C_h^2$$
$$\overset{③}{\leq} 2\tfrac{(\beta^t)^2}{\chi^2}\|\mathbf{y}^{t+1} - \mathbf{y}^t\|_2^2 + 2C_h^2(\tfrac{6}{t} - \tfrac{6}{t+1}),$$

where step ① uses $\|\mathbf{a} + \mathbf{b}\|_2^2 \leq 2\|\mathbf{a}\|_2^2 + 2\|\mathbf{a}\|_2^2$; step ② uses $\tfrac{1}{\mu^t}$-smoothness of $h_{\mu^t}(\mathbf{y})$ for all $\mathbf{y}$, and Lemma 3.9(f) that $\|\nabla h_{\mu_1}(\mathbf{y}) - \nabla h_{\mu_2}(\mathbf{y})\| \leq (\tfrac{\mu_1}{\mu_2} - 1)C_h$ for all $0 < \mu_2 < \mu_1$; step ③ uses $\mu^t = \tfrac{\chi}{\beta^t}$ and Lemma A.3 that $(\tfrac{\mu^{t-1}}{\mu^t} - 1)^2 \leq \tfrac{6}{t} - \tfrac{6}{t+1}$ for any integer $t \geq 1$. $\qquad \square$

## D.3  PROOF OF LEMMA 5.5

*Proof.* We define $\bar{z} \triangleq \max(\|\mathbf{z}^0\|, C_h)$, and $\bar{y} \triangleq \max(\|\mathbf{y}^0\|, \tfrac{2}{\beta^0}\bar{z} + \|\mathbf{A}\|\bar{x})$.

**Part (a)**. The conclusion $\|\mathbf{x}^t\| \leq \bar{x}$ directly follows by assumption.

**Part (b)**. We now show that $\|\mathbf{z}^t\| \leq \bar{\mathbf{z}}$. Using Lemma 5.3(a), we have $\forall t \geq 0$, $\mathbf{z}^{t+1} = \nabla h_{\mu^t}(\mathbf{y}^{t+1})$. This leads to $\forall t \geq 1$, $\|\mathbf{z}^t\| \leq \|\nabla h_{\mu^{t-1}}(\mathbf{y}^t)\| \leq C_h$. Therefore, it holds that $\forall t \geq 0$, $\|\mathbf{z}^t\| \leq \max(\|\mathbf{z}^0\|, C_h) \triangleq \bar{\mathbf{z}}$.

**Part (c)**. We now show that $\|\mathbf{y}^t\| \leq \bar{\mathbf{y}}$. For all $t \geq 0$, we have:

$$\|\mathbf{y}^{t+1}\| \overset{①}{=} \|\tfrac{1}{\beta^t}(\mathbf{z}^{t+1} - \mathbf{z}^t) - \mathbf{A}\mathbf{x}^{t+1}\|$$

$$\overset{②}{\leq} \tfrac{1}{\beta^0}(\|\mathbf{z}^{t+1}\| + \|\mathbf{z}^t\|) + \|\mathbf{A}\|\|\mathbf{x}^{t+1}\|$$

$$\overset{③}{\leq} \tfrac{2}{\beta^0}\bar{\mathbf{z}} + \|\mathbf{A}\|\bar{\mathbf{x}},$$

where step ① uses $\mathbf{z}^{t+1} = \mathbf{z}^t + \beta^t(\mathbf{A}\mathbf{x}^{t+1} - \mathbf{y}^{t+1})$; step ② uses the triangle inequality, the norm inequality, and $\beta^0 \leq \beta^t$; step ③ uses the boundedness of $\mathbf{z}^t$ and $\mathbf{x}^t$. Therefore, it holds that $\forall t \geq 0$, $\|\mathbf{y}^t\| \leq \max(\|\mathbf{y}^0\|, \tfrac{2}{\beta^0}\bar{\mathbf{z}} + \|\mathbf{A}\|\bar{\mathbf{x}}) \triangleq \bar{\mathbf{y}}$.

$\square$

### D.4 PROOF OF LEMMA 5.6

*Proof*. We define $\mathcal{U}(\mathbf{x}, \mathbf{y}; \mathbf{z}; \beta, \mu) \triangleq \delta(\mathbf{x}) + f(\mathbf{x}) - g(\mathbf{x}) + h_\mu(\mathbf{y}) + \langle \mathbf{A}\mathbf{x} - \mathbf{y}, \mathbf{z} \rangle + \tfrac{\beta}{2}\|\mathbf{A}\mathbf{x} - \mathbf{y}\|_2^2$.

We define $\underline{\mathbf{v}} \triangleq 8\bar{\mathbf{z}}^2 + \tfrac{1}{2}\chi\bar{\mathbf{z}}^2$, and $\bar{\mathbf{v}} \triangleq 16\bar{\mathbf{z}}^2$.

First, given $h(\mathbf{y})$ is convex, for all $\mathbf{y}, \mathbf{y}' \in \mathbb{R}^m$, we have:

$$h(\mathbf{y}') - h(\mathbf{y}) \leq \langle \partial h(\mathbf{y}'), \mathbf{y}' - \mathbf{y} \rangle. \tag{26}$$

Second, for any $\mathbf{y} \in \mathbb{R}^m$ and $t \geq 1$, we have:

$$\langle \mathbf{A}\mathbf{x}^t - \mathbf{y}^t, \mathbf{z}^t - \partial h(\mathbf{y}) \rangle \overset{①}{=} \tfrac{1}{\beta^{t-1}}\langle \mathbf{z}^t - \mathbf{z}^{t-1}, \mathbf{z}^t - \partial h(\mathbf{y}) \rangle$$

$$\overset{②}{\leq} \tfrac{2}{\beta^t}\|\mathbf{z}^t - \mathbf{z}^{t-1}\| \cdot \|\mathbf{z}^t - \partial h(\mathbf{y})\|$$

$$\overset{③}{\leq} \tfrac{2}{\beta^t} \cdot 2\bar{\mathbf{z}} \cdot (\bar{\mathbf{z}} + \|\partial h(\mathbf{y})\|)$$

$$\overset{④}{\leq} \tfrac{2}{\beta^t} \cdot 2\bar{\mathbf{z}} \cdot 2\bar{\mathbf{z}}, \tag{27}$$

where step ① uses the $\mathbf{z}^{t+1} - \mathbf{z}^t = \beta^t(\mathbf{A}\mathbf{x}^{t+1} - \mathbf{y}^{t+1})$; step ② uses the norm inequality and $\beta^t \leq 2\beta^{t-1}$; step ③ uses $\|\mathbf{z}^t\| \leq \bar{\mathbf{z}}$ and $\|\nabla h_{\mu^t}(\mathbf{y})\| \leq C_h \leq \bar{\mathbf{z}}$, as shown in Lemma D.3.

Third, for any $t \geq 1$, we have:

$$\tfrac{\beta^t}{2}\|\mathbf{A}\mathbf{x}^t - \mathbf{y}^t\|_2^2 \overset{①}{=} \beta^t\|\tfrac{1}{\beta^{t-1}}(\mathbf{z}^t - \mathbf{z}^{t-1})\|_2^2$$

$$\overset{②}{\leq} \beta^t \tfrac{2}{(\beta^t)^2}\|\mathbf{z}^t - \mathbf{z}^{t-1}\|_2^2$$

$$\overset{③}{\leq} \tfrac{2}{\beta^t}\|\mathbf{z}^t - \mathbf{z}^{t-1}\|_2^2$$

$$\overset{④}{\leq} \tfrac{2}{\beta^t}(2\|\mathbf{z}^t\|_2^2 + 2\|\mathbf{z}^{t-1}\|_2^2)$$

$$\overset{⑤}{\leq} \tfrac{8}{\beta^t}\bar{\mathbf{z}}^2, \tag{28}$$

where step ① uses the $\mathbf{z}^{t+1} - \mathbf{z}^t = \beta^t(\mathbf{A}\mathbf{x}^{t+1} - \mathbf{y}^{t+1})$; step ② uses $\beta^t \leq 2\beta^{t-1}$; step ③ uses $\beta^t \geq \beta^0$ for all $t \geq 0$; step ④ uses $\|\mathbf{a} + \mathbf{b}\|_2^2 \leq 2\|\mathbf{a}\|_2^2 + 2\|\mathbf{b}\|_2^2$; step ⑤ uses $\|\mathbf{z}^t\| \leq \bar{\mathbf{z}}$.

**Part (a).** We now derive the lower bound for any $t \geq 1$, as follows:

$$
\begin{aligned}
\mathcal{U}(\mathbf{x}^t, \mathbf{y}^t; \mathbf{z}^t; \beta^t, \mu^t) &\triangleq f(\mathbf{x}^t) + \delta(\mathbf{x}^t) - g(\mathbf{x}^t) + \langle \mathbf{A}\mathbf{x}^t - \mathbf{y}^t, \mathbf{z}^t \rangle + \tfrac{\beta^t}{2} \|\mathbf{A}\mathbf{x}^t - \mathbf{y}^t\|_2^2 + h_{\mu^t}(\mathbf{y}^t) \\
&\overset{①}{\geq} \underline{\mathrm{F}} \cdot d(\mathbf{x}^t) - h(\mathbf{A}\mathbf{x}^t) + \langle \mathbf{A}\mathbf{x}^t - \mathbf{y}^t, \mathbf{z}^t \rangle + \tfrac{\beta^t}{2} \|\mathbf{A}\mathbf{x}^t - \mathbf{y}^t\|_2^2 + h_{\mu^t}(\mathbf{y}^t) \\
&\overset{②}{\geq} \underline{\mathrm{F}} \cdot d(\mathbf{x}^t) + h(\mathbf{y}^t) - h(\mathbf{A}\mathbf{x}^t) + \langle \mathbf{A}\mathbf{x}^t - \mathbf{y}^t, \mathbf{z}^t \rangle + \{h_{\mu^t}(\mathbf{y}^t) - h(\mathbf{y}^t)\} \\
&\overset{③}{\geq} \underline{\mathrm{F}} \cdot d(\mathbf{x}^t) + \langle \mathbf{A}\mathbf{x}^t - \mathbf{y}^t, \mathbf{z}^t - \partial h(\mathbf{A}\mathbf{x}^t) \rangle - \tfrac{\mu^t}{2} C_h^2 \\
&\overset{④}{\geq} \underline{\mathrm{F}} \cdot d(\mathbf{x}^t) - \tfrac{8\overline{\mathbf{z}}^2}{\beta^t} - \tfrac{\chi \overline{\mathbf{z}}^2}{2\beta^t} \\
&\overset{⑤}{=} \underline{\mathrm{F}} \cdot d(\mathbf{x}^t) - \tfrac{\underline{\mathrm{v}}}{\beta^t},
\end{aligned}
$$

where step ① uses $F(\mathbf{x}^t) \triangleq \frac{f(\mathbf{x}^t)+\delta(\mathbf{x}^t)-g(\mathbf{x}^t)+h(\mathbf{A}\mathbf{x}^t)}{d(\mathbf{x}^t)} \geq \underline{\mathrm{F}}$; step ② uses $\tfrac{\beta^t}{2}\|\mathbf{A}\mathbf{x}^t - \mathbf{y}^t\|_2^2 \geq 0$; step ③ uses Inequality (26) with $\mathbf{y} = \mathbf{y}^t$ and $\mathbf{y}' = \mathbf{A}\mathbf{x}^t$, and the fact that $h(\mathbf{y}) \leq h_\mu(\mathbf{y}) + \tfrac{\mu}{2} C_h^2$ (Lemma 3.9(b)); step ④ uses Inequality (27), $\mu^t = \tfrac{\chi}{\beta^t}$, and $C_h \leq \overline{\mathbf{z}}$; step ⑤ uses the definition of $\underline{\mathrm{v}}$.

**Part (b).** We now derive the upper bound for any $t \geq 1$, as follows:

$$
\begin{aligned}
\mathcal{U}(\mathbf{x}^t, \mathbf{y}^t; \mathbf{z}^t; \beta^t, \mu^t) &\triangleq f(\mathbf{x}^t) + \delta(\mathbf{x}^t) - g(\mathbf{x}^t) + \langle \mathbf{A}\mathbf{x}^t - \mathbf{y}^t, \mathbf{z}^t \rangle + \tfrac{\beta^t}{2} \|\mathbf{A}\mathbf{x}^t - \mathbf{y}^t\|_2^2 + h_{\mu^t}(\mathbf{y}^t) \\
&\overset{①}{\leq} \overline{\mathrm{F}} d(\mathbf{x}^t) - h(\mathbf{A}\mathbf{x}^t) + \langle \mathbf{A}\mathbf{x}^t - \mathbf{y}^t, \mathbf{z}^t \rangle + \tfrac{\beta^t}{2} \|\mathbf{A}\mathbf{x}^t - \mathbf{y}^t\|_2^2 + h_{\mu^t}(\mathbf{y}^t) \\
&\overset{②}{\leq} \overline{\mathrm{F}} \cdot \overline{\mathrm{d}} - h(\mathbf{A}\mathbf{x}^t) + \langle \mathbf{A}\mathbf{x}^t - \mathbf{y}^t, \mathbf{z}^t \rangle + \tfrac{\beta^t}{2} \|\mathbf{A}\mathbf{x}^t - \mathbf{y}^t\|_2^2 + h(\mathbf{y}^t) \\
&\overset{③}{\leq} \overline{\mathrm{F}} \cdot \overline{\mathrm{d}} + \langle \mathbf{A}\mathbf{x}^t - \mathbf{y}^t, \mathbf{z}^t - \partial h(\mathbf{y}^t) \rangle + \tfrac{\beta^t}{2} \|\mathbf{A}\mathbf{x}^t - \mathbf{y}^t\|_2^2 \\
&\overset{④}{\leq} \overline{\mathrm{F}} \cdot \overline{\mathrm{d}} + \tfrac{(8+8)\overline{\mathbf{z}}^2}{\beta^t} \\
&\leq \overline{\mathrm{F}} \cdot \overline{\mathrm{d}} + \tfrac{\overline{\mathrm{v}}}{\beta^t},
\end{aligned}
$$

where step ① uses $F(\mathbf{x}^t) \triangleq \frac{f(\mathbf{x}^t)+\delta(\mathbf{x}^t)-g(\mathbf{x}^t)+h(\mathbf{A}\mathbf{x}^t)}{d(\mathbf{x}^t)} \leq \overline{\mathrm{F}}$; step ② uses $d(\mathbf{x}^t) \leq \overline{\mathrm{d}}$, and $h_\mu(\mathbf{y}) \leq h(\mathbf{y})$ for all $\mathbf{y}$ and $\mu$; step ③ uses Inequality (26) with $\mathbf{y} = \mathbf{A}\mathbf{x}^t$ and $\mathbf{y}' = \mathbf{y}^t$; step ④ uses Inequalities (27) and (28).

$\square$

### D.5 PROOF OF LEMMA 5.9

*Proof.* **Part (a).** For any $t \geq 0$, we have:

$$
\frac{\beta^{t+1}}{\beta^t} = \frac{1+\xi(t+1)^p}{1+\xi t^p} \overset{①}{\leq} \frac{1+\xi(t^p+1)}{1+\xi t^p} \overset{②}{\leq} 1 + \xi, \tag{29}
$$

where step ① uses the fact that $(t+1)^p \leq t^p + 1^p$ for all $p \in (0,1)$ and $t \geq 0$; step ② uses the fact that $\frac{a+\xi}{a} \leq 1 + \xi$ for all $a \geq 1$ and $\xi \geq 0$.

**Part (b).** For all $t \geq 1$, we derive the upper bound and lower bound for $\lambda^t$:

$$
\lambda^t \triangleq \frac{\mathcal{U}(\mathbf{x}^t, \mathbf{y}^t; \mathbf{z}^t; \beta^t, \mu^t)}{d(\mathbf{x}^t)} \leq \frac{\overline{\mathrm{F}} \cdot \overline{\mathrm{d}} + \overline{\mathrm{v}}/\beta^t}{\underline{\mathrm{d}}} \leq \frac{\overline{\mathrm{F}} \cdot \overline{\mathrm{d}} + \overline{\mathrm{v}}/\beta^0}{\underline{\mathrm{d}}} \triangleq \overline{\lambda}.
$$

$$
\lambda^t \triangleq \frac{\mathcal{U}(\mathbf{x}^t, \mathbf{y}^t; \mathbf{z}^t; \beta^t, \mu^t)}{d(\mathbf{x}^t)} \geq \frac{\underline{\mathrm{F}} \cdot \underline{\mathrm{d}} - \underline{\mathrm{v}}/\beta^t}{\overline{\mathrm{d}}} \geq \frac{\underline{\mathrm{F}} \cdot \underline{\mathrm{d}} - \underline{\mathrm{v}}/\beta^0}{\overline{\mathrm{d}}} \triangleq \underline{\lambda}.
$$

**Part (c).** For all $t \geq 0$, we derive the upper bound and lower bound for $\alpha^{t+1}$:

$$
\alpha^{t+1} \triangleq \frac{\sqrt{d(\mathbf{x}^t)}}{\mathcal{U}(\mathbf{x}^t, \mathbf{y}^t; \mathbf{z}^t; \beta^t, \mu^t)} \leq \frac{\sqrt{\overline{\mathrm{d}}}}{\underline{\mathrm{F}} \cdot \underline{\mathrm{d}} - \underline{\mathrm{v}}/\beta^t} \leq \frac{\sqrt{\overline{\mathrm{d}}}}{\underline{\mathrm{F}} \cdot \underline{\mathrm{d}} - \underline{\mathrm{v}}/\beta^0} \triangleq \overline{\alpha}.
$$

$$
\alpha^{t+1} \triangleq \frac{\sqrt{d(\mathbf{x}^t)}}{\mathcal{U}(\mathbf{x}^t, \mathbf{y}^t; \mathbf{z}^t; \beta^t, \mu^t)} \geq \frac{\sqrt{\underline{\mathrm{d}}}}{\overline{\mathrm{F}} \cdot \overline{\mathrm{d}} + \overline{\mathrm{v}}/\beta^t} \geq \frac{\sqrt{\underline{\mathrm{d}}}}{\overline{\mathrm{F}} \cdot \overline{\mathrm{d}} + \overline{\mathrm{v}}/\beta^0} \triangleq \underline{\alpha}.
$$

**Part (d)**. We first focus on FADMM-D with $\ell(\beta^t) \triangleq L_f + \beta^t \|\mathbf{A}\|_2^2 + \lambda^t W_d$. For all $t \geq 1$, we have:

$$\ell(\beta^t) \geq \beta^t \|\mathbf{A}\|_2^2.$$
$$\ell(\beta^t) \leq \frac{\beta^t L_f}{\beta^0} + \beta^t \|\mathbf{A}\|_2^2 + \frac{\beta^t}{\beta^0} \overline{\lambda} W_d.$$

We now focus on FADMM-Q with $\ell(\beta^t) \triangleq L_f + \beta^t \|\mathbf{A}\|_2^2 + (2/\alpha^t) W_d$. For all $t \geq 1$, we have:

$$\ell(\beta^t) \geq \beta^t \|\mathbf{A}\|_2^2.$$
$$\ell(\beta^t) \leq \frac{\beta^t L_f}{\beta^0} + \beta^t \|\mathbf{A}\|_2^2 + \frac{\beta^t}{\beta^0} \frac{2}{\underline{\alpha}} W_d.$$

$\square$

# E   PROOFS FOR SECTION 5

## E.1   PROOF OF LEMMA 5.10

*Proof.* We define $\mathcal{S}(\mathbf{x}, \mathbf{y}; \mathbf{z}; \beta) \triangleq f(\mathbf{x}) + \langle \mathbf{Ax} - \mathbf{y}, \mathbf{z} \rangle + \frac{\beta}{2} \|\mathbf{Ax} - \mathbf{y}\|_2^2$.

We define $s(\mathbf{x}) \triangleq \mathcal{S}(\mathbf{x}, \mathbf{y}^t, \mathbf{z}^t; \beta^t)$, where $t$ is known from context.

We define $\mathcal{L}(\mathbf{x}, \mathbf{y}; \mathbf{z}; \beta, \mu) \triangleq \frac{1}{d(\mathbf{x})} \cdot \mathcal{U}(\mathbf{x}, \mathbf{y}; \mathbf{z}; \beta, \mu)$.

We define $\mathcal{U}(\mathbf{x}, \mathbf{y}; \mathbf{z}; \beta, \mu) \triangleq \mathcal{S}(\mathbf{x}, \mathbf{y}; \mathbf{z}; \beta) + \delta(\mathbf{x}) - g(\mathbf{x}) + h_\mu(\mathbf{y})$.

We define $\ell(\beta^t) \triangleq L_f + \beta^t \|\mathbf{A}\|_2^2 + \lambda^t W_d$, where $\lambda^t = \frac{\mathcal{U}(\mathbf{x}^t, \mathbf{y}^t; \mathbf{z}^t; \beta^t, \mu^t)}{d(\mathbf{x}^t)}$.

We define $\varepsilon_x \triangleq (\theta - 1)\underline{\ell}/(2\overline{d}) > 0$.

Initially, using the optimality condition of $\mathbf{x}^{t+1} \in \arg\min_{\mathbf{x}} \dot{\mathcal{M}}^t(\mathbf{x}; \mathbf{x}^t, \lambda^t)$, we have: $\dot{\mathcal{M}}^t(\mathbf{x}^{t+1}; \mathbf{x}^t, \lambda^t) \leq \dot{\mathcal{M}}^t(\mathbf{x}^t; \mathbf{x}^t, \lambda^t)$. This leads to $\langle \mathbf{x}^{t+1} - \mathbf{x}^t, \nabla s(\mathbf{x}^t) - \partial g(\mathbf{x}^t) - \lambda^t \partial d(\mathbf{x}^t) \rangle + \frac{\theta \ell(\beta^t)}{2} \|\mathbf{x}^{t+1} - \mathbf{x}^t\|_2^2 + \delta(\mathbf{x}^{t+1}) \leq 0 + 0 + \delta(\mathbf{x}^t)$. Rearranging terms yields:

$$\delta(\mathbf{x}^{t+1}) - \delta(\mathbf{x}^t) + \frac{\theta \ell(\beta^t)}{2} \|\mathbf{x}^{t+1} - \mathbf{x}^t\|_2^2$$
$$\leq \langle \mathbf{x}^t - \mathbf{x}^{t+1}, \nabla s(\mathbf{x}^t) - \partial g(\mathbf{x}^t) - \lambda^t \partial d(\mathbf{x}^t) \rangle$$
$$\overset{\text{①}}{\leq} s(\mathbf{x}^t) - s(\mathbf{x}^{t+1}) + \frac{L_f + \beta^t \|\mathbf{A}\|_2^2}{2} \|\mathbf{x}^{t+1} - \mathbf{x}^t\|_2^2 + g(\mathbf{x}^{t+1}) - g(\mathbf{x}^t)$$
$$\quad + \lambda^t (d(\mathbf{x}^{t+1}) - d(\mathbf{x}^t)) + \lambda^t \frac{W_d}{2} \|\mathbf{x}^{t+1} - \mathbf{x}^t\|_2^2$$
$$\overset{\text{②}}{=} s(\mathbf{x}^t) - s(\mathbf{x}^{t+1}) + \frac{\ell(\beta^t)}{2} \|\mathbf{x}^{t+1} - \mathbf{x}^t\|_2^2 + g(\mathbf{x}^{t+1}) - g(\mathbf{x}^t) + \lambda^t (d(\mathbf{x}^{t+1}) - d(\mathbf{x}^t)), \quad (30)$$

where step ① uses the facts that the function $s(\mathbf{x})$ is $(L_f + \beta^t \|\mathbf{A}\|_2^2)$-smooth *w.r.t.* $\mathbf{x}$, $\lambda^t > 0$, $g(\mathbf{x})$ is convex, and $d(\mathbf{x})$ is $W_d$-weakly convex, yielding the following inequalities:

$$s(\mathbf{x}^{t+1}) - s(\mathbf{x}^t) \leq \langle \mathbf{x}^{t+1} - \mathbf{x}^t, \nabla s(\mathbf{x}^t) \rangle + \frac{L_f + \beta^t \|\mathbf{A}\|_2^2}{2} \|\mathbf{x}^{t+1} - \mathbf{x}^t\|_2^2,$$
$$g(\mathbf{x}^t) - g(\mathbf{x}^{t+1}) \leq \langle \partial g(\mathbf{x}^t), \mathbf{x}^t - \mathbf{x}^{t+1} \rangle,$$
$$\lambda^t d(\mathbf{x}^t) - \lambda^t d(\mathbf{x}^{t+1}) \leq \lambda^t \langle \partial d(\mathbf{x}^t), \mathbf{x}^t - \mathbf{x}^{t+1} \rangle + \lambda^t \frac{W_d}{2} \|\mathbf{x}^{t+1} - \mathbf{x}^t\|_2^2; \quad (31)$$

step ② uses the definition of $\ell(\beta^t) = L_f + \beta^t \|\mathbf{A}\|_2^2 + \lambda^t W_d$.

We further derive:

$$
\begin{aligned}
&\mathcal{L}(\mathbf{x}^{t+1}, \mathbf{y}^t; \mathbf{z}^t; \beta^t, \mu^t) - \mathcal{L}(\mathbf{x}^t, \mathbf{y}^t; \mathbf{z}^t; \beta^t, \mu^t) \\
&\overset{\text{①}}{=} \tfrac{1}{d(\mathbf{x}^{t+1})}\{h_{\mu^t}(\mathbf{y}^t) + \delta(\mathbf{x}^{t+1}) + s(\mathbf{x}^{t+1}) - g(\mathbf{x}^{t+1})\} - \lambda^t \\
&\overset{\text{②}}{\leq} \tfrac{1}{d(\mathbf{x}^{t+1})}\{h_{\mu^t}(\mathbf{y}^t) + \delta(\mathbf{x}^t) - g(\mathbf{x}^t) + \tfrac{\ell(\beta^t)(1-\theta)}{2}\|\mathbf{x}^{t+1} - \mathbf{x}^t\|_2^2 + s(\mathbf{x}^t) - \lambda^t d(\mathbf{x}^t)\} + \lambda^t - \lambda^t \\
&\overset{\text{③}}{=} \tfrac{1}{d(\mathbf{x}^{t+1})}\{h_{\mu^t}(\mathbf{y}^t) + \delta(\mathbf{x}^t) - g(\mathbf{x}^t) + \tfrac{\ell(\beta^t)(1-\theta)}{2}\|\mathbf{x}^{t+1} - \mathbf{x}^t\|_2^2 - \{\delta(\mathbf{x}^t) - g(\mathbf{x}^t) + h_{\mu^t}(\mathbf{y}^t)\}\} \\
&= \tfrac{1}{d(\mathbf{x}^{t+1})}\tfrac{\ell(\beta^t)(1-\theta)}{2}\|\mathbf{x}^{t+1} - \mathbf{x}^t\|_2^2 \\
&\overset{\text{④}}{\leq} \tfrac{\ell(\beta^t)\|\mathbf{x}^{t+1}-\mathbf{x}^t\|_2^2}{2\overline{d}} \cdot \{1 - \theta\} \\
&\overset{\text{⑤}}{\leq} -\varepsilon_x \beta^t \|\mathbf{x}^{t+1} - \mathbf{x}^t\|_2^2,
\end{aligned}
$$

where step ① uses the definition of $\mathcal{L}(\mathbf{x}^t, \mathbf{y}^t; \mathbf{z}^t; \beta^t, \mu^t) = \lambda^t$; step ② uses Inequality (30); step ③ uses $\lambda^t d(\mathbf{x}^t) - s(\mathbf{x}^t) = \delta(\mathbf{x}^t) - g(\mathbf{x}^t) + h_{\mu^t}(\mathbf{y}^t)$; step ④ uses $d(\mathbf{x}) \leq \overline{d}$ and $\theta > 1$; step ⑤ uses the definition of $\varepsilon_x \triangleq (\theta - 1)\underline{\ell}/(2\overline{d}) > 0$.

$\square$

### E.2  PROOF OF LEMMA 5.11

*Proof.* We define $\mathcal{L}(\mathbf{x}, \mathbf{y}; \mathbf{z}; \beta, \mu) \triangleq \tfrac{1}{d(\mathbf{x})} \cdot \{f(\mathbf{x}) - g(\mathbf{x}) + h_\mu(\mathbf{y}) + \langle \mathbf{A}\mathbf{x} - \mathbf{y}, \mathbf{z}\rangle + \tfrac{\beta}{2}\|\mathbf{A}\mathbf{x} - \mathbf{y}\|_2^2\}$.

We define $\mathbb{T}^t \triangleq 12(1+\xi)C_h^2/(\beta^0 \underline{d}t)$, and $\mathbb{T}^t \triangleq C_h^2 \mu^t/(2\underline{d})$.

We define $\varepsilon_z \triangleq \xi/(2\overline{d})$, and $\varepsilon_y \triangleq \{1 - 4(1+\xi)/\chi^2\}/(2\overline{d})$.

First, we focus on a decrease for the function $\mathcal{L}(\mathbf{x}, \mathbf{y}; \mathbf{z}; \beta, \mu)$ *w.r.t.* $\mathbf{y}$. We have:

$$
\begin{aligned}
&\mathcal{L}(\mathbf{x}^{t+1}, \mathbf{y}^{t+1}; \mathbf{z}^t; \beta^t, \mu^t) - \mathcal{L}(\mathbf{x}^{t+1}, \mathbf{y}^t; \mathbf{z}^t; \beta^t, \mu^t) \\
&\overset{\text{①}}{=} \tfrac{1}{d(\mathbf{x}^{t+1})}\{\langle \mathbf{y}^t - \mathbf{y}^{t+1}, \mathbf{z}^t\rangle + \tfrac{\beta^t}{2}\|\mathbf{A}\mathbf{x}^{t+1} - \mathbf{y}^{t+1}\|_2^2 - \tfrac{\beta^t}{2}\|\mathbf{A}\mathbf{x}^{t+1} - \mathbf{y}^t\|_2^2 + h_{\mu^t}(\mathbf{y}^{t+1}) - h_{\mu^t}(\mathbf{y}^t)\} \\
&\overset{\text{②}}{=} \tfrac{1}{d(\mathbf{x}^{t+1})}\{\langle \mathbf{y}^t - \mathbf{y}^{t+1}, \mathbf{z}^t + \beta^t(\mathbf{A}\mathbf{x}^{t+1} - \mathbf{y}^{t+1})\rangle + h_{\mu^t}(\mathbf{y}^{t+1}) - h_{\mu^t}(\mathbf{y}^t) - \tfrac{\beta^t}{2}\|\mathbf{y}^{t+1} - \mathbf{y}^t\|_2^2\} \\
&\overset{\text{③}}{=} \tfrac{1}{d(\mathbf{x}^{t+1})}\{\langle \mathbf{y}^t - \mathbf{y}^{t+1}, \mathbf{z}^{t+1}\rangle + h_{\mu^t}(\mathbf{y}^{t+1}) - h_{\mu^t}(\mathbf{y}^t) - \tfrac{\beta^t}{2}\|\mathbf{y}^{t+1} - \mathbf{y}^t\|_2^2\} \\
&\overset{\text{④}}{\leq} \tfrac{1}{d(\mathbf{x}^{t+1})}\{\langle \mathbf{y}^t - \mathbf{y}^{t+1}, \mathbf{z}^{t+1} - \nabla h_{\mu^t}(\mathbf{y}^{t+1})\rangle - \tfrac{\beta^t}{2}\|\mathbf{y}^{t+1} - \mathbf{y}^t\|_2^2\} \\
&\overset{\text{⑤}}{=} \tfrac{-\beta^t\|\mathbf{y}^{t+1}-\mathbf{y}^t\|_2^2}{2d(\mathbf{x}^{t+1})},
\end{aligned}
\tag{32}
$$

where step ① uses the definition of $\mathcal{L}(\mathbf{x}, \mathbf{y}; \mathbf{z}; \beta, \mu)$; step ② uses the Pythagoras relation that $\tfrac{1}{2}\|\mathbf{a} - \mathbf{c}\|_2^2 - \tfrac{1}{2}\|\mathbf{b} - \mathbf{c}\|_2^2 = -\tfrac{1}{2}\|\mathbf{a} - \mathbf{b}\|_2^2 + \langle \mathbf{a} - \mathbf{c}, \mathbf{a} - \mathbf{b}\rangle$; step ③ uses $\mathbf{z}^{t+1} - \mathbf{z}^t = \beta^t(\mathbf{A}\mathbf{x}^{t+1} - \mathbf{y}^{t+1})$; step ④ uses the convexity of $h_{\mu^t}(\cdot)$; step ⑤ uses the optimality for $\mathbf{y}^{t+1}$ as in Lemma 5.3(a) that: $\nabla h_{\mu^t}(\mathbf{y}^{t+1}) = \mathbf{z}^{t+1}$.

Second, we focus on a decrease for the function $\mathcal{L}(\mathbf{x}, \mathbf{y}; \mathbf{z}; \beta, \mu)$ *w.r.t.* $\{\mathbf{z}, \beta\}$. We have:

$$\frac{\xi}{2\overline{d}} \frac{1}{\beta} \|\mathbf{z}^{t+1} - \mathbf{z}^t\|_2^2 + \mathcal{L}(\mathbf{x}^{t+1}, \mathbf{y}^{t+1}; \mathbf{z}^{t+1}; \beta^{t+1}, \mu^t) - \mathcal{L}(\mathbf{x}^{t+1}, \mathbf{y}^{t+1}; \mathbf{z}^t; \beta^t, \mu^t)$$

$$\overset{①}{\leq} \frac{\xi}{2d(\mathbf{x}^{t+1})} \frac{1}{\beta^t} \|\mathbf{z}^{t+1} - \mathbf{z}^t\|_2^2 + \{\mathcal{L}(\mathbf{x}^{t+1}, \mathbf{y}^{t+1}; \mathbf{z}^{t+1}; \beta^t, \mu^t) - \mathcal{L}(\mathbf{x}^{t+1}, \mathbf{y}^{t+1}; \mathbf{z}^t; \beta^t, \mu^t)\}$$
$$\quad + \{\mathcal{L}(\mathbf{x}^{t+1}, \mathbf{y}^{t+1}; \mathbf{z}^{t+1}; \beta^{t+1}, \mu^t) - \mathcal{L}(\mathbf{x}^{t+1}, \mathbf{y}^{t+1}; \mathbf{z}^{t+1}; \beta^t, \mu^t)\}$$

$$\overset{②}{=} \frac{1}{d(\mathbf{x}^{t+1})} \cdot \{\frac{\xi}{2\beta^t} \|\mathbf{z}^{t+1} - \mathbf{z}^t\|_2^2 + \langle \mathbf{A}\mathbf{x}^{t+1} - \mathbf{y}^{t+1}, \mathbf{z}^{t+1} - \mathbf{z}^t \rangle + \frac{(\beta^{t+1} - \beta^t)\|\mathbf{A}\mathbf{x}^{t+1} - \mathbf{y}^{t+1}\|_2^2}{2}\}$$

$$\overset{③}{\leq} \frac{1}{d(\mathbf{x}^{t+1})} \cdot \{\frac{\xi}{2\beta^t} \|\mathbf{z}^{t+1} - \mathbf{z}^t\|_2^2 + \frac{\|\mathbf{z}^{t+1} - \mathbf{z}^t\|_2^2}{\beta^t} + \frac{(\beta^t(1+\xi) - \beta^t)\|\mathbf{z}^{t+1} - \mathbf{z}^t\|_2^2}{2(\beta^t)^2}\}$$

$$= \frac{1}{d(\mathbf{x}^{t+1})} (\frac{\xi}{2} + 1 + \frac{\xi}{2}) \frac{1}{\beta^t} \|\mathbf{z}^{t+1} - \mathbf{z}^t\|_2^2$$

$$\overset{④}{\leq} \frac{1}{d(\mathbf{x}^{t+1})} \frac{1+\xi}{\beta^t} \{\frac{2(\beta^t)^2}{\chi^2} \|\mathbf{y}^{t+1} - \mathbf{y}^t\|_2^2 + 12C_h^2 (\frac{1}{t} - \frac{1}{t+1})\}$$

$$\overset{⑥}{\leq} \{\frac{1}{d(\mathbf{x}^{t+1})} \frac{2(1+\xi)}{\chi^2} \beta^t \|\mathbf{y}^{t+1} - \mathbf{y}^t\|_2^2\} + \frac{12(1+\xi)C_h^2}{\beta^0 \underline{d}}(\frac{1}{t} - \frac{1}{t+1}), \tag{33}$$

where step ① uses $d(\mathbf{x}^{t+1}) \leq \overline{d}$; step ② uses the definition of $\mathcal{L}(\mathbf{x}, \mathbf{y}; \mathbf{z}; \beta, \mu)$; step ③ uses $\beta^t(\mathbf{A}\mathbf{x}^{t+1} - \mathbf{y}^{t+1}) = \mathbf{z}^{t+1} - \mathbf{z}^t$ and $\beta^{t+1} \leq \beta^t(1 + \xi)$; step ④ uses Lemma 5.4(b); step ⑤ uses Lemma 5.4; step ⑥ uses $d(\mathbf{x}^{t+1}) \geq \underline{d}$, and $\beta^t \geq \beta^0$.

Third, we focus on a decrease for the function $\mathcal{L}(\mathbf{x}, \mathbf{y}; \mathbf{z}; \beta, \mu)$ *w.r.t.* $\mu$. We have:

$$\mathcal{L}(\mathbf{x}^{t+1}, \mathbf{y}^{t+1}; \mathbf{z}^{t+1}; \beta^{t+1}, \mu^{t+1}) - \mathcal{L}(\mathbf{x}^{t+1}, \mathbf{y}^{t+1}; \mathbf{z}^{t+1}; \beta^{t+1}, \mu^t)$$

$$= \frac{1}{d(\mathbf{x}^{t+1})} \{h_{\mu^{t+1}}(\mathbf{y}^{t+1}) - h_{\mu^t}(\mathbf{y}^{t+1})\}$$

$$\overset{①}{\leq} \frac{1}{2d(\mathbf{x}^{t+1})} (\mu^t - \mu^{t+1}) \cdot C_h^2$$

$$\overset{②}{\leq} \frac{1}{2\underline{d}} (\mu^t - \mu^{t+1}) \cdot C_h^2, \tag{34}$$

where step ① uses Lemma 3.9(e); step ② uses $d(\mathbf{x}^t) \geq \underline{d}$.

Adding Inequalities (32), (33), and (34), we have:

$$\mathcal{L}(\mathbf{x}^{t+1}, \mathbf{y}^{t+1}; \mathbf{z}^{t+1}; \beta^{t+1}, \mu^{t+1}) - \mathcal{L}(\mathbf{x}^{t+1}, \mathbf{y}^t; \mathbf{z}^t; \beta^t, \mu^t) + \mathbb{T}^{t+1} + \mathbb{U}^{t+1} - \mathbb{T}^t - \mathbb{U}^t$$

$$\leq -\varepsilon_z \beta^t \|\mathbf{A}\mathbf{x}^{t+1} - \mathbf{y}^{t+1}\|_2^2 - \frac{1}{2d(\mathbf{x}^{t+1})} \cdot \beta^t \|\mathbf{y}^{t+1} - \mathbf{y}^t\|_2^2 \cdot \{1 - \frac{4(1+\xi)}{\chi^2}\}$$

$$\overset{①}{\leq} -\varepsilon_z \beta^t \|\mathbf{A}\mathbf{x}^{t+1} - \mathbf{y}^{t+1}\|_2^2 - \beta^t \|\mathbf{y}^{t+1} - \mathbf{y}^t\|_2^2 \cdot \varepsilon_y,$$

where step ① uses the definition of $\varepsilon_y \triangleq \{1 - 4(1+\xi)/\chi^2\}/(2\overline{d})$.

$\square$

### E.3  PROOF OF LEMMA 5.12

*Proof.* We define $\mathbb{L}^t \triangleq \mathcal{L}(\mathbf{x}^t, \mathbf{y}^t; \mathbf{z}^t; \beta^t, \mu^t)$, and $\mathbb{T}^t = 12(1+\xi)C_h^2/(\beta^0 \underline{d} t)$, and $\mathbb{U}^t = C_h^2 \mu^t/(2\underline{d})$. We define $\mathbb{P}^t \triangleq \mathbb{L}^t + \mathbb{T}^t + \mathbb{U}^t$.

**Part (a)**. We derive the following inequalities:

$$\mathbb{P}^t \triangleq \mathbb{L}^t + \mathbb{T}^t + \mathbb{U}^t$$

$$\overset{①}{\geq} \mathcal{L}(\mathbf{x}^t, \mathbf{y}^t; \mathbf{z}^t; \beta^t, \mu^t) + 0$$

$$\overset{②}{\geq} \frac{\mathcal{U}(\mathbf{x}^t, \mathbf{y}^t; \mathbf{z}^t; \beta^t, \mu^t)}{d(\mathbf{x}^t)}$$

$$\overset{③}{\geq} \frac{\mathbf{F} \cdot \mathbf{d} - \mathbf{v}/\beta^t}{\overline{d}}$$

$$\overset{④}{\geq} \frac{\mathbf{F} \cdot \mathbf{d} - \mathbf{v}/\beta^0}{\overline{d}} \triangleq \mathbb{P},$$

where step ① uses $\mathbb{T}^t \geq 0$ and $\mathbb{U}^t \geq 0$; step ② uses the definition of $\mathcal{L}(\mathbf{x}^t, \mathbf{y}^t; \mathbf{z}^t; \beta^t, \mu^t)$; step ③ uses $d(\mathbf{x}^t) \leq \overline{\mathrm{d}}$, and the lower bound of $\mathcal{U}(\mathbf{x}^t, \mathbf{y}^t; \mathbf{z}^t; \beta^t, \mu^t)$ in Lemma 5.6; step ④ uses $\beta^t \geq \beta^0$.

**Part (b)**. Combing Lemmas (5.10) and (5.11) together, we have:

$$\varepsilon_y \beta^t \|\mathbf{y}^{t+1} - \mathbf{y}^t\|_2^2 + \varepsilon_x \beta^t \|\mathbf{x}^{t+1} - \mathbf{x}^t\|_2^2 + \varepsilon_z \beta^t \|\mathbf{A}\mathbf{x}^{t+1} - \mathbf{y}^{t+1}\|_2^2$$
$$\leq \mathbb{L}^t - \mathbb{L}^t + \mathbb{T}^t - \mathbb{T}^{t+1} + \mathbb{U}^t - \mathbb{U}^{t+1}$$
$$= \mathbb{P}^t - \mathbb{P}^{t+1}.$$

$\square$

### E.4 PROOF OF THEOREM 5.13

*Proof.* We define $c_0 \triangleq \frac{1}{\min(\varepsilon_x, \varepsilon_y, \varepsilon_z)} \cdot \{\mathbb{P}^1 - \underline{\mathbb{P}}\}$.

We define $\mathcal{E}^t \triangleq \beta^t \{\|\mathbf{x}^{t+1} - \mathbf{x}^t\|_2^2 + \|\mathbf{y}^{t+1} - \mathbf{y}^t\|_2^2 + \|\mathbf{A}\mathbf{x}^{t+1} - \mathbf{y}^{t+1}\|_2^2\}$.

We define $\mathcal{E}_+^t \triangleq \beta^t \{\|\mathbf{x}^{t+1} - \mathbf{x}^t\| + \|\mathbf{y}^{t+1} - \mathbf{y}^t\| + \|\mathbf{A}\mathbf{x}^{t+1} - \mathbf{y}^{t+1}\|\}$.

Using Lemma 5.12, we have:

$$0 \leq -\min(\varepsilon_x, \varepsilon_y, \varepsilon_z)\mathcal{E}^t + \mathbb{P}^t - \mathbb{P}^{t+1}. \tag{35}$$

Telescoping Inequality (35) over $t$ from $1$ to $T$, we have:

$$0 \leq \frac{1}{\min(\varepsilon_x, \varepsilon_y, \varepsilon_z)} \cdot \sum_{t=1}^T \{\mathbb{P}^t - \mathbb{P}^{t+1}\} - \sum_{t=1}^T \mathcal{E}^t$$
$$= \frac{1}{\min(\varepsilon_x, \varepsilon_y, \varepsilon_z)} \cdot \{\mathbb{P}^1 - \mathbb{P}^{T+1}\} - \sum_{t=1}^T \mathcal{E}^t$$
$$\overset{①}{\leq} \frac{1}{\min(\varepsilon_x, \varepsilon_y, \varepsilon_z)} \cdot \{\mathbb{P}^1 - \underline{\mathbb{P}}\} - \sum_{t=1}^T \mathcal{E}^t$$
$$\overset{②}{\leq} c_0 - \frac{1}{\beta^T} \sum_{t=1}^T \|\beta^t(\mathbf{x}^{t+1} - \mathbf{x}^t)\|_2^2 + \|\beta^t(\mathbf{y}^{t+1} - \mathbf{y}^t)\|_2^2 + \|\beta^t(\mathbf{A}\mathbf{x}^{t+1} - \mathbf{y}^{t+1})\|_2^2$$
$$\overset{③}{\leq} c_0 - \frac{1}{\beta^T} \frac{1}{3T} \{\sum_{t=1}^T \|\beta^t(\mathbf{x}^{t+1} - \mathbf{x}^t)\| + \|\beta^t(\mathbf{y}^{t+1} - \mathbf{y}^t)\| + \|\beta^t(\mathbf{A}\mathbf{x}^{t+1} - \mathbf{y}^{t+1})\|\}^2$$
$$\overset{④}{\leq} c_0 - \frac{1}{\beta^T 3T} \{\sum_{t=1}^T \mathcal{E}_+^t\}^2 \tag{36}$$

where step ① uses $\mathbb{P}^t \geq \underline{\mathbb{P}}$ for all $t$; step ② uses the definition of $c_0$, the definition of $\mathcal{E}^t$, and the Hölder's inequality that $\langle \mathbf{a}, \mathbf{b} \rangle \leq \|\mathbf{a}\|_1 \|\mathbf{b}\|_\infty$; step ③ uses the fact that $\|\mathbf{a}\|_2^2 \geq \frac{1}{n}\|\mathbf{a}\|_1^2$; step ④ uses the definition of $\mathcal{E}_+^t$. We further obtain from Inequality (43) that

$$\sum_{t=1}^T \mathcal{E}_+^t \leq (3W)^{1/2}(\beta^T T)^{1/2} \quad \overset{①}{\Rightarrow} \quad \frac{1}{T}\sum_{t=1}^T \mathcal{E}_+^t \leq \mathcal{O}(T^{(p-1)/2}),$$

where step ① $\beta^t = \beta^0(1 + \xi t^p) = \mathcal{O}(t^p)$.

$\square$

### E.5 PROOF OF LEMMA 5.15

*Proof.* We define $\mathcal{S}(\mathbf{x}, \mathbf{y}; \mathbf{z}; \beta) \triangleq f(\mathbf{x}) + \langle \mathbf{A}\mathbf{x} - \mathbf{y}, \mathbf{z} \rangle + \frac{\beta}{2}\|\mathbf{A}\mathbf{x} - \mathbf{y}\|_2^2$.

We define $s(\mathbf{x}) \triangleq \mathcal{S}(\mathbf{x}, \mathbf{y}^t, \mathbf{z}^t; \beta^t)$, where $t$ is known from context.

We define $\mathcal{K}(\alpha, \mathbf{x}, \mathbf{y}, \mathbf{z}; \beta, \mu) = -2\alpha\sqrt{d(\mathbf{x})} + \alpha^2 \mathcal{U}(\mathbf{x}, \mathbf{y}; \mathbf{z}; \beta, \mu)$.

We define $\mathcal{U}(\mathbf{x}, \mathbf{y}; \mathbf{z}; \beta, \mu) \triangleq \mathcal{S}(\mathbf{x}, \mathbf{y}; \mathbf{z}; \beta) + \delta(\mathbf{x}) - g(\mathbf{x}) + h_\mu(\mathbf{y})$.

We define $\ell(\beta^t) \triangleq L_f + \beta^t \|\mathbf{A}\|_2^2 + \frac{2}{\alpha^{t+1}}W_d$, where $\alpha^{t+1} = \sqrt{d(\mathbf{x}^t)}/\mathcal{U}(\mathbf{x}^t, \mathbf{y}^t; \mathbf{z}^t; \beta^t, \mu^t)$.

We define $\varepsilon_x \triangleq \frac{1}{2}\underline{\alpha}^2 \underline{\ell}(\theta - 1)$.

Initially, using the optimality condition of $\mathbf{x}^{t+1} \in \arg\min_{\mathbf{x}} \ddot{\mathcal{M}}^t(\mathbf{x}; \mathbf{x}^t, \alpha^{t+1})$, we have $\ddot{\mathcal{M}}^t(\mathbf{x}^{t+1}; \mathbf{x}^t, \alpha^{t+1}) \leq \ddot{\mathcal{M}}^t(\mathbf{x}^t; \mathbf{x}^t, \alpha^{t+1})$. This results in $\langle \mathbf{x}^{t+1} - \mathbf{x}^t, \nabla s(\mathbf{x}^t) - \partial g(\mathbf{x}^t) - $

$\frac{2}{\alpha^{t+1}}\partial\sqrt{d(\mathbf{x}^t)}\rangle + \frac{\theta\ell(\beta^t)}{2}\|\mathbf{x}^{t+1} - \mathbf{x}^t\|_2^2 + \delta(\mathbf{x}^{t+1}) \leq 0 + 0 + \delta(\mathbf{x}^t)$. Rearranging terms yields:

$$\delta(\mathbf{x}^{t+1}) - \delta(\mathbf{x}^t) + \frac{\theta\ell(\beta^t)}{2}\|\mathbf{x}^{t+1} - \mathbf{x}^t\|_2^2$$

$$\leq \langle \mathbf{x}^t - \mathbf{x}^{t+1}, \nabla s(\mathbf{x}^t) - \partial g(\mathbf{x}^t) - \frac{2}{\alpha^{t+1}}\partial\sqrt{d(\mathbf{x}^t)}\rangle$$

$$\overset{\textcircled{1}}{\leq} s(\mathbf{x}^t) - s(\mathbf{x}^{t+1}) + \frac{L_f + \beta^t\|\mathbf{A}\|_2^2}{2}\|\mathbf{x}^{t+1} - \mathbf{x}^t\|_2^2 + g(\mathbf{x}^{t+1}) - g(\mathbf{x}^t)$$

$$+ \frac{2}{\alpha^{t+1}}\{\sqrt{d(\mathbf{x}^t)} - \sqrt{d(\mathbf{x}^{t+1})} + \frac{W_d}{2}\|\mathbf{x}^{t+1} - \mathbf{x}^t\|_2^2\}$$

$$\overset{\textcircled{2}}{\leq} s(\mathbf{x}^t) - s(\mathbf{x}^{t+1}) + \frac{\ell(\beta^t)}{2}\|\mathbf{x}^{t+1} - \mathbf{x}^t\|_2^2 + g(\mathbf{x}^{t+1}) - g(\mathbf{x}^t) - \frac{2}{\alpha^{t+1}}[\sqrt{d(\mathbf{x}^t)} - \sqrt{d(\mathbf{x}^{t+1})}],$$
$$(37)$$

where step ① uses the facts that the function $s(\mathbf{x})$ is $(L_f + \beta^t\|\mathbf{A}\|_2^2)$-smooth *w.r.t.* $\mathbf{x}$, $\alpha^{t+1} > 0$, $g(\mathbf{x})$ is convex, and $\sqrt{d(\mathbf{x})}$ is $W_d$-weakly convex, yielding the following inequalities:

$$s(\mathbf{x}^{t+1}) \leq s(\mathbf{x}^t) + \langle \mathbf{x}^{t+1} - \mathbf{x}^t, \nabla_\mathbf{x} s(\mathbf{x}^t)\rangle + \frac{L_f + \beta^t\|\mathbf{A}\|_2^2}{2}\|\mathbf{x}^{t+1} - \mathbf{x}^t\|_2^2$$

$$g(\mathbf{x}^t) \leq g(\mathbf{x}^{t+1}) + \langle \mathbf{x}^t - \mathbf{x}^{t+1}, \partial g(\mathbf{x}^t)\rangle$$

$$\frac{2}{\alpha^{t+1}}\{\sqrt{d(\mathbf{x}^t)} - \sqrt{d(\mathbf{x}^{t+1})}\} \leq \frac{2}{\alpha^{t+1}}\{\langle\partial\sqrt{d(\mathbf{x}^t)}, \mathbf{x}^t - \mathbf{x}^{t+1}\rangle + \frac{W_d}{2}\|\mathbf{x}^{t+1} - \mathbf{x}^t\|_2^2\}; \quad (38)$$

step ② uses the definition of $\ell(\beta^t) = L_f + \beta^t\|\mathbf{A}\|_2^2 + \frac{2}{\lambda^{t+1}}W_d$.

We further derive:

$$\mathcal{K}(\alpha^{t+1}, \mathbf{x}^{t+1}, \mathbf{y}^t; \mathbf{z}^t; \beta^t, \mu^t) - \mathcal{K}(\alpha^t, \mathbf{x}^t, \mathbf{y}^t; \mathbf{z}^t; \beta^t, \mu^t)$$

$$\overset{\textcircled{1}}{\leq} \mathcal{K}(\alpha^{t+1}, \mathbf{x}^{t+1}, \mathbf{y}^t; \mathbf{z}^t; \beta^t, \mu^t) - \mathcal{K}(\alpha^{t+1}, \mathbf{x}^t, \mathbf{y}^t; \mathbf{z}^t; \beta^t, \mu^t)$$

$$\overset{\textcircled{2}}{=} (\alpha^{t+1})^2\{s(\mathbf{x}^{t+1}) - s(\mathbf{x}^t) + \delta(\mathbf{x}^{t+1}) - \delta(\mathbf{x}^t) - g(\mathbf{x}^{t+1}) + g(\mathbf{x}^t) - \frac{2}{\alpha^{t+1}}[\sqrt{d(\mathbf{x}^{t+1})} - \sqrt{d(\mathbf{x}^t)}]\}$$

$$\overset{\textcircled{3}}{\leq} (\alpha^{t+1})^2 \cdot \frac{(1-\theta)}{2}\ell(\beta^t)\|\mathbf{x}^{t+1} - \mathbf{x}^t\|_2^2$$

$$\overset{\textcircled{4}}{\leq} \underbrace{\frac{1}{2}\underline{\alpha}^2\underline{\ell}(1 - \theta)}_{\triangleq -\varepsilon_x} \cdot \beta^t\|\mathbf{x}^{t+1} - \mathbf{x}^t\|_2^2,$$

where step ① uses the fact that $\alpha^{t+1} = \arg\min_\alpha \mathcal{K}(\alpha, \mathbf{x}^t, \mathbf{y}^t; \mathbf{z}^t; \beta^t, \mu^t)$, which implies the inequality $\mathcal{K}(\alpha^{t+1}, \mathbf{x}^t, \mathbf{y}^t; \mathbf{z}^t; \beta^t, \mu^t) \leq \mathcal{K}(\alpha^t, \mathbf{x}^t, \mathbf{y}^t; \mathbf{z}^t; \beta^t, \mu^t)$; step ② uses the definition of the function $\mathcal{K}(\alpha, \mathbf{x}, \mathbf{y}; \mathbf{z}; \beta, \mu)$; step ③ uses Inequality (37); step ④ uses $1 - \theta < 0$, $\alpha^t \geq \underline{\alpha}$, and $\ell(\beta^t) \geq \beta^t\underline{\ell}$. $\qquad\square$

### E.6 PROOF OF LEMMA 5.16

*Proof.* We define $\mathcal{K}(\alpha, \mathbf{x}, \mathbf{y}; \mathbf{z}; \beta, \mu) = -2\alpha\sqrt{d(\mathbf{x})} + \alpha^2\{f(\mathbf{x}) + \delta(\mathbf{x}) - g(\mathbf{x}) + h_\mu(\mathbf{y}) + \langle\mathbf{A}\mathbf{x} - \mathbf{y}, \mathbf{z}\rangle + \frac{\beta}{2}\|\mathbf{A}\mathbf{x} - \mathbf{y}\|_2^2)\}$.

We define $\mathbb{T}^t \triangleq 12\overline{\alpha}^2(1 + \xi)C_h^2/(\beta^0 t)$, and $\mathbb{U}^t \triangleq \frac{1}{2}\overline{\alpha}^2 C_h^2\mu^t$.

We define $\varepsilon_z \triangleq \frac{1}{2}\underline{\alpha}^2\xi$, and $\varepsilon_y \triangleq \frac{1}{2}\underline{\alpha}^2\{1 - 4(1 + \xi)/(\chi^2)\}$.

First, we focus on the sufficient decrease for variables $\mathbf{y}^{t+1}$, we have:

$$\mathcal{K}(\alpha^{t+1}, \mathbf{x}^{t+1}, \mathbf{y}^{t+1}; \mathbf{z}^t; \beta^t, \mu^t) - \mathcal{K}(\alpha^{t+1}, \mathbf{x}^{t+1}, \mathbf{y}^t; \mathbf{z}^t; \beta^t, \mu^t)$$

$$\overset{\textcircled{1}}{=} (\alpha^{t+1})^2\{\frac{\beta^t}{2}\|\mathbf{A}\mathbf{x}^{t+1} - \mathbf{y}^{t+1}\|_2^2 - \frac{\beta^t}{2}\|\mathbf{A}\mathbf{x}^{t+1} - \mathbf{y}^t\|_2^2 + h_{\mu^t}(\mathbf{y}^{t+1}) - h_{\mu^t}(\mathbf{y}^t) + \langle\mathbf{z}^t, \mathbf{y}^t - \mathbf{y}^{t+1}\rangle\}$$

$$\overset{\textcircled{2}}{=} (\alpha^{t+1})^2\{\frac{\beta^t}{2}\|\mathbf{A}\mathbf{x}^{t+1} - \mathbf{y}^{t+1}\|_2^2 - \frac{\beta^t}{2}\|\mathbf{A}\mathbf{x}^{t+1} - \mathbf{y}^t\|_2^2 + \langle\mathbf{y}^{t+1} - \mathbf{y}^t, \nabla h_{\mu^t}(\mathbf{y}^{t+1}) - \mathbf{z}^t\rangle\}$$

$$\overset{\textcircled{3}}{=} (\alpha^{t+1})^2\{-\frac{\beta^t}{2}\|\mathbf{y}^{t+1} - \mathbf{y}^t\|_2^2 + \langle\mathbf{y}^{t+1} - \mathbf{y}^t, \nabla h_{\mu^t}(\mathbf{y}^{t+1}) - \mathbf{z}^t - \beta^t(\mathbf{A}\mathbf{x}^{t+1} - \mathbf{y}^{t+1})\rangle\}$$

$$\overset{\textcircled{4}}{=} (\alpha^{t+1})^2\{-\frac{\beta^t}{2}\|\mathbf{y}^{t+1} - \mathbf{y}^t\|_2^2 + \langle\mathbf{y}^{t+1} - \mathbf{y}^t, \nabla h_{\mu^t}(\mathbf{y}^{t+1}) - \mathbf{z}^t - (\mathbf{z}^{t+1} - \mathbf{z}^t)\rangle\}$$

$$\overset{\textcircled{5}}{=} -\beta^t(\alpha^{t+1})^2\frac{1}{2}\|\mathbf{y}^{t+1} - \mathbf{y}^t\|_2^2, \quad (39)$$

where step ① uses the definition of $\mathcal{K}(\alpha, \mathbf{x}, \mathbf{y}; \mathbf{z}; \beta, \mu)$; step ② uses the convexity of $h_{\mu^t}(\cdot)$ that $h_\mu(\mathbf{y}') - h_\mu(\mathbf{y}) \leq \langle \mathbf{y}' - \mathbf{y}, \nabla h_\mu(\mathbf{y}') \rangle$ for all $\mathbf{y}, \mathbf{y}' \in \mathbb{R}^m$, and $\mu > 0$; step ③ uses the Pythagoras relation that $\frac{1}{2}\|\mathbf{a} - \mathbf{c}\|_2^2 - \frac{1}{2}\|\mathbf{b} - \mathbf{c}\|_2^2 = -\frac{1}{2}\|\mathbf{a} - \mathbf{b}\|_2^2 + \langle \mathbf{a} - \mathbf{c}, \mathbf{a} - \mathbf{b} \rangle$; step ④ uses the optimality for $\mathbf{y}^{t+1}$ that: $\nabla h_{\mu^t}(\mathbf{y}^{t+1}) = \mathbf{z}^t + \beta^t(\mathbf{A}\mathbf{x}^{t+1} - \mathbf{y}^{t+1})$.

Second, we focus on the sufficient decrease for variables $\{\mathbf{z}, \beta\}$. We have:

$$\frac{1}{2}\xi\underline{\alpha}^2\beta^t\|\mathbf{A}\mathbf{x}^{t+1} - \mathbf{y}^{t+1}\|_2^2 + \mathcal{K}(\alpha^{t+1}, \mathbf{x}^{t+1}, \mathbf{y}^{t+1}; \mathbf{z}^{t+1}; \beta^{t+1}, \mu^t) - \mathcal{K}(\alpha^{t+1}, \mathbf{x}^{t+1}, \mathbf{y}^{t+1}; \mathbf{z}^t; \beta^t, \mu^t)$$

$$\overset{①}{\leq} (\alpha^{t+1})^2 \tfrac{\xi\|\mathbf{z}^{t+1} - \mathbf{z}^t\|_2^2}{2\beta^t} + \{\mathcal{K}(\alpha^{t+1}, \mathbf{x}^{t+1}, \mathbf{y}^{t+1}; \mathbf{z}^{t+1}; \beta^t, \mu^t) - \mathcal{K}(\alpha^{t+1}, \mathbf{x}^{t+1}, \mathbf{y}^{t+1}; \mathbf{z}^t; \beta^t, \mu^t)\}$$

$$\quad + \{K(\alpha^{t+1}, \mathbf{x}^{t+1}; \mathbf{y}^{t+1}, \mathbf{z}^{t+1}; \beta^{t+1}, \mu^t) - K(\alpha^{t+1}, \mathbf{x}^{t+1}, \mathbf{y}^{t+1}, \mathbf{z}^{t+1}; \beta^t, \mu^t)\}$$

$$\overset{②}{=} (\alpha^{t+1})^2 \cdot \{\tfrac{\xi\|\mathbf{z}^{t+1} - \mathbf{z}^t\|_2^2}{2\beta^t} + \langle \mathbf{A}\mathbf{x}^{t+1} - \mathbf{y}^{t+1}, \mathbf{z}^{t+1} - \mathbf{z}^t \rangle + \tfrac{\beta^{t+1} - \beta^t}{2}\|\mathbf{A}\mathbf{x}^{t+1} - \mathbf{y}^{t+1}\|_2^2\}$$

$$\overset{③}{\leq} (\alpha^{t+1})^2 \cdot \{\tfrac{\xi}{2\beta^t} + \tfrac{1}{\beta^t} + \tfrac{\beta^t(1+\xi) - \beta^t}{2(\beta^t)^2}\} \cdot \|\mathbf{z}^{t+1} - \mathbf{z}^t\|_2^2$$

$$= (\alpha^{t+1})^2 \cdot \{(1 + \xi) \cdot \tfrac{1}{\beta^t}\} \cdot \|\mathbf{z}^{t+1} - \mathbf{z}^t\|_2^2$$

$$\overset{④}{\leq} (\alpha^{t+1})^2(1 + \xi)\tfrac{1}{\beta^t}\{\|\nabla h_{\mu^t}(\mathbf{y}^{t+1}) - \nabla h_{\mu^{t-1}}(\mathbf{y}^t)\|_2^2\}$$

$$\overset{⑤}{\leq} (\alpha^{t+1})^2(1 + \xi)\tfrac{1}{\beta^t} \cdot \{\tfrac{2(\beta^t)^2}{\chi^2}\|\mathbf{y}^{t+1} - \mathbf{y}^t\|_2^2 + 12C_h^2(\tfrac{1}{t} - \tfrac{1}{t+1})\}$$

$$\overset{⑥}{\leq} \{(\alpha^{t+1})^2(1 + \xi)\tfrac{2}{\chi^2}\beta^t\|\mathbf{y}^{t+1} - \mathbf{y}^t\|_2^2\} + \tfrac{12(1+\xi)C_h^2\overline{\alpha}^2}{\beta^0} \cdot (\tfrac{1}{t} - \tfrac{1}{t+1}), \tag{40}$$

where step ① uses $\underline{\alpha} \leq \alpha^t$; step ② uses the definition of $\mathcal{K}(\alpha, \mathbf{x}, \mathbf{y}; \mathbf{z}; \beta, \mu)$; step ③ uses $\beta^t(\mathbf{A}\mathbf{x}^{t+1} - \mathbf{y}^{t+1}) = \mathbf{z}^{t+1} - \mathbf{z}^t$ and $\beta^{t+1} \leq \beta^t(1 + \xi)$; step ④ uses Lemma 5.4(b); step ⑤ uses Lemma 5.4; step ⑥ uses $\alpha^t \leq \overline{\alpha}$ for all $t \geq 1$.

Third, we focus on the sufficient decrease for variable $\mu$. We have:

$$\mathcal{K}(\alpha^{t+1}, \mathbf{x}^{t+1}, \mathbf{y}^{t+1}; \mathbf{z}^{t+1}; \beta^{t+1}, \mu^{t+1}) - \mathcal{K}(\alpha^{t+1}, \mathbf{x}^{t+1}, \mathbf{y}^{t+1}; \mathbf{z}^{t+1}; \beta^{t+1}, \mu^t)$$

$$= (\alpha^{t+1})^2 \cdot (h_{\mu^{t+1}}(\mathbf{y}^{t+1}) - h_{\mu^t}(\mathbf{y}^{t+1}))$$

$$\overset{①}{\leq} \tfrac{1}{2}C_h^2(\alpha^{t+1})^2 \cdot (\mu^t - \mu^{t+1})$$

$$\overset{②}{\leq} \tfrac{1}{2}C_h^2\overline{\alpha}^2(\mu^t - \mu^{t+1}) = \mathbb{U}^t - \mathbb{U}^{t+1} \tag{41}$$

where step ① uses Lemma 3.9(e); step ② uses $\alpha^t \leq \overline{\alpha}$ for all $t \geq 1$.

Adding Inequalities (39), (40), and (41), we have:

$$\mathcal{K}(\lambda^{t+1}, \mathbf{x}^{t+1}, \mathbf{y}^{t+1}; \mathbf{z}^{t+1}; \beta^{t+1}, \mu^{t+1}) - \mathcal{K}(\lambda^{t+1}, \mathbf{x}^{t+1}, \mathbf{y}^t; \mathbf{z}^t; \beta^t, \mu^t)$$

$$\leq \mathbb{T}^t + \mathbb{U}^t - \mathbb{T}^{t+1} - \mathbb{U}^{t+1} - (\alpha^{t+1})^2\beta^t\|\mathbf{y}^{t+1} - \mathbf{y}^t\|_2^2 \cdot \tfrac{1}{2} \cdot \{1 - 4(1 + \xi)/(\chi^2)\}$$

$$\overset{①}{\leq} \mathbb{T}^t + \mathbb{U}^t - \mathbb{T}^{t+1} - \mathbb{U}^{t+1} - \beta^t\|\mathbf{y}^{t+1} - \mathbf{y}^t\|_2^2 \cdot \varepsilon_y,$$

where step ① uses $\alpha^{t+1} \geq \underline{\alpha}$, and the definition of $\varepsilon_y \triangleq \tfrac{1}{2}\underline{\alpha}^2\{1 - 4(1 + \xi)/(\chi^2)\}$.

$\square$

### E.7 PROOF OF LEMMA 5.17

*Proof.* We define $\mathbb{P}^t \triangleq \mathbb{K}^t + \mathbb{T}^t + \mathbb{U}^t$.

We define $\mathbb{K}^t \triangleq \mathcal{K}(\alpha^t, \mathbf{x}^t, \mathbf{y}^t; \mathbf{z}^t; \beta^t, \mu^t)$, and $\mathbb{T}^t = 12\overline{\alpha}^2(1 + \xi)C_h^2/(\beta^0\underline{d}t)$, and $\mathbb{U}^t = \tfrac{1}{2}C_h^2\overline{\alpha}^2\mu^t$.

**Part (a)**. We derive the following inequalities:

$$\mathbb{P}^t \triangleq \mathbb{K}^t + \mathbb{T}^t + \mathbb{U}^t$$

$$\overset{①}{\geq} \mathcal{K}(\alpha^t, \mathbf{x}^t, \mathbf{y}^t; \mathbf{z}^t; \beta^t, \mu^t) + 0$$

$$\overset{②}{=} -2\alpha^t\sqrt{d(\mathbf{x}^t)} + (\alpha^t)^2 \mathcal{U}(\mathbf{x}^t, \mathbf{y}^t; \mathbf{z}^t; \beta^t, \mu^t)$$

$$\overset{③}{\geq} -2\overline{\alpha}\sqrt{\overline{d}} + \underline{\alpha}^2 \cdot \{\underline{Fd} - \underline{v}/\beta^t\}$$

$$\overset{④}{\geq} -2\overline{\alpha}\sqrt{\overline{d}} + \underline{\alpha}^2 \cdot \{\underline{Fd} - \underline{v}/\beta^0\} \triangleq \underline{\mathbb{P}},$$

where step ① uses $\mathbb{T}^t \geq 0$ and $\mathbb{U}^t \geq 0$; step ② uses the definition of $\mathcal{K}(\alpha^t, \mathbf{x}^t, \mathbf{y}^t; \mathbf{z}^t; \beta^t, \mu^t)$; step ③ uses $d(\mathbf{x}^t) \leq \overline{d}$, $\alpha^t \leq \overline{\alpha}$, and the lower bound of $\mathcal{U}(\mathbf{x}^t, \mathbf{y}^t; \mathbf{z}^t; \beta^t, \mu^t)$ in Lemma 5.6; step ④ uses $\beta^t \geq \beta^0$.

**Part (b)**. Combing Lemmas (5.15) and (5.16) together, we have:

$$\varepsilon_y \beta^t \|\mathbf{y}^{t+1} - \mathbf{y}^t\|_2^2 + \varepsilon_x \beta^t \|\mathbf{x}^{t+1} - \mathbf{x}^t\|_2^2 + \varepsilon_z \beta^t \|\mathbf{A}\mathbf{x}^{t+1} - \mathbf{y}^{t+1}\|_2^2$$
$$\leq \quad \mathbb{K}^t - \mathbb{K}^t + \mathbb{T}^t - \mathbb{T}^{t+1} + \mathbb{U}^t - \mathbb{U}^{t+1}$$
$$= \quad \mathbb{P}^t - \mathbb{P}^{t+1}.$$

$\square$

### E.8    PROOF OF THEOREM 5.18

*Proof.* We define $c_0 \triangleq \frac{1}{\min(\varepsilon_x, \varepsilon_y, \varepsilon_z)} \cdot \{\mathbb{P}^1 - \underline{\mathbb{P}}\}$.

We define $\mathcal{E}^t \triangleq \beta^t \{\|\mathbf{x}^{t+1} - \mathbf{x}^t\|_2^2 + \|\mathbf{y}^{t+1} - \mathbf{y}^t\|_2^2 + \|\mathbf{A}\mathbf{x}^{t+1} - \mathbf{y}^{t+1}\|_2^2\}$.

We define $\mathcal{E}_+^t \triangleq \beta^t \{\|\mathbf{x}^{t+1} - \mathbf{x}^t\| + \|\mathbf{y}^{t+1} - \mathbf{y}^t\| + \|\mathbf{A}\mathbf{x}^{t+1} - \mathbf{y}^{t+1}\|\}$.

Using Lemma 5.17, we have:

$$0 \leq -\min(\varepsilon_x, \varepsilon_y, \varepsilon_z)\mathcal{E}^t + \mathbb{P}^t - \mathbb{P}^{t+1}. \tag{42}$$

Telescoping Inequality (42) over $t$ from 1 to $T$, we have:

$$0 \leq \frac{1}{\min(\varepsilon_x, \varepsilon_y, \varepsilon_z)} \cdot \sum_{t=1}^T \{\mathbb{P}^t - \mathbb{P}^{t+1}\} - \sum_{t=1}^T \mathcal{E}^t$$

$$= \frac{1}{\min(\varepsilon_x, \varepsilon_y, \varepsilon_z)} \cdot \{\mathbb{P}^1 - \mathbb{P}^{T+1}\} - \sum_{t=1}^T \mathcal{E}^t$$

$$\overset{①}{\leq} \frac{1}{\min(\varepsilon_x, \varepsilon_y, \varepsilon_z)} \cdot \{\mathbb{P}^1 - \underline{\mathbb{P}}\} - \sum_{t=1}^T \mathcal{E}^t$$

$$\overset{②}{\leq} c_0 - \frac{1}{\beta^T} \sum_{t=1}^T \|\beta^t(\mathbf{x}^{t+1} - \mathbf{x}^t)\|_2^2 + \|\beta^t(\mathbf{y}^{t+1} - \mathbf{y}^t)\|_2^2 + \|\beta^t(\mathbf{A}\mathbf{x}^{t+1} - \mathbf{y}^{t+1})\|_2^2$$

$$\overset{③}{\leq} c_0 - \frac{1}{\beta^T}\frac{1}{3T}\{\sum_{t=1}^T \|\beta^t(\mathbf{x}^{t+1} - \mathbf{x}^t)\| + \|\beta^t(\mathbf{y}^{t+1} - \mathbf{y}^t)\| + \|\beta^t(\mathbf{A}\mathbf{x}^{t+1} - \mathbf{y}^{t+1})\|\}^2$$

$$\overset{④}{\leq} c_0 - \frac{1}{\beta^T 3T}\{\sum_{t=1}^T \mathcal{E}_+^t\}^2 \tag{43}$$

where step ① uses $\mathbb{P}^t \geq \underline{\mathbb{P}}$ for all $t$; step ② uses the definition of $c_0$, the definition of $\mathcal{E}^t$, and the Hölder's inequality that $\langle \mathbf{a}, \mathbf{b} \rangle \leq \|\mathbf{a}\|_1 \|\mathbf{b}\|_\infty$; step ③ uses the fact that $\|\mathbf{a}\|_2^2 \geq \frac{1}{n}\|\mathbf{a}\|_1^2$; step ④ uses the definition of $\mathcal{E}_+^t$. We further obtain from Inequality (43) that

$$\sum_{t=1}^T \mathcal{E}_+^t \leq (3W)^{1/2}(\beta^T T)^{1/2} \quad \overset{①}{\Rightarrow} \quad \frac{1}{T}\sum_{t=1}^T \mathcal{E}_+^t \leq \mathcal{O}(T^{(p-1)/2}),$$

where step ① $\beta^t = \beta^0(1 + \xi t^p) = \mathcal{O}(t^p)$.

$\square$

# F    PROOFS FOR SECTION 6

## F.1    PROOF OF THEOREM 6.3

*Proof.* We let $t \geq 1$.

We define $\mathrm{Crit}(\mathbf{x}^+, \mathbf{x}, \mathbf{y}^+, \mathbf{y}, \mathbf{z}^+, \mathbf{z}) \triangleq \|\mathbf{x}^+ - \mathbf{x}\| + \|\mathbf{y}^+ - \mathbf{y}\| + \|\mathbf{z}^+ - \mathbf{z}\| + \|\mathbf{A}\mathbf{x}^+ - \mathbf{y}^+\| + \|\partial h(\mathbf{y}^+) - \mathbf{z}^+\|_2^2 + \|\partial\delta(\mathbf{x}^+) + \nabla f(\mathbf{x}^+) - \partial g(\mathbf{x}) + \mathbf{A}^{\mathsf{T}}\mathbf{z}^+ - \varphi(\mathbf{x}, \mathbf{y})\partial d(\mathbf{x})\|$, where $\varphi(\mathbf{x}, \mathbf{y}) = \{f(\mathbf{x}) + \delta(\mathbf{x}) - g(\mathbf{x}) + h(\mathbf{y})\}/d(\mathbf{x})$.

We define $\Gamma_1^t \triangleq \|\partial\delta(\mathbf{x}^{t+1}) + \nabla f(\mathbf{x}^{t+1}) + \mathbf{A}^{\mathsf{T}}\mathbf{z}^{t+1} - \partial g(\mathbf{x}^t) - \varphi(\mathbf{x}^t, \mathbf{y}^t)\partial d(\mathbf{x}^t)\|$.

We define $\Gamma_2^t \triangleq \|\mathbf{z}^{t+1} - \mathbf{z}^t\| + \|\partial h(\breve{\mathbf{y}}^{t+1}) - \mathbf{z}^{t+1}\| + \|\mathbf{A}\mathbf{x}^{t+1} - \mathbf{y}^{t+1}\|$.

We define $\Gamma_3^t \triangleq \|\mathbf{y}^{t+1} - \breve{\mathbf{y}}^{t+1}\| + \|\mathbf{x}^{t+1} - \mathbf{x}^t\| + \|\breve{\mathbf{y}}^{t+1} - \mathbf{y}^t\|$.

**Part (a)**. We now focus on FADMM-D.

We define $\lambda^t = \{f(\mathbf{x}^t) + \delta(\mathbf{x}^t) - g(\mathbf{x}^t) + \langle \mathbf{A}\mathbf{x}^t - \mathbf{y}^t, \mathbf{z}^t \rangle + \frac{1}{2}\beta^t\|\mathbf{A}\mathbf{x}^t - \mathbf{y}^t\|_2^2 + h_{\mu^t}(\mathbf{y}^t)\}/d(\mathbf{x}^t)$.

First, we focus on the optimality condition of the $\mathbf{x}$-subproblem. We have:

$$
\begin{aligned}
&- \partial\delta(\mathbf{x}^{t+1}) + \partial g(\mathbf{x}^t) + \lambda^t \partial d(\mathbf{x}^t) \\
&\ni \theta\ell(\beta^t)(\mathbf{x}^{t+1} - \mathbf{x}^t) + \nabla_{\mathbf{x}}\mathcal{S}^t(\mathbf{x}^t, \mathbf{y}^t; \mathbf{z}^t; \beta^t) \\
&= \theta\ell(\beta^t)(\mathbf{x}^{t+1} - \mathbf{x}^t) + \nabla f(\mathbf{x}^t) + \mathbf{A}^{\mathsf{T}}\mathbf{z}^t + \beta^t \mathbf{A}^{\mathsf{T}}(\mathbf{A}\mathbf{x}^t - \mathbf{y}^t).
\end{aligned}
\tag{44}
$$

Second, we derive the following inequalities:

$$
\begin{aligned}
&\|\partial d(\mathbf{x}^t) \cdot \{\lambda^t - \varphi(\mathbf{x}^t, \mathbf{y}^t)\}\| \\
&\overset{①}{\leq} C_d \cdot \left| \tfrac{f(\mathbf{x}^t) - g(\mathbf{x}^t) + \langle \mathbf{A}\mathbf{x}^t - \mathbf{y}^t, \mathbf{z}^t \rangle + \frac{1}{2}\beta^t\|\mathbf{A}\mathbf{x}^t - \mathbf{y}^t\|_2^2 + h_{\mu^t}(\mathbf{y}^t)}{d(\mathbf{x}^t)} - \tfrac{f(\mathbf{x}^t) - g(\mathbf{x}^t) + h(\mathbf{y}^t)}{d(\mathbf{x}^t)} \right| \\
&\overset{②}{\leq} \tfrac{C_d}{\underline{d}} \cdot \left| \langle \mathbf{A}\mathbf{x}^t - \mathbf{y}^t, \mathbf{z}^t \rangle + \tfrac{1}{2}\beta^t\|\mathbf{A}\mathbf{x}^t - \mathbf{y}^t\|_2^2 + h_{\mu^t}(\mathbf{y}^t) - h(\mathbf{y}^t) \right| \\
&\overset{③}{\leq} \tfrac{C_d}{\underline{d}} \cdot \left\{ \tfrac{\overline{z}}{\beta^{t-1}}\|\mathbf{z}^t - \mathbf{z}^{t-1}\| + \tfrac{\beta^t}{2(\beta^{t-1})^2}\|\mathbf{z}^t - \mathbf{z}^{t-1}\|_2^2 + \tfrac{1}{2}C_h^2\mu^t \right\} \\
&\overset{④}{\leq} \tfrac{C_d}{\underline{d}} \cdot \left\{ \tfrac{2\overline{z}^2}{\beta^{t-1}} + \tfrac{\beta^t}{2(\beta^{t-1})^2}(2\overline{z})^2 + \tfrac{1}{2\beta^t}C_h^2\chi \right\} \\
&\overset{⑤}{=} \mathcal{O}(\tfrac{1}{\beta^t}),
\end{aligned}
\tag{45}
$$

where step ① uses the fact that $d(\mathbf{x})$ is $C_d$-Lipschitz continuous, and the definitions of $\lambda^t$ and $\varphi(\mathbf{x}^t, \mathbf{y}^t)$; step ② uses $d(\mathbf{x}^t) \geq \underline{d}$; step ③ uses $0 < h(\mathbf{y}) - h_\mu(\mathbf{y}) \leq \frac{\mu}{2}C_h^2$ (refer to Lemma 3.9(b)), the Cauchy-Schwarz Inequality, and the fact that $\mathbf{A}\mathbf{x}^t - \mathbf{y}^t = \frac{1}{\beta^{t-1}}(\mathbf{z}^t - \mathbf{z}^{t-1})$; step ④ uses $\|\mathbf{z}^t - \mathbf{z}^{t-1}\| \leq \|\mathbf{z}^t\| + \|\mathbf{z}^{t-1}\| \leq 2\overline{z}$; step ⑤ uses $\beta^{t-1} \leq \beta^t \leq (1 + \xi)\beta^{t-1}$.

Third, we derive the following results:

$$
\begin{aligned}
\Gamma_1^t &\triangleq \|\partial\delta(\mathbf{x}^{t+1}) + \nabla f(\mathbf{x}^{t+1}) + \mathbf{A}^{\mathsf{T}}\mathbf{z}^{t+1} - \partial g(\mathbf{x}^t) - \varphi(\mathbf{x}^t, \mathbf{y}^t)\partial d(\mathbf{x}^t)\| \\
&\overset{①}{\leq} \|\lambda^t\partial d(\mathbf{x}^t) - \varphi(\mathbf{x}^t, \mathbf{y}^t)\partial d(\mathbf{x}^t) + \nabla f(\mathbf{x}^{t+1}) - \nabla f(\mathbf{x}^t) \\
&\qquad + \mathbf{A}^{\mathsf{T}}(\mathbf{z}^{t+1} - \mathbf{z}^t) - \{\theta\ell(\beta^t)(\mathbf{x}^{t+1} - \mathbf{x}^t) + \beta^t\mathbf{A}^{\mathsf{T}}(\mathbf{A}\mathbf{x}^t - \mathbf{y}^t)\}\| \\
&\overset{②}{\leq} \|\lambda^t\partial d(\mathbf{x}^t) - \varphi(\mathbf{x}^t, \mathbf{y}^t)\partial d(\mathbf{x}^t)\| + \|\mathbf{A}^{\mathsf{T}}(\mathbf{z}^{t+1} - \mathbf{z}^t)\| + \|\nabla f(\mathbf{x}^{t+1}) - \nabla f(\mathbf{x}^t)\| \\
&\qquad + \theta\ell(\beta^t)\|\mathbf{x}^{t+1} - \mathbf{x}^t\| + \beta^t\|\mathbf{A}^{\mathsf{T}}(\mathbf{A}\mathbf{x}^t - \mathbf{y}^t)\| \\
&\overset{③}{\leq} \mathcal{O}(\tfrac{1}{\beta^t}) + \mathcal{O}(\beta^t\|\mathbf{A}\mathbf{x}^{t+1} - \mathbf{y}^{t+1}\|) + \mathcal{O}(\beta^t\|\mathbf{x}^{t+1} - \mathbf{x}^t\|) + \mathcal{O}(\beta^{t-1}\|\mathbf{A}\mathbf{x}^t - \mathbf{y}^t\|),
\end{aligned}
\tag{46}
$$

where step ① uses Equalities (44); step ② uses the triangle inequality; step ③ uses Inequality (45), $\beta^{t-1} \leq \beta^t \leq (1+\xi)\beta^{t-1}$, and $\mathbf{z}^{t+1} - \mathbf{z}^t = \beta^t(\mathbf{A}\mathbf{x}^{t+1} - \mathbf{y}^{t+1})$.

Fourth, we have the following inequalities:

$$\Gamma_2^t \triangleq \|\mathbf{z}^{t+1} - \mathbf{z}^t\| + \|\partial h(\breve{\mathbf{y}}^{t+1}) - \mathbf{z}^{t+1}\| + \|\mathbf{A}\mathbf{x}^{t+1} - \mathbf{y}^{t+1}\|$$

$$\overset{①}{\leq} \mathcal{O}(\beta^t \|\mathbf{A}\mathbf{x}^{t+1} - \mathbf{y}^{t+1}\|), \tag{47}$$

where step ① uses $\mathbf{z}^{t+1} \in \partial h(\breve{\mathbf{y}}^{t+1})$ (refer to Lemma 5.3), and $\mathbf{z}^{t+1} - \mathbf{z}^t = \beta^t(\mathbf{A}\mathbf{x}^{t+1} - \mathbf{y}^{t+1})$.

Fifth, we have the following results:

$$\Gamma_3^t \triangleq \|\mathbf{y}^{t+1} - \breve{\mathbf{y}}^{t+1}\| + \|\mathbf{x}^{t+1} - \mathbf{x}^t\| + \|\breve{\mathbf{y}}^{t+1} - \mathbf{y}^t\|$$

$$\overset{①}{\leq} \|\mathbf{y}^{t+1} - \breve{\mathbf{y}}^{t+1}\| + \|\mathbf{x}^{t+1} - \mathbf{x}^t\| + \|\breve{\mathbf{y}}^{t+1} - \mathbf{y}^{t+1}\| + \|\mathbf{y}^{t+1} - \mathbf{y}^t\|$$

$$\overset{②}{\leq} \mu^t C_h + \|\mathbf{x}^{t+1} - \mathbf{x}^t\| + \mu^t C_h + \|\mathbf{y}^{t+1} - \mathbf{y}^t\|$$

$$\overset{③}{\leq} \mathcal{O}(\tfrac{1}{\beta^t}) + \mathcal{O}(\beta^t \|\mathbf{x}^{t+1} - \mathbf{x}^t\|) + \mathcal{O}(\beta^t \|\mathbf{y}^{t+1} - \mathbf{y}^t\|), \tag{48}$$

where step ① uses the triangle inequality; step ② uses Lemma (3.10)(c); step ③ uses $\mu^t = \frac{\chi}{\beta^t} = \mathcal{O}(\frac{1}{\beta^t})$, and $1 \leq \frac{\beta^t}{\beta^0} = \mathcal{O}(\beta^t)$.

**Part (b)**. We now focus on FADMM-Q.

We define $\mathcal{U}(\alpha^t, \mathbf{x}^t, \mathbf{y}^t; \mathbf{z}^t; \beta^t, \mu^t) \triangleq f(\mathbf{x}^t) + \delta(\mathbf{x}^t) - g(\mathbf{x}^t) + h_{\mu^t}(\mathbf{y}^t) + \langle \mathbf{A}\mathbf{x}^t - \mathbf{y}^t, \mathbf{z}^t \rangle + \frac{\beta^t}{2} \|\mathbf{A}\mathbf{x}^t - \mathbf{y}^t\|_2^2$, and $\alpha^{t+1} \triangleq \sqrt{d(\mathbf{x}^t)}/\mathcal{U}(\mathbf{x}^t, \mathbf{y}^t; \mathbf{z}^t; \beta^t, \mu^t)$.

First, we focus on the optimality condition of the $\mathbf{x}$-subproblem. We have:

$$\partial \delta(\mathbf{x}^{t+1}) - \partial g(\mathbf{x}^t) - \tfrac{2}{\alpha^{t+1}} \partial \sqrt{d(\mathbf{x}^t)}$$

$$\ni -\theta \ell(\beta^t)(\mathbf{x}^{t+1} - \mathbf{x}^t) - \nabla_{\mathbf{x}} \mathcal{S}(\mathbf{x}^t, \mathbf{y}^t; \mathbf{z}^t; \beta^t)$$

$$= -\theta \ell(\beta^t)(\mathbf{x}^{t+1} - \mathbf{x}^t) - \nabla f(\mathbf{x}^t) - \mathbf{A}^\top \mathbf{z}^t - \beta^t \mathbf{A}^\top (\mathbf{A}\mathbf{x}^t - \mathbf{y}^t). \tag{49}$$

Second, we derive the following inequalities:

$$\|\tfrac{2}{\alpha^{t+1}} \partial \sqrt{d(\mathbf{x}^t)} - \varphi(\mathbf{x}^t, \mathbf{y}^t) \partial d(\mathbf{x}^t)\|$$

$$\overset{①}{=} \|\tfrac{2}{\alpha^{t+1}} \tfrac{1}{2} d(\mathbf{x}^t)^{-1/2} \partial d(\mathbf{x}^t) - \varphi(\mathbf{x}^t, \mathbf{y}^t) \partial d(\mathbf{x}^t)\|$$

$$\overset{②}{=} \|\tfrac{\mathcal{U}(\mathbf{x}^t, \mathbf{y}^t; \mathbf{z}^t; \beta^t, \mu^t)}{d(\mathbf{x}^t)} \partial d(\mathbf{x}^t) - \varphi(\mathbf{x}^t, \mathbf{y}^t) \partial d(\mathbf{x}^t)\|$$

$$\overset{③}{\leq} C_d |\tfrac{\mathcal{U}(\mathbf{x}^t, \mathbf{y}^t; \mathbf{z}^t; \beta^t, \mu^t)}{d(\mathbf{x}^t)} - \varphi(\mathbf{x}^t, \mathbf{y}^t)|$$

$$\overset{④}{\leq} \tfrac{C_d}{\underline{d}} |\mathcal{U}(\mathbf{x}^t, \mathbf{y}^t; \mathbf{z}^t; \beta^t, \mu^t) - \{f(\mathbf{x}^t) + \delta(\mathbf{x}^t) - g(\mathbf{x}^t) + h(\mathbf{y}^t)\}|$$

$$\overset{⑤}{\leq} \tfrac{C_d}{\underline{d}} \cdot \{|\langle \mathbf{A}\mathbf{x}^t - \mathbf{y}^t, \mathbf{z}^t \rangle| + \tfrac{\beta^t}{2} \|\mathbf{A}\mathbf{x}^t - \mathbf{y}^t\|_2^2 + |h_{\mu^t}(\mathbf{y}^t) - h(\mathbf{y}^t)|\}$$

$$\overset{⑥}{\leq} \tfrac{C_d}{\underline{d}} \cdot \{\tfrac{\overline{z}}{\beta^{t-1}} \|\mathbf{z}^t - \mathbf{z}^{t-1}\| + \tfrac{\beta^t}{2(\beta^{t-1})^2} \|\mathbf{z}^t - \mathbf{z}^{t-1}\|_2^2 + \tfrac{\mu^t}{2} C_h^2\}$$

$$\overset{⑦}{\leq} \tfrac{C_d}{\underline{d}} \cdot \{\tfrac{2\overline{z}^2}{\beta^{t-1}} + \tfrac{\beta^t}{2(\beta^{t-1})^2}(2\overline{z})^2 + \tfrac{\chi}{2\beta^t} C_h^2\}$$

$$\overset{⑧}{=} \mathcal{O}(\tfrac{1}{\beta^t}), \tag{50}$$

where step ① uses $\partial \sqrt{d(\mathbf{x}^t)} = \frac{1}{2} d(\mathbf{x}^t)^{-1/2} \partial d(\mathbf{x}^t)$; step ② uses the fact that $\alpha^{t+1} = \sqrt{d(\mathbf{x}^t)}/\mathcal{U}(\mathbf{x}^t, \mathbf{y}^t; \mathbf{z}^t; \beta^t)$; step ③ uses the fact that $d(\mathbf{x})$ is $C_d$-Lipschitz continuous; step ④ uses $d(\mathbf{x}^t) \geq \underline{d}$; step ⑤ uses the definitions of $\mathcal{U}(\mathbf{x}^t, \mathbf{y}^t; \mathbf{z}^t; \beta^t, \mu^t)$; step ⑥ uses $0 < h(\mathbf{y}) - h_\mu(\mathbf{y}) \leq \frac{\mu}{2} C_h^2$ (refer to Lemma 3.9(b)), the Cauchy-Schwarz Inequality, and the fact that $\mathbf{A}\mathbf{x}^t - \mathbf{y}^t = \frac{1}{\beta^{t-1}}(\mathbf{z}^t - \mathbf{z}^{t-1})$; step ⑦ uses $\|\mathbf{z}^t - \mathbf{z}^{t-1}\| \leq \|\mathbf{z}^t\| + \|\mathbf{z}^{t-1}\| \leq 2\overline{z}$ and $\mu^t = \chi/\beta^t$; step ⑧ uses $\beta^{t-1} \leq \beta^t \leq (1+\xi)\beta^{t-1}$.

Third, we derive the following results:

$$\Gamma_1^t \triangleq \|\partial\delta(\mathbf{x}^{t+1}) + \nabla f(\mathbf{x}^{t+1}) + \mathbf{A}^\mathsf{T}\mathbf{z}^{t+1} - \partial g(\mathbf{x}^t) - \varphi(\mathbf{x}^t, \mathbf{y}^t)\partial d(\mathbf{x}^t)\|$$

$$\overset{①}{\le} \|\tfrac{2}{\alpha^{t+1}}\partial\sqrt{d(\mathbf{x}^t)} - \varphi(\mathbf{x}^t, \mathbf{y}^t)\partial d(\mathbf{x}^t) + \nabla f(\mathbf{x}^{t+1}) - \nabla f(\mathbf{x}^t)$$
$$\quad + \mathbf{A}^\mathsf{T}(\mathbf{z}^{t+1} - \mathbf{z}^t) - \{\theta\ell(\beta^t)(\mathbf{x}^{t+1} - \mathbf{x}^t) + \beta^t\mathbf{A}^\mathsf{T}(\mathbf{A}\mathbf{x}^t - \mathbf{y}^t)\}\|$$

$$\overset{②}{\le} \|\tfrac{2}{\alpha^{t+1}}\partial\sqrt{d(\mathbf{x}^t)} - \varphi(\mathbf{x}^t, \mathbf{y}^t)\partial d(\mathbf{x}^t)\| + \|\mathbf{A}^\mathsf{T}(\mathbf{z}^{t+1} - \mathbf{z}^t)\|$$
$$\quad + \|\nabla f(\mathbf{x}^{t+1}) - \nabla f(\mathbf{x}^t)\| + \theta\ell(\beta^t)\|\mathbf{x}^{t+1} - \mathbf{x}^t\| + \beta^t\|\mathbf{A}^\mathsf{T}(\mathbf{A}\mathbf{x}^t - \mathbf{y}^t)\|$$

$$\overset{③}{\le} \mathcal{O}(\tfrac{1}{\beta^t}) + \mathcal{O}(\beta^t\|\mathbf{A}\mathbf{x}^{t+1} - \mathbf{y}^{t+1}\|) + \mathcal{O}(\beta^t\|\mathbf{x}^{t+1} - \mathbf{x}^t\|) + \mathcal{O}(\beta^{t-1}\|\mathbf{A}\mathbf{x}^t - \mathbf{y}^t\|), \quad (51)$$

where step ① uses Equalities (49); step ② uses the triangle inequality; step ③ uses Inequality (50), $\beta^{t-1} \le \beta^t \le (1+\xi)\beta^{t-1}$, and $\mathbf{z}^{t+1} - \mathbf{z}^t = \beta^t(\mathbf{A}\mathbf{x}^{t+1} - \mathbf{y}^{t+1})$.

Fourth, we have the following inequalities:

$$\Gamma_2^t \triangleq \|\mathbf{z}^{t+1} - \mathbf{z}^t\| + \|\partial h(\breve{\mathbf{y}}^{t+1}) - \mathbf{z}^{t+1}\| + \|\mathbf{A}\mathbf{x}^{t+1} - \mathbf{y}^{t+1}\|$$
$$\overset{①}{\le} \mathcal{O}(\beta^t\|\mathbf{A}\mathbf{x}^{t+1} - \mathbf{y}^{t+1}\|), \quad (52)$$

where step ① uses $\mathbf{z}^{t+1} \in \partial h(\breve{\mathbf{y}}^{t+1})$ (refer to Lemma 5.3), and $\mathbf{z}^{t+1} - \mathbf{z}^t = \beta^t(\mathbf{A}\mathbf{x}^{t+1} - \mathbf{y}^{t+1})$.

Fifth, we have the following results:

$$\Gamma_3^t \triangleq \|\mathbf{y}^{t+1} - \breve{\mathbf{y}}^{t+1}\| + \|\mathbf{x}^{t+1} - \mathbf{x}^t\| + \|\breve{\mathbf{y}}^{t+1} - \mathbf{y}^t\|$$

$$\overset{①}{\le} \|\mathbf{y}^{t+1} - \breve{\mathbf{y}}^{t+1}\| + \|\mathbf{x}^{t+1} - \mathbf{x}^t\| + \|\breve{\mathbf{y}}^{t+1} - \mathbf{y}^{t+1}\| + \|\mathbf{y}^{t+1} - \mathbf{y}^t\|$$

$$\overset{②}{\le} \mu^t C_h + \|\mathbf{x}^{t+1} - \mathbf{x}^t\| + \mu^t C_h + \|\mathbf{y}^{t+1} - \mathbf{y}^t\|$$

$$\overset{③}{\le} \mathcal{O}(\tfrac{1}{\beta^t}) + \mathcal{O}(\beta^t\|\mathbf{x}^{t+1} - \mathbf{x}^t\|) + \mathcal{O}(\beta^t\|\mathbf{y}^{t+1} - \mathbf{y}^t\|), \quad (53)$$

where step ① uses the triangle inequality; step ② uses Lemma (3.10)(c); step ③ uses $\mu^t = \frac{\chi}{\beta^t} = \mathcal{O}(\frac{1}{\beta^t})$, and $1 \le \frac{\beta^t}{\beta^0} = \mathcal{O}(\beta^t)$.

**Part (c)**. Finally, we continue our analysis for both FADMM-D and FADMM-Q, deriving the following inequalities:

$$\tfrac{1}{T}\sum_{t=1}^T \mathrm{Crit}(\mathbf{x}^{t+1}, \mathbf{x}^t, \breve{\mathbf{y}}^{t+1}, \mathbf{y}^t, \mathbf{z}^{t+1}, \mathbf{z}^t)$$

$$\overset{①}{\le} \tfrac{1}{T}\sum_{t=1}^T \{\Gamma_1^t + \Gamma_2^t + \Gamma_3^t\}$$

$$\overset{②}{\le} \tfrac{1}{T}\sum_{t=1}^T \{\mathcal{O}(\beta^t\|\mathbf{A}\mathbf{x}^{t+1} - \mathbf{y}^{t+1}\|) + \mathcal{O}(\beta^t\|\mathbf{x}^{t+1} - \mathbf{x}^t\|) + \mathcal{O}(\beta^{t-1}\|\mathbf{A}\mathbf{x}^t - \mathbf{y}^t\|)$$
$$\quad + \mathcal{O}(\beta^t\|\mathbf{x}^{t+1} - \mathbf{x}^t\|) + \mathcal{O}(\beta^t\|\mathbf{y}^{t+1} - \mathbf{y}^t\|) + \mathcal{O}(\tfrac{1}{\beta^t})\}$$

$$\overset{③}{=} \mathcal{O}(T^{(p-1)/2}) + \tfrac{1}{T}\sum_{t=1}^T \mathcal{O}(\tfrac{1}{\beta^t})$$

$$\overset{④}{=} \mathcal{O}(T^{(p-1)/2}) + \mathcal{O}(\tfrac{1}{T}T^{1-p}), \quad (54)$$

where step ① uses the definition of $\mathrm{Crit}(\mathbf{x}^+, \mathbf{x}, \mathbf{y}^+, \mathbf{y}, \mathbf{z}^+, \mathbf{z})$, and the triangle inequality that $\|\mathbf{A}\mathbf{x}^{t+1} - \breve{\mathbf{y}}^{t+1}\| \le \|\mathbf{A}\mathbf{x}^{t+1} - \mathbf{y}^{t+1}\| + \|\breve{\mathbf{y}}^{t+1} - \mathbf{y}^{t+1}\|$; step ② uses Inequalities (46), (47), and (48) for FADMM-D and Inequalities (51), (52), and (53) for FADMM-Q; step ③ uses Theorem 5.13 for FADMM-D and Theorem 5.18 for FADMM-Q that $\tfrac{1}{T}\sum_{t=1}^T \{\beta^t\|\mathbf{x}^{t+1} - \mathbf{x}^t\| + \beta^t\|\mathbf{y}^{t+1} - \mathbf{y}^t\| + \beta^t\|\mathbf{A}\mathbf{x}^{t+1} - \mathbf{y}^{t+1}\|\} \le \mathcal{O}(T^{(p-1)/2})$; step ④ uses $\sum_{t=1}^T \tfrac{1}{\beta^t} = \mathcal{O}(\sum_{t=1}^T \tfrac{1}{t^p}) = \mathcal{O}(T^{1-p})$, as presented in Lemma A.5.

We define $\mathcal{W}^t \triangleq \{\mathbf{x}^{t+1}, \mathbf{x}^t, \breve{\mathbf{y}}^{t+1}, \mathbf{y}^t, \mathbf{z}^{t+1}, \mathbf{z}^t\}$. With the choice $p = 1/3$, we have from Inequality (54) that $\tfrac{1}{T}\sum_{t=1}^T \mathrm{Crit}(\mathcal{W}^t) \le \mathcal{O}(T^{-1/3})$. In other words, there exists $1 \le \bar{t} \le T$ such that: $\mathrm{Crit}(\mathcal{W}^{\bar{t}}) \le \epsilon$, provided that $T \ge \mathcal{O}(\tfrac{1}{\epsilon^3})$

□

# G    COMPUTING PROXIMAL OPERATORS

In this section, we demonstrate how to compute the proximal operator for various functions involved in this paper. The proximal operator is defined as follows:

$$\min_{\mathbf{x}\in\mathbb{R}^r} p(\mathbf{x}) + \tfrac{1}{2\mu}\|\mathbf{x}-\mathbf{x}'\|_2^2. \tag{55}$$

Here, $\mathbf{x}' \in \mathbb{R}^r$ and $\mu > 0$ are given.

## G.1    ORTHOGONALITY CONSTRAINT

When $p(\mathbf{x}) = \iota_\Omega(\mathrm{mat}(\mathbf{x}))$ with $\Omega$ being the set of orthogonality constraints, Problem (55) simplifies to the following nonconvex optimization problem:

$$\bar{\mathbf{x}} \in \arg\min_{\mathbf{x}} \tfrac{1}{2\mu}\|\mathbf{x}-\mathbf{x}'\|_2^2,\ s.t.\ \mathrm{mat}(\mathbf{x}) \in \Omega \triangleq \{\mathbf{V}\,|\,\mathbf{V}^\mathsf{T}\mathbf{V} = \mathbf{I}\}.$$

This is the nearest orthogonality matrix problem, where the optimal solution is given by $\bar{\mathbf{x}} = \mathrm{vec}(\hat{\mathbf{U}}\hat{\mathbf{V}}^\mathsf{T})$ with $\mathrm{mat}(\mathbf{x}') = \hat{\mathbf{U}}\mathrm{Diag}(\mathbf{s})\hat{\mathbf{U}}^\mathsf{T}$ being the singular value decomposition of the matrix $\mathrm{mat}(\mathbf{x}')$. See (Lai & Osher, 2014) for reference.

## G.2    GENERALIZED $\ell_1$ NORM

When $p(\mathbf{x}) = \rho_2\|\mathbf{x}\|_1 + \iota_\Omega(\mathbf{x})$ with $\Omega \triangleq \{\mathbf{x}\,|\,\|\mathbf{x}\|_\infty \leq \rho_0\}$, Problem (55) simplifies to the following strongly convex problem:

$$\bar{\mathbf{x}} \in \arg\min_{\mathbf{x}\in\mathbb{R}^r} \rho_2\|\mathbf{x}\|_1 + \tfrac{1}{2\mu}\|\mathbf{x}-\mathbf{x}'\|_2^2,\ \mathrm{s.\,t.}\ \|\mathbf{x}\|_\infty \leq \rho_0.$$

This problem can be decomposed into $r$ dependent sub-problems

$$\bar{\mathbf{x}}_i = \arg\min_x q_i(x) \triangleq \tfrac{1}{2\mu}(x-\mathbf{x}_i')^2 + \rho_2|x|,\ \mathrm{s.\,t.}\ -\rho_0 \leq x \leq \rho_0. \tag{56}$$

We define $\mathcal{P}_{[l,u]}(x) \triangleq \max(l, \min(u, x))$. We consider five cases for $x$. (*i*) $x_1 = 0$. (*ii*) $x_2 = -\rho_0$. (*iii*) $x_3 = \rho_0$. (*iv*) $x_4 > 0$ and $x_4 < \rho_0$. By omitting the bound constraints, the first-order optimality condition gives $\tfrac{1}{\mu}(x_4 - \mathbf{x}_i') + \rho_2 = 0$, leading to $x_4 = \mathbf{x}_i' - \mu\rho_2$. When the bound constraints are included, we have $x_4 = \mathcal{P}_{[0,\rho_0]}(\mathbf{x}_i' - \mu\rho_2)$. (*v*) $x < 0$ and $x > -\rho_0$. By dropping the bound constraints, the first-order optimality condition yields $\tfrac{1}{\mu}(x_5 - \mathbf{x}_i') - \rho_2 = 0$, leading to $x_5 = \mathbf{x}_i' + \mu\rho_2$. When the bound constraints are considered, we have $x_5 = \mathcal{P}_{[-\rho_0,0]}(\mathbf{x}_i' + \mu\rho_2)$. Therefore, the one-dimensional sub-problem in Problem (56) contains five critical points, and the optimal solution can be computed as:

$$\bar{\mathbf{x}}_i = \arg\min_x q_i(x),\ \mathrm{s.\,t.}\ x \in \{x_1, x_2, x_3, x_4, x_5\}.$$

## G.3    SIMPLEX CONSTRAINT

When $p(\mathbf{x}) = \iota_\Omega(\mathbf{x})$ with $\Omega \triangleq \{\mathbf{x}\,|\,\mathbf{x} \geq \mathbf{0},\ \mathbf{x}^\mathsf{T}\mathbf{1} = 1\}$, Problem (55) simplifies to the following strongly convex problem:

$$\bar{\mathbf{x}} \in \arg\min_{\mathbf{x}} \tfrac{1}{2\mu}\|\mathbf{x}-\mathbf{x}'\|_2^2,\ \mathrm{s.\,t.}\ \mathbf{x} \geq \mathbf{0},\ \mathbf{x}^\mathsf{T}\mathbf{1} = 1.$$

This problem is referred to as the Euclidean projection onto the probability simplex. It can be solved exactly in $\mathcal{O}(n\log(n))$ time (Duchi et al., 2008).

## G.4    GENERALIZED MAX FUNCTION

When $p(\mathbf{x}) = \max(0, \max(\mathbf{x}+\mathbf{b}))$ with $\mathbf{b} \in \mathbb{R}^r$, Problem (55) simplifies to the following strongly convex problem: $\bar{\mathbf{x}} \in \arg\min_{\mathbf{x}} \tfrac{1}{2\mu}\|\mathbf{x}-\mathbf{x}'\|_2^2 + \max(0, \max(\mathbf{x}+\mathbf{b}))$. Using the variable substitution that $\mathbf{x} + \mathbf{b} = \mathbf{v}$, we have the following equivalent problem:

$$\bar{\mathbf{v}} \in \arg\min_{\mathbf{v}} q(\mathbf{v}) \triangleq \tfrac{1}{2\mu}\|\mathbf{v}-\mathbf{v}'\|_2^2 + \max(0, \max(\mathbf{v})), \tag{57}$$

where $\mathbf{v}' \triangleq \mathbf{x}' + \mathbf{b}$.

In what follows, we address Problem (57) by considering two cases for $\mathbf{v}'$. (*i*) $\max(\mathbf{v}') \leq 0$. The optimal solution can be computed as $\bar{\mathbf{v}} = \mathbf{v}'$, and it holds that $q(\bar{\mathbf{v}}) = 0$. (*ii*) $\max(\mathbf{v}') > 0$. In this case, there exists an index $i \in [r]$ such that $\mathbf{v}' > 0$. It is not difficult to verify that the optimal solution $\bar{\mathbf{v}}$ satisfies $\max(\bar{\mathbf{v}}) \geq 0$. Problem (57) reduces to:

$$\bar{\mathbf{v}} = \arg\min_{\mathbf{v}} \tfrac{1}{2\mu}\|\mathbf{v} - \mathbf{v}'\|_2^2 + \max(\mathbf{v}). \tag{58}$$

This problem can be equivalently reformulated as: $\min_{\mathbf{v},\tau} \tfrac{1}{2\mu}\|\mathbf{v} - \mathbf{v}'\|_2^2 + \tau$, s.t. $\mathbf{v} \leq \tau\mathbf{1}$, whose dual problem is given by:

$$\bar{\mathbf{z}} = \arg\max_{\mathbf{z}} -\tfrac{\mu}{2}\|\mathbf{z}\|_2^2 + \langle \mathbf{z}, \mathbf{v}'\rangle, \text{ s.t. } \mathbf{z} \geq 0, \ \|\mathbf{z}\|_1 = 1. \tag{59}$$

The unique optimal solution $\bar{\mathbf{z}}$ for the dual problem in Problem (59) can be computed in $\mathcal{O}(n\log(n))$ time (Duchi et al., 2008). Finally, the optimal solution $\bar{\mathbf{v}}$ for Problem (58) can then be recovered as $\bar{\mathbf{v}} = \mathbf{v}' - \mu\bar{\mathbf{z}}$.

## H  IMPLEMENTATION OF THE FULL SPLITTING ALGORITHM (FSA)

This section details the implementation of the Full Splitting Algorithm (FSA) (Boţ et al., 2023b) for solving Problem (1), as summarized in Algorithm 2. For simplicity, we set $\beta = 1$ and use a constant step size $\gamma^t = \gamma'$ for all $t$, where $\gamma' \in \{10^{-3}, 10^{-4}\}$.

---

**Algorithm 2:** FSA: Bot et al.'s Full Splitting Algorithm for Solving Problem (1).

---

**(S0)** Initialize $\{\mathbf{x}^0, \mathbf{z}^0, \mathbf{u}^0\}$.
**(S1)** Choose suitable $\beta \in (0, 2)$, $\{\gamma^t\}_{t=0}^T$.
**(S2)** Set $\{\alpha^t\} = \{1/\gamma^t\}$ for all $t$.
**for** $t$ *from 0 to T* **do**

    **(S3)** Let $g^t \in \nabla f(x^t) + \mathbf{A}^\mathsf{T}\mathbf{z}^t - \theta^t \partial d(\mathbf{x}^t)$.

    **(S4)** $\mathbf{x}^{t+1} \in \arg\min_{\mathbf{x}} \delta(\mathbf{x}) + \tfrac{\alpha^t}{2}\|\mathbf{x} - (\mathbf{u}^t - \mathbf{g}^t/\alpha^t)\|_2^2$.

    **(S5)** $\mathbf{u}^{t+1} = (1 - \beta)\mathbf{u}^t + \beta\mathbf{x}^{t+1}$.

    **(S6)** $\mathbf{z}^{t+1} = \mathrm{Prox}(\tfrac{\mathbf{A}\mathbf{x}^{t+1}}{\gamma^t}; h^*, \tfrac{1}{\gamma^t})$.

    **(S7)** $\theta^{t+1} = \tfrac{L(\mathbf{x}^t, \mathbf{z}^t, u^t, \alpha^t, \gamma^t)}{d(\mathbf{x}^t)}$, where

    $L(\mathbf{x}, \mathbf{z}, \mathbf{u}, \alpha, \gamma) \triangleq f(\mathbf{x}) + \langle \mathbf{z}, \mathbf{A}\mathbf{x}\rangle - h^*(\mathbf{z}) + \delta(\mathbf{x}) + \tfrac{\alpha}{2}\|\mathbf{x} - \mathbf{u}\|_2^2 - \tfrac{\gamma}{2}\|\mathbf{z}\|_2^2$.

**end**

---

## I  ADDITIONAL EXPERIMENTAL DETAILS AND RESULTS

In this section, we offer further experimental details on the datasets used in the experiments, and include additional results.

▶ **Datasets**. (*i*) For sparse FDA, robust SRM, and robust sparse recovery problems, we incorporate several datasets in our experiments, including randomly generated data and publicly available real-world data. These datasets serve as our data matrices $\mathbf{Q} \in \mathbb{R}^{\dot{m} \times \dot{d}}$ and the label vectors $\mathbf{p} \in \mathbb{R}^{\dot{m}}$. The dataset names are as follows: 'madelon-$\dot{m}$-$\dot{d}$', 'TDT2-1-2-$\dot{m}$-$\dot{d}$', 'TDT2-3-4-$\dot{m}$-$\dot{d}$', 'mnist-$\dot{m}$-$\dot{d}$', 'mushroom-$\dot{m}$-$\dot{d}$', 'gisette-$\dot{m}$-$\dot{d}$', and 'randn-$\dot{m}$-$\dot{d}$'. Here, '$\mathrm{randn}(\dot{m}, \dot{d})'$ represents a function that generates a standard Gaussian random matrix with dimensions $\dot{m} \times \dot{d}$, and 'TDT2-$i$-$j$' refers to the subset of the original dataset 'TDT2' consisting of data points with labels $i$ and $j$. The matrix $\mathbf{Q} \in \mathbb{R}^{\dot{m} \times \dot{d}}$ is constructed by randomly selecting $\dot{m}$ examples and $\dot{d}$ dimensions from the original real-world dataset (https://www.csie.ntu.edu.tw/~cjlin/libsvm/). We normalize each column of $\mathbf{D}$ to have a unit norm. As 'randn-$\dot{m}$-$\dot{d}$' does not have labels, we randomly and uniformly assign to binary labels with $\mathbf{p} \in \{-1, +1\}^{\dot{m}}$. (*ii*) For sparse FDA as in Problem (2), we let $\mathbf{D} \triangleq (\boldsymbol{\mu}_{(1)} - \boldsymbol{\mu}_{(2)})(\boldsymbol{\mu}_{(1)} - \boldsymbol{\mu}_{(2)})^\mathsf{T}$, $\mathbf{C} = \boldsymbol{\Sigma}_{(1)} + \boldsymbol{\Sigma}_{(2)}$, where $\boldsymbol{\mu}_{(i)} \in \mathbb{R}^n$ and $\boldsymbol{\Sigma}_{(i)} \in \mathbb{R}^{n \times n}$ represent the mean vector and covariance matrix of class $i$ ($i = 1$ or 2), respectively, generated by $\{\mathbf{Q}, \mathbf{p}\}$. We normalize the matrices $\mathbf{C}$ and $\mathbf{D}$ as $\mathbf{C} = \mathbf{C}/\|\mathbf{C}\|_\mathsf{F}$, and $\mathbf{D} = \mathbf{D}/\|\mathbf{D}\|_\mathsf{F}$. (*iii*) For

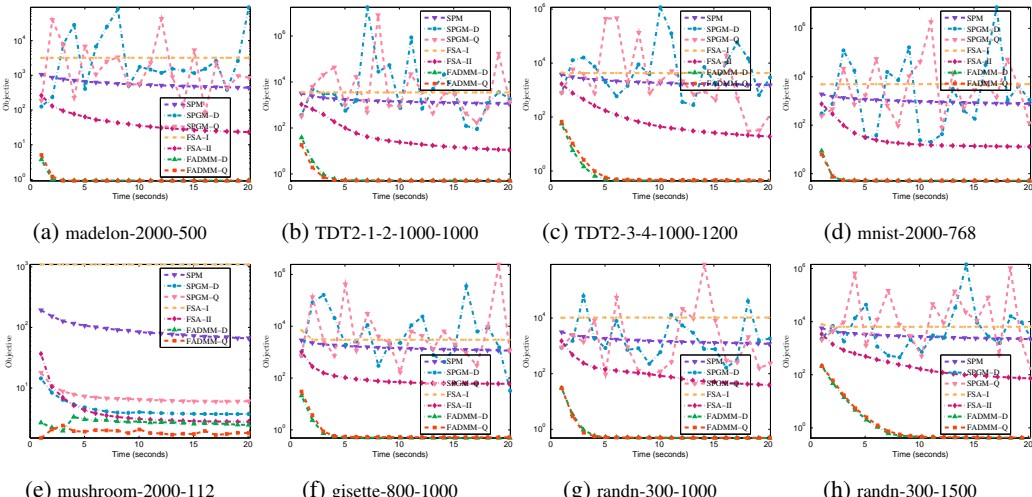

Figure 3: Experimental results on sparse FDA on different datasets with $\rho = 100$.

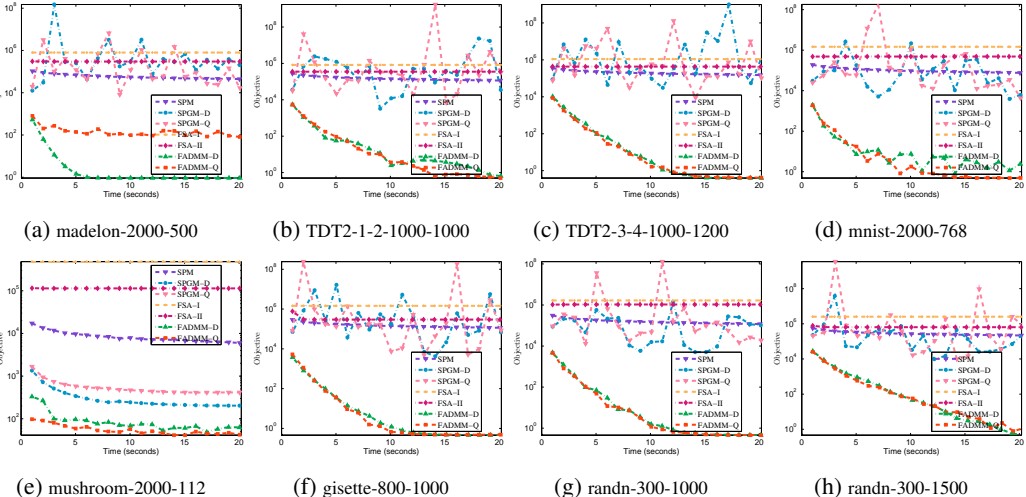

Figure 4: Experimental results on sparse FDA on different datasets with $\rho = 10000$.

robust SRM as in Problem (14), we let $\mathbf{D} = \mathbf{Q}$ and $\mathbf{b} = \mathbf{p}$. Following (Boţ et al., 2023b), we generate $p$ matrices $\{\mathbf{C}_{(i)}\}_{i=1}^{p}$, where each $\mathbf{C}_{(i)} \in \mathbb{R}^{n \times n}$ is constructed as $\mathbf{C}_{(i)} = \frac{1}{n}\mathbf{Y}\mathbf{Y}^{\mathsf{T}}$, with $\mathbf{Y} = \text{randn}(n, n) \times 10$. We let $p = 100$. (*iv*) For robust sparse recovery as in Problem (15), we simply let $\mathbf{A} = \mathbf{Q}$ and $\mathbf{b} = \mathbf{p}$, where $\mathbf{p} \in \{-1, +1\}^{\dot{m}}$ represents the data labels.

▶ **Additional Experimental Results on Sparse FDA**. We present additional experimental results on sparse FDA for $\rho \in \{100, 10000\}$ in Figures 3 and 4. These results reinforce our conclusions presented in the main paper.

▶ **Experimental Results on Robust SRM**. We consider solving Problem (14) using the proposed methods. For all methods, we set $\beta^0 = 0.001$. The results of the algorithms are shown in Figure 5. We draw the following conclusions. (*i*) SPM appears to outperform both SPGM-D and SPGM-Q. We attribute this to the fact that the subgradient provides a good approximation of the negative descent direction. (*ii*) Both variants, FADMM-D and FADMM-Q, generally demonstrate better performance than the other methods, achieving lower objective function values.

▶ **Experimental Results on Robust Sparse Recovery**. We consider solving Problem (15) using the following parameters $(\rho_1, \rho_2, \rho_3) \in \{(10, 1, \infty), (10, 10, \infty), (10, 100, \infty), (100, 1, \infty),$

$(100, 100, \infty)\}$. For all methods, we initialize with $\beta^0 = 0.001$. Since $\sqrt{\|\mathbf{x}\|_{[k]}}$ is not necessarily weakly convex, FADMM-Q is not applicable; thus, we only implement FADMM-D. The results of the compared methods are presented in Figures 6, 7, 8, 9, 10, from which we draw the following conclusions: Although SPM, SPGM-D, and FSA yield comparable or better results than FADMM-D in certain cases, the proposed FADMM-D generally achieves the best performance among the compared methods in terms of speed. These results further corroborate our earlier findings that the proposed method is faster and more numerically robust.

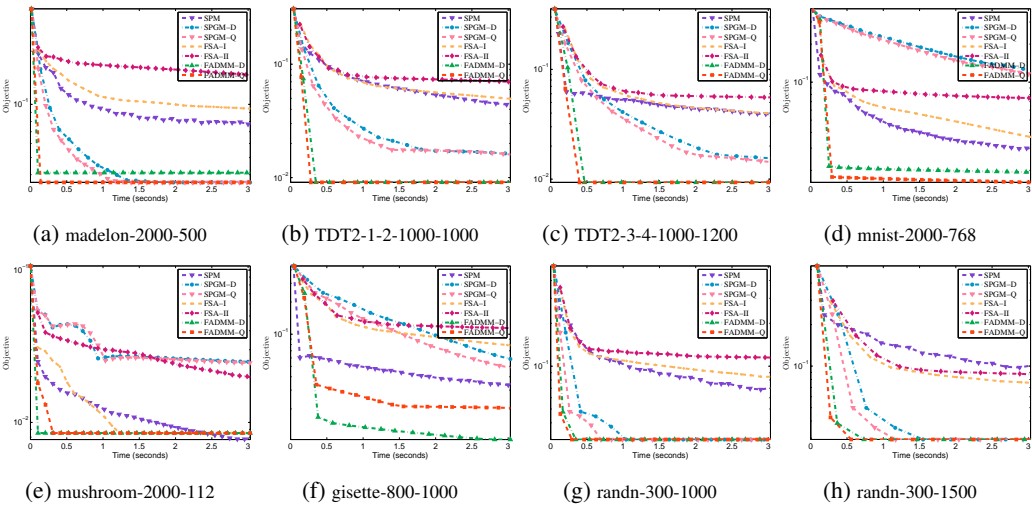

Figure 5: Results on Sharpe ratio maximization on different datasets.

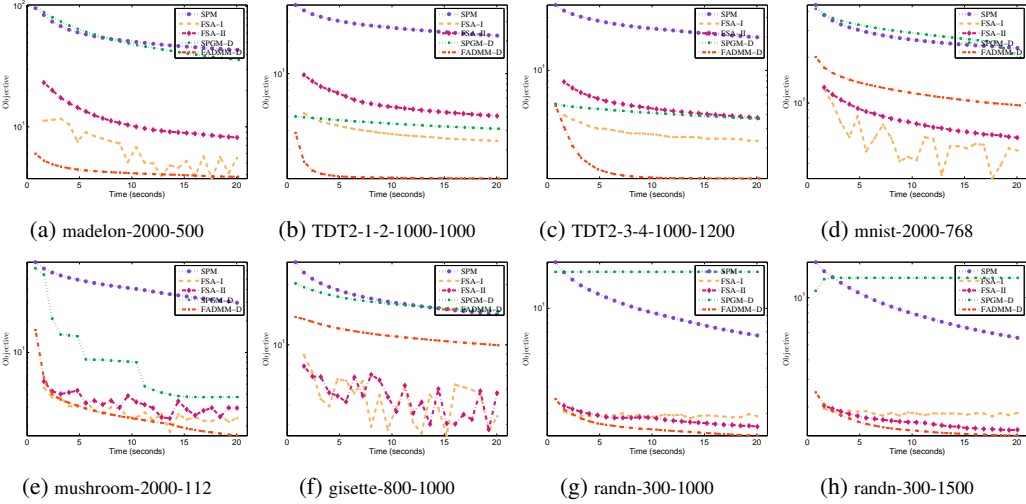

Figure 6: Results on robust sparse recovery on different datasets with $(\rho_1, \rho_2, \rho_0) = (10, 1, \infty)$.

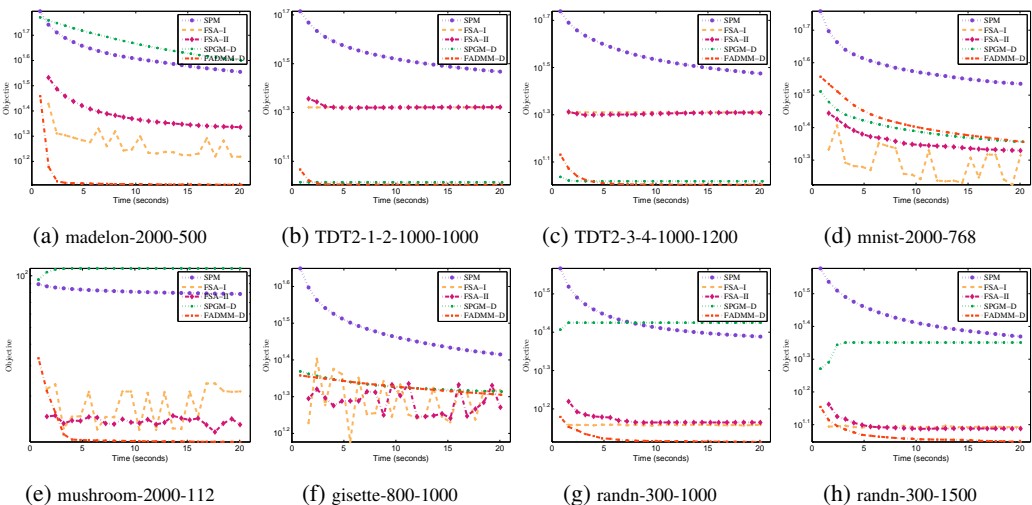

Figure 7: Results on robust sparse recovery on different datasets with $(\rho_1, \rho_2, \rho_0) = (10, 10, \infty)$.

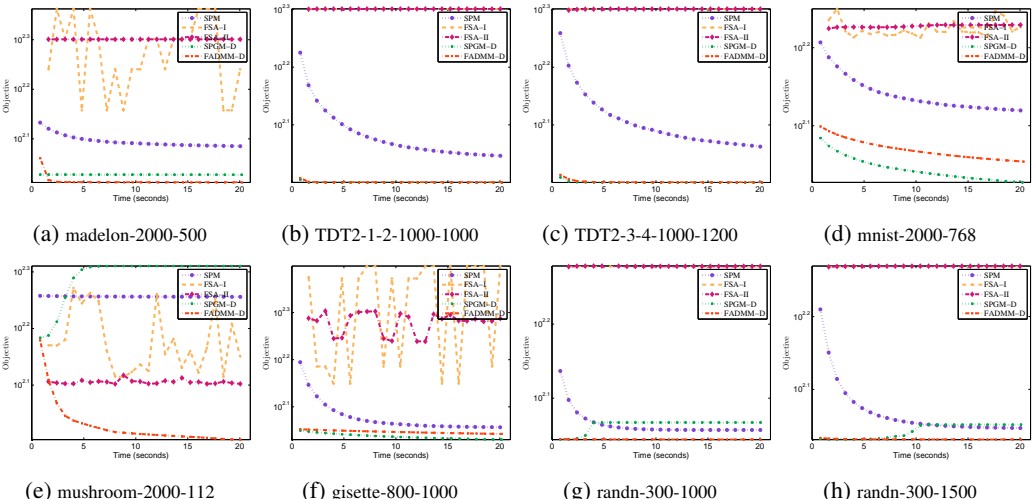

Figure 8: Results on robust sparse recovery on different datasets with $(\rho_1, \rho_2, \rho_0) = (10, 100, \infty)$.

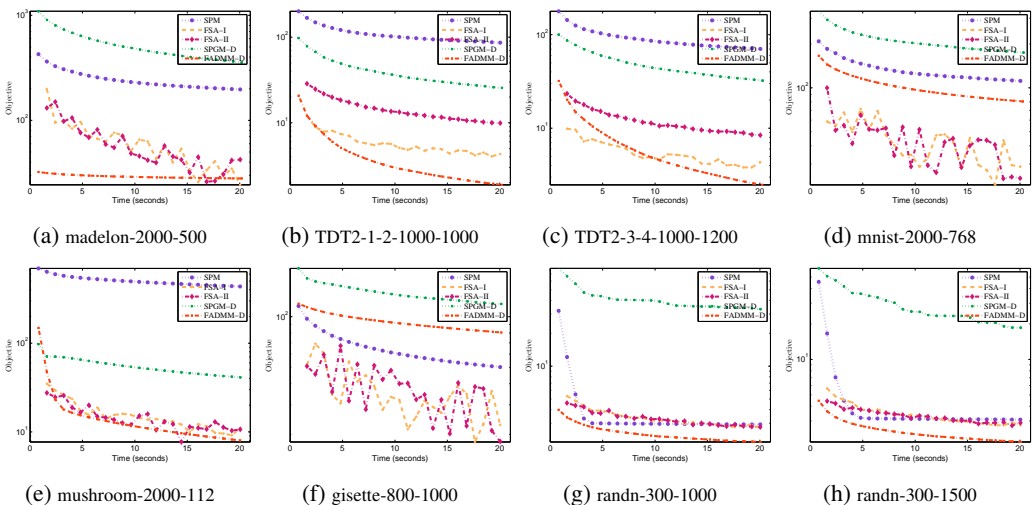

Figure 9: Results on robust sparse recovery on different datasets with $(\rho_1, \rho_2, \rho_0) = (100, 1, \infty)$.

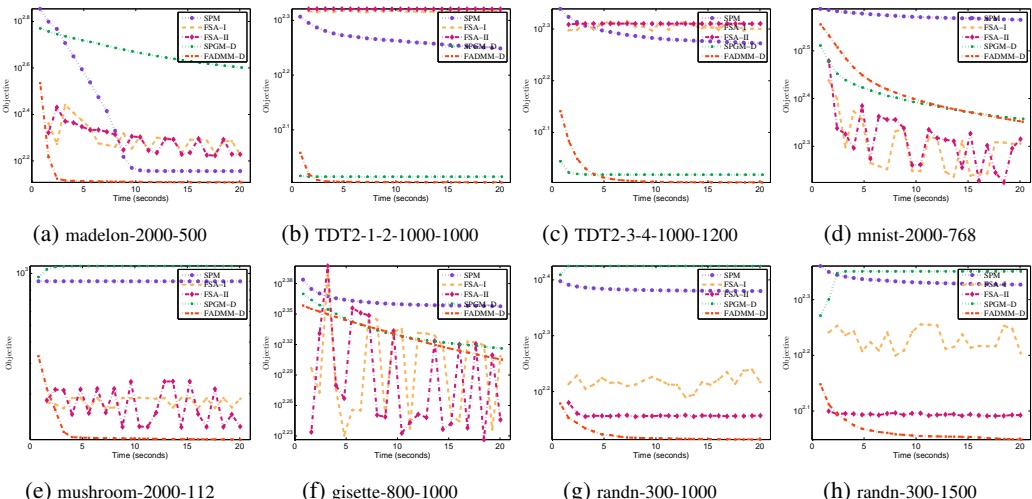

Figure 10: Results on robust sparse recovery on different datasets with $(\rho_1, \rho_2, \rho_0) = (100, 100, \infty)$.

