# OpenReview forum: "ADMM for Structured Fractional Minimization"
_ICLR.cc/2025/Conference — ICLR 2025 Poster_

### Official Review · Reviewer_EekD · 2024-10-25

**Soundness:** 3
**Presentation:** 1
**Contribution:** 2
**Rating:** 6
**Confidence:** 4

**Summary:**

This paper develops and analyzes alternating direction method of multipliers (ADMM) algorithms for solving structured fractional minimization problems. Building on recent work (Bot et al., 2023a,b; Li & Zhang, 2022), the main idea is to transform the fractional minimization problem into an equivalent composite minimization problem more amenable to operator splitting. The authors consider two classical transformations following Bot et al., 2023a,b and Li & Zhang, 2022 - Dinkelbach's parametric method and the quadratic transform method. Rather than exactly solving the transformed problems at each iteration (which could be costly), they propose to linearize them and solve this majorized version (which leads to ADMM style updates). This produces two algorithms, FADMM-D and FADMM-Q, corresponding to the two transformation approaches.

A key technical contribution is the use of smoothing via the Moreau envelope to handle the nonsmooth components in the numerator of the fractional objective. The authors establish convergence rates for both D and Q variants, showing they reach eps-approximate critical points within O(1/eps^3) iterations. The theoretical analysis is rigorous and all results are precisely stated. The definition of eps-approximate critical point, however, is nonstandard.

The two methods are validated on three applications that fit loosely within their abstract framework: sparse Fisher discriminant analysis, robust Sharpe ratio maximization, and robust sparse recovery. Note that these problems do not make full use of the relaxed assumptions outlined in the paper. Numerical experiments demonstrate that both variants typically outperform existing approaches in terms of wall-clock time to convergence and seem convincing.

**Strengths:**

- The class of problems addressed is broader than what was previously treated in the literature.

- The theoretical results about the convergence of the methods and their rates are strong and significant.

- The arguments made were rigorous and clearly stated. I wasn't able to review all the proofs in detail as there were many that were relegated to the appendix and the paper itself is quite long at nearly 40 pages.

- The methods appear to perform well in the numerical experiments compared to previous methods that can solve fractional minimization problems and other smoothing algorithms.

**Weaknesses:**

- The technical density of the presentation harms the readability.

- The discussion after Remark 3.5, on the stationarity conditions for this problem, feels very rushed and I don't understand exactly the justifications for all the claims made. In particular there are many references but they are vague - I think it would improve a lot the clarity of the paper if you could specify a lemma or result from those papers that clearly justifies what you are claiming with the subdifferential caclulus here. This also applies to Lemma A.1.

- Definition 3.8 is the definition of the Moreau envelope - I am confused to see it called Nesterov smoothing. This comes up again in the questions section because some of these results about Moreau envelopes are already well-known, even in papers cited by the authors (i.e., Bohm and Wright).

- Many hyperparameters in the method with little guidance about how to choose them or their effect on convergence.

**Questions:**

- All the numerical examples given in the paper have a denominator which is convex. This leads me to question whether the generalization of the denonimator to weakly convex functions is even well-motivated; are there problems that really necessitate this assumption?

- Similarly, what is the significance of not assuming that delta is convex if later you assume that it's a simple function? When you take delta to be the indicator function of the set of orthogonal matrices, the proximal operator is not well-defined.

- In Remark 3.11 it is claimed that Lemma 3.9 and 3.10 are novel contributions but in fact many (perhaps all?) of these results are well-known results about Moreau envelopes so in what sense are they novel? The authors should either clarify this comment and be specific about what exactly is novel or to remove this remark and just cite the known references stating these results.

- Very little is said about the effect that the parameter choices have on convergence. In practice how does one go about choosing beta_0 or other hyperparameters? For instance, in the sparse FDA experiments it's written that beta_0 = 100rho which is actually quite large (1000 to 100,000).

Minor Questions:

- What is meant in line 73 when it is claimed that when rho is large, ||X||_[k] approximates ||X||_1 ?

- In line 76, shouldn't h(Ax) have a factor of rho?

- In lin 76, isn't it more clear to simply write that d and d^1/2 are convex?

- How much effort was made to tune the parameters used in all of the different methods?

---

> ### Author Response · Authors · 2024-11-22
>
> Thank you for your efforts in evaluating our manuscript.
>
> **Weakness 1. The discussion after Remark 3.5, on the stationarity conditions for this problem, feels very rushed and I don't understand exactly the justifications for all the claims made. In particular there are many references but they are vague - I think it would improve a lot the clarity of the paper if you could specify a lemma or result from those papers that clearly justifies what you are claiming with the subdifferential caclulus here. This also applies to Lemma A.1.**
>
> **Response.** Thank you for pointing this out. We will specify the exact lemmas from the referenced papers to provide clearer justification. This section primarily serves to provide the necessary background and introduce the notion of critical points. These results are likely well-known and are not the contributions of this paper.
>
> **Weakness 2. Definition 3.8 is the definition of the Moreau envelope - I am confused to see it called Nesterov smoothing. This comes up again in the questions section because some of these results about Moreau envelopes are already well-known, even in papers cited by the authors (i.e., Bohm and Wright).**
>
> **Response.**
>
> 1. All the results are based on the definition of Nesterov's smoothing function, which is why we refer to it as Nesterov smoothing.
>
> 2. Interestingly, as demonstrated in Lemma 3.9(d), Nesterov's smoothing function is essentially equivalent to the Moreau envelope smoothing function.
>
> 3. Notably, even if the Moreau envelope smoothing function is used instead of Nesterov's smoothing function, Lemmas 3.9(e,f) remain novel contributions.
>
> 4. In our revision, we will clarify that Nesterov's smoothing function is fundamentally equivalent to the Moreau envelope smoothing function.
>
> **Weakness 3. Many hyperparameters in the method with little guidance about how to choose them or their effect on convergence.**
>
> **Response.** The algorithm involves five parameters: $(\beta^0, \xi, \theta, p, \chi)$. However, we argue that these parameters are primarily chosen to ensure the theoretical convergence of the algorithm, and the algorithm's performance is not highly sensitive to their specific values.
>
> For all our experiments (see the updated manuscript), we use the parameter settings.
>
> 1. $\xi=1/2$, $p=1/3$, $\chi>2\sqrt{1+\xi}$, $\theta>1$.
>
> 2. The proximal parameter $\theta$ is typically set to a constant slightly greater than $1$, such as $1.01$.
>
> 3. The parameter $\chi$ is typically set to a constant slightly greater than $2\sqrt{1+\xi}$, such as $2\sqrt{1+\xi}+10^{-14}$.
>
> 4. In our experiments, we only slightly tune the parameter $\beta^0$ for different applications.

---

> ### Author Response · Authors · 2024-11-22
>
> **Question 1. All the numerical examples given in the paper have a denominator which is convex. This leads me to question whether the generalization of the denonimator to weakly convex functions is even well-motivated**
>
> **Response.** When the denominator is an $L$-smooth (not necessarily convex) function, it is also $L$-weakly convex. This setting includes an important class of fractional programs where the denominator is $L$-smooth.
>
> We will include this clarification to emphasize that such generalization is well-motivated.
>
>
> **Question 2. Similarly, what is the significance of not assuming that delta is convex if later you assume that it's a simple function? When you take delta to be the indicator function of the set of orthogonal matrices, the proximal operator is not well-defined.**
>
> **Response.** We do not assume the convexity of $\delta$ since the indicator function for the orthogonality constraint is non-convex. Instead, we assume the convexity of $g(\cdot)$ and $h(\cdot)$.
>
> Even when $\delta$ is the indicator function of the set of orthogonal matrices, the proximal operator remains well-defined. The solution set is compact, and the proximal operator has an efficient closed-form solution. For details on the computation of the proximal operator, please refer to Section "G.1 ORTHOGONALITY CONSTRAINT" in the manuscript.
>
>
> **Question 3. In Remark 3.11 it is claimed that Lemma 3.9 and 3.10 are novel contributions but in fact many (perhaps all?) of these results are well-known results about Moreau envelopes so in what sense are they novel? The authors should either clarify this comment and be specific about what exactly is novel or to remove this remark and just cite the known references stating these results.**
>
> **Response.**
>
> 1. Lemma 3.9(a) is a standard result in the literature, and we have appropriately cited it in the proof.
>
> 2. Lemma 3.9(b,c) may already exist in the literature or in lecture notes, but we were unable to locate appropriate references. It is important to note that the results discussed in *Amir Beck's First-Order Methods in Optimization* (SIAM, 2017, Chapter 6) pertain to Moreau envelope smoothing, not Nesterov’s smoothing.
>
> 3. In Lemma 3.9(d), we establish for the first time that the Moreau envelope smoothing function
>
> $h_{\mu}^{more}(y) = \min_{v} \tfrac{1}{2\mu} ||v-y||_2^2 + h(v)$
>
> is equivalent to Nesterov's smoothing function:
>
> $h_{\mu}^{nest}(y) = \max_{v} <y,v> - h^*(v) - 0.5 \mu ||v||_2^2$.
>
> In other words, we have: $h_{\mu}^{more}(y)=h_{\mu}^{nest}(y)$.
>
> 4. We argure that Lemma 3.9(d,e,f) and Lemma 3.10 represent novel contributions of this paper. Notably, even if the Moreau envelope smoothing function is used instead of Nesterov's smoothing function, Lemmas 3.9(e,f) remain novel.
>
> 5. In our revision, we have changed it to: "Lemma 3.9 and Lemma 3.10 can be derived using standard convex analysis and
> play an essential role in the analysis of the proposed FADMM algorithm. Interestingly, as
> demonstrated in Lemma 3.9(d), Nesterov’s smoothing function is essentially equivalent to the Moreau envelope smoothing function (Beck, 2017; Böhm & Wright, 2021)." See L204-208 in the updated manuscript.
>
> **Question 4. Very little is said about the effect that the parameter choices have on convergence. In practice how does one go about choosing beta_0 or other hyperparameters? For instance, in the sparse FDA experiments it's written that beta_0 = 100rho which is actually quite large (1000 to 100,000).**
>
>
> **Response.**
>
> In all our experiments, we use the following parameter settings:
> $$\xi=1/2, p=1/3, \chi=2\sqrt{1+\xi}+10^{-14}, \theta=1.01$$
>
> For the parameter $\beta^0$, the following values work well across different applications:
>
> 1. $\beta^0= 100 \rho$ for sparse FDA
>
> 2. $\beta^0= 0.001$ for robust SRM
>
> 3. $\beta^0=0.001$ for robust sparse recovery.
>
> For the sparse FDA experiments, according to the exact penalty theory  (Gotoh et al., 2018; Bi et al., 2014), the value of $\beta^t$ is expected to be at least larger than $\rho$. This is why we consider a relatively large value for $\beta^0$.
>
> While it is possible to choose a smaller $\beta^0$, this may result in a little slower convergence.

---

> ### Author Response · Authors · 2024-11-22
>
> **Question 5. What is meant in line 73 when it is claimed that when rho is large, ||X||_[k] approximates ||X||_1 ?**
>
>
> **Response.** According to (Gotoh et al., 2018; Bi et al., 2014), when $\rho$ exceeds a certain threshold, the exact penalty function:
>
> $$\min_X f(X)+\rho( ||X||_1 - ||X||_{[k]} ) $$
>
> becomes equivalent to the original sparsity-constrained problem $$\min_X f(X), s.t. ||X||_0\leq k$$
>
> in the sense that these two problems have the same global optimal solution set.
>
> We will briefly mention this theoretical result in the updated manuscript.
>
>
>
>
> **Question 6. In line 76, shouldn't h(Ax) have a factor of rho?**
>
>
> **Response.** You are correct. Thank you for your careful reading and for pointing this out.
>
>
> **Question 7. In lin 76, isn't it more clear to simply write that d and d^1/2 are convex?**
>
>
> **Response.** We will change it to "d or d^1/2 are weakly convex". Thank you for your suggestion.
>
>
> **Question 8. How much effort was made to tune the parameters used in all of the different methods?**
>
>
> **Response.** We only tune one parameter $\beta^0$:
>
> 1. $\beta^0=\rho*100$ for sparse FDA
>
> 2. $\beta^0= 0.001$ for robust SRM
>
> 3. $\beta^0=0.001$ for robust sparse recovery.
>
> For **all** experiments (refer to the updated manuscript), we consider the following fixed constants:
> $$\xi=1/2, p=1/3, \chi=2\sqrt{1+\xi}+10^{-14}, \theta=1.01.$$

---

> ### Comment · Reviewer_EekD · 2024-11-22
> **Response to rebuttal**
>
> **Regarding Weakness 2**
>
> There is no distinction between the Moreau envelope and Nesterov’s smoothing, except that the Moreau envelope was discovered and published in the 1960s by both Moreau and Yosida. To call it Nesterov’s smoothing doesn’t make sense as it is historically incorrect. It is nonsensical to suggest using Nesterov’s smoothing instead of the Moreau envelope - there is no difference so I don’t understand at all what you're claiming when you write “Notably, even if the Moreau envelope smoothing function is used instead of Nesterov's smoothing function, Lemmas 3.9(e,f) remain novel contributions.” These are two names for the same object; they are not just fundamentally equivalent, they are the same.
>
> **Regarding Question 1**
>
> I am aware that an L-smooth function is weakly convex. The point is that you don’t give any examples with these functions int he denominator - what is the point of generalizing if you don’t have examples that require it?
>
> **Regarding Question 2**
>
> When I write that the proximal operator is not well-defined for the indicator of the set of orthogonal matrices, I mean that it is not unique in general. What you’ve written in G.1 is in the argmin of the projection subproblem but it is not unique - what happens if I pick something else in this set? There can be more than one orthogonal matrix whose distance to the current matrix is equal.
>
> **Regarding Question 3**
>
> **I completely disagree with your response and the edited submission.**
> * Lemma 3.9 (b) and (c) are indeed known, for instance proposition 2.1 of “Generalized Conditional Gradient with Augmented Lagrangian for Composite Minimization” by Silveti-Falls et al which cites the well-known textbook of Bauschke and Combettes on convex analysis.
> * Lemma 3.9 (d) is NOT a novel contribution, it’s literally on the wikipedia page for Moreau envelope https://en.wikipedia.org/wiki/Moreau_envelope#Properties as well as in various lecture notes, e.g., https://candes.su.domains/teaching/math301/Lectures/Moreau-Yosida.pdf 22.3.32 Dual Viewpoint. I absolutely insist that these results are well-known and that it is not reasonable to claim these are novel contributions at all.
> * Lemma 3.9 (e) and (f) are also standard results; there is no doubt about this, they are looser versions of what is written in Silveti-Falls et al but you can see these reuslts also in Appendix A1 of "A Conditional Gradient Framework for Composite Convex Minimization with Applications to Semidefinite Programming" by Yurtsever et al or in Appendix A Lemma 10 "A Smooth Primal-Dual Optimization Framework for Nonsmooth Composite Convex Minimization" by Tran-Dinh et al among many. It's well-known that subgradients of Lipschitz functions are bounded in norm by the Lipschitz constant (see many textbooks for this result, e.g., Shai Shalev-Shwartz et al. Online learning and online convex optimization.).
>
> **Regarding Question 4**
>
> “While it is possible to choose a smaller, this may result in a little slower convergence.”
>
> Comments like this, along with supporting experimental evidence, would greatly improve the quality of the submission.

---

> > ### Author Response · Authors · 2024-11-22
> > **Reply to: Regarding Question 3 (Lemmas 3.9 and 3.10)**
> >
> > Note that in our revised submission, we did not claim Lemma 3.9 as our contribution and explicitly stated:
> >
> > "Lemma 3.9 and Lemma 3.10 can be derived using standard convex analysis and are fundamental to the analysis of the proposed FADMM algorithm."
> >
> > Thank you for bringing these references to our attention. We will include citations to them in the revised paper.

---

> > ### Author Response · Authors · 2024-12-02
> > **Reply to: Regarding Question 4**
> >
> > We acknowledge the reviewer’s concern regarding the choice of the parameter $\beta^0$. However, we have not included detailed experiments with varying $\beta^0$ for the following two reasons:
> >
> > 1. We have already conducted experiments with 72 different settings (9x8 figures), which provide valuable insights into the robustness of the proposed methods. Including additional experiments would result in excessive redundancy, especially for a theoretical paper, detracting from its clarity and conciseness.
> >
> > 2. All the MATLAB code for reproducibility is provided in the supplementary materials. Therefore, we believe that the experimental section does not represent a weakness of the paper.

---

> ### Author Response · Authors · 2024-11-22
> **Reply to: Regarding Weakness 2**
>
> We now focus on Moreau envelope and Nesterov’s smoothing:
>
> $h_{\mu}^{more}(y) = \min_{v} \tfrac{1}{2\mu} ||v-y||_2^2 + h(v)$
>
> $h_{\mu}^{nest}(y) = \max_{v} <y,v> - h^*(v) - 0.5 \mu ||v||_2^2$.
>
> 1. Although these formulations are essentially equivalent with $h_{\mu}^{more}(y)=h_{\mu}^{nest}(y)$, they take different forms.
>
> 2. The Moreau envelope function $h_{\mu}^{more}(y)$ involves adding **a strongly convex term** to the **primal** minimization problem, while Nesterov’s smoothing function $h_{\mu}^{nest}(y)$ incorporates **a strongly concave term** into the **dual** maximization problem.
>
> 3. This distinction leads to different strategies for deriving analytical solutions to the proximal subproblem and results in primal-dual algorithms.
>
> 4. Finally, we use the term Nesterov’s smoothing technique in Definition 3.8, along with the summarized properties of Nesterov’s smoothing function. These results remain valid.

---

> ### Author Response · Authors · 2024-11-22
> **Reply to: Regarding Question 2**
>
> **Question. When I write that the proximal operator is not well-defined for the indicator of the set of orthogonal matrices, I mean that it is not unique in general. What you’ve written in G.1 is in the argmin of the projection subproblem but it is not unique - what happens if I pick something else in this set? There can be more than one orthogonal matrix whose distance to the current matrix is equal.**
>
> **Response.**
>
> The subproblem is not required to have a unique solution but must be solved to global optimality (See Assumption 3.3). Note that we obtain the following two essential conditions for the nonconvex subproblem:
>
> 1. The necessary first-order optimality condition
>
> 2. The necessary and sufficient zero-order optimality condition (See L1343-1344 and L1514-1515)
>
> To address your question, we do not "pick something else in this set" because we select the globally optimal solution for the nonconvex proximal subproblem. The subproblem does not need to be unique, just as the critical point is not necessarily unique.

---

> ### Author Response · Authors · 2024-12-02
> **Reply to: Regarding Question 1**
>
> Given that the class of $L$-smooth functions for the denominator is quite broad, we discuss such a general case for future reference, following the work by Radu Ioan Bot et al. (Inertial Proximal Block Coordinate Method for a Class of Nonsmooth Sum-of-Ratios Optimization Problems, SIOPT 2023).
>
> In the following, we discuss two examples of weakly convex denominator functions.
>
> 1. Consider the Sparse FDA problem described in Section 1.1. If the denominator takes the form $d(X) = \text{trace}(X'DX) + c$, where $D$ is not necessarily positive semidefinite and $c > 0$ is sufficiently large to ensure $d(X) > 0$, then $d(X)$ is non-convex. However, it is $(2||D||)$-weakly convex, and the proposed FADMM can still be applied.
>
> 2. Consider another example where the denominator is a logarithmically convex function (but not necessarily convex), such as $d(x) = \log(||Ax||_2^2 + 1)$. Although it is nonconvex, $d(x)$ can still be weakly convex under mild conditions. In this case, the proposed FADMM can still be applied.

---

### Official Review · Reviewer_AqZe · 2024-11-03

**Soundness:** 3
**Presentation:** 2
**Contribution:** 3
**Rating:** 6
**Confidence:** 2

**Summary:**

In this paper, the authors introduce FADMM, the first ADMM algorithm designed to solve general structured fractional minimization problems.  FADMM decouples the original problem into linearized proximal subproblems, featuring two variants: one using Dinkelbach’s parametric method (FADMM-D) and the other using the quadratic transform method (FADMM-Q). The proposed algorithm improves the slow convergence speed and numerical stability issues of traditional subgradient methods and smoothing proximal gradient methods. The authors conduct a convergence analysis of the FADMM algorithm by introducing a novel Lyapunov function, and they validate the effectiveness of the FADMM algorithm through extensive synthetic and real-world data.

**Strengths:**

1.The authors provide a comprehensive analysis of the proposed FADMM algorithm, including two specific variants: FADMM-D and FADMM-Q.

2.Comprehensive theoretical analysis, with proofs on convergence.

3.The authors conduct extensive experiments on both synthetic and real-world data, effectively demonstrating the efficiency of the FADMM algorithm.

**Weaknesses:**

I think the writing of this paper can be further improved.

**Questions:**

I am not an expert in non-convex optimization, I can only give some advice on writing papers：

1.The first sentence of the abstract is too long. It is recommended to split it to improve readability.

2.Line 73, "sufficient large" should be "sufficiently large".

3.Line 138, "To the best of our knowledge......" is too long. It is better to break it into short sentences for reading.

4.Line 236,  "is a widely used to develop practical optimization algorithms ", delete ”a“.

5.Line 454, "Additioanl"  -> "Additional".

---

> ### Author Response · Authors · 2024-11-22
>
> We are grateful to the reviewer for the time spent reviewing our manuscript.
>
> **Question 1: I think the writing of this paper can be further improved.**
>
> **Response:**
>
> Thank you for your thorough review and for pointing out the typos in our manuscript. We will carefully consider your suggestions and make the necessary revisions to improve the writing.

---

### Official Review · Reviewer_7KB3 · 2024-11-04

**Soundness:** 3
**Presentation:** 2
**Contribution:** 3
**Rating:** 8
**Confidence:** 4

**Summary:**

This paper proposes an ADMM for solving a class of structured fractional minimization problems. The main techniques are based on smoothing methods and two well-established approaches for fractional minimization problems. The convergence rate of the proposed ADMM is established. Some numerical results are reported to show the efficiency of the proposed ADMM. Overall, the topic and results are interesting.

**Strengths:**

The authors discuss a class of fractional minimization problems, which seem not to be well addressed in the literature. The authors provide corresponding theoretical analysis and numerical results.

**Weaknesses:**

The results in Lemmas 3.9 and 3.10 are standard in the literature. The authors do not need to prove them in the appendix and should not claim the results ``represent novel contributions.''

After using the smoothing techniques, the hard term $h(Ax)$ will become $h_{\mu}(Ax)$. The authors could then use some standard methods from the fractional minimization community to solve this problem. The corresponding complexity will also be $\mathcal{O}(\epsilon^{-3})$. The authors might want to comment on this. It seems that Bot et al. (2023) also used this smoothing technique.

**Questions:**

- The authors do not explain Assumption 5.1 well in Remark 5.2. By the algorithm, the iterate $x^t$ is only in the domain of $\delta(\cdot)$. You cannot say that it lies in a bounded set. The same issue occurs in Lemma 5.5. It might not be safe to assume that $x^t$ is bounded.

- The notation $\underline{\mathrm{Fd}}$ and $\overline{\mathrm{Fd}}$ should be $\underline{F} \, \underline{d}$ and $\overline{F} \, \overline{d}$, respectively.

- If the $\epsilon$-critical point is similar to that in Bot et al. (2023b), will the same complexity results hold?

- What is the method SGM mentioned in Section 7?

- The authors might need to compare their method to that in Bot et al. (2023b), at least for the Robust SRM problem.

---

> ### Author Response · Authors · 2024-11-22
>
> Thank you for your efforts in evaluating our manuscript.
>
> **Question 1:-- The results in Lemmas 3.9 and 3.10 are standard in the literature. The authors do not need to prove them in the appendix and should not claim the results ``represent novel contributions.**
>
> **Response:**
>
> In our revision, we have changed it to: "Lemma 3.9 and Lemma 3.10 can be derived using standard convex analysis and
> play an essential role in the analysis of the proposed FADMM algorithm. Interestingly, as
> demonstrated in Lemma 3.9(d), Nesterov’s smoothing function is essentially equivalent to the Moreau envelope smoothing function (Beck, 2017; Böhm & Wright, 2021)." See L204-208 in the updated manuscript.
>
> **Question 2: After using the smoothing techniques, the hard term $h(Ax)$ will become $h_{\mu}(Ax)$. The authors could then use some standard methods from the fractional minimization community to solve this problem. The corresponding complexity will also be $\epsilon^{-3}$. The authors might want to comment on this. It seems that Bot et al. (2023) also used this smoothing technique.**
>
> **Response:** The strategy suggested by the reviewer is essentially the Smoothing Proximal Gradient Method (SPGM) (Beck & Rosset, 2023; Böhm & Wright, 2021) applied to fractional programs. We have discussed this method in the "RELATED WORK" section under "General Algorithms for Solving Problem (1) — Smoothing Proximal Gradient Methods (SPGM).
>
> Bot et al. (2023) employ a smoothing technique that involves adding a stronger concave term to the dual maximization problem, which can be interpreted as another primal-dual method. However, their analysis relies on the Kurdyka–Łojasiewicz (KL) inequality of the problem, and no iteration complexity is provided. We have also included a numerical comparison with this method.
>
> **Question 3: The authors do not explain Assumption 5.1 well in Remark 5.2. By the algorithm, the iterate $x^t$ is only in the domain of $\delta(\cdot)$. You cannot say that it lies in a bounded set. The same issue occurs in Lemma 5.5. It might not be safe to assume that $x^t$ is bounded.**
>
> **Response:** Note that we assume the constraint set is **compact**, satisfying $||x^t|| \leq R$ for some $R$. (This assumption holds for all three applications considered.)
>
> If $x \in \operatorname{dom}(f)\triangleq \{ x : f(x) < +\infty \}$, then $x$ is feasible and it holds that $||x^t|| \leq R$.
>
>
> **Question 4: The notation  $\underline{Fd}$ and $\overline{Fd}$ should be $\underline{F}$， $\underline{d}$ and $\underline{Fd}$, respectively.**
>
> **Response:** We will change $\underline{Fd}$ to $\underline{F}\cdot\underline{d}$.
>
>
>
> **Question 5: If the $\epsilon$-critical point is similar to that in Bot et al. (2023b), will the same complexity results hold?**
>
> **Response:**
>
> 1. Note that an exact critical point does not depend on specific algorithms, whereas the definition of an $\epsilon$-critical point, whether in Bot et al.'s work or ours, does.
>
> 2. Both notions of $\epsilon$-critical points are reasonable, as they converge to the exact critical point when $\epsilon = 0$ and depend only on the solution.
>
> Therefore, these two notions are not directly comparable.
>
> To further illustrate, let us consider a simple example. Assume that $x=4$ is an exact critical point. One definition asserts that $(x,y,z)$ is an $\epsilon$-critical point if $|x-y|+|x-z|+|y-4|\leq \epsilon$. Another definition may assert that  $(x,y)$ is an $\epsilon$-critical point if $|x-y|+|\sqrt{x}-2|\leq \epsilon$. Both definitions are reasonable.

---

> ### Author Response · Authors · 2024-11-22
>
> **Question 6: What is the method SGM mentioned in Section 7?**
>
> **Response:** This is a typo; it should read "Subgradient Projection Methods (SPM)." Thank you for your careful reading and for pointing this out.
>
> **Question 7:The authors might need to compare their method to that in Bot et al. (2023b), at least for the Robust SRM problem.**
>
> **Response:**
>
> 1. Upon request, we have included a comparison with Bot et al.'s Fully Splitting Algorithm (FSA) across all three applications: sparse FDA, robust SRM, and robust sparse recovery.
>
> 2. We adapted the original algorithm from (Bot et al., 2023b) to our notation to solve Problem (1). Refer to Section H for the implementation details.
>
>
> 3. **Figures 1,2,3,4,5,6,7,8,9** in the updated manuscript present the experimental results. Additionally, we have updated the supplementary material to ensure reproducibility.

---

> > ### Comment · Reviewer_7KB3 · 2024-11-25
> >
> > Thank you to the authors for the clarification. I have increased the score from 5 to 6.

---

### Meta-Review · Area_Chair_QNUA · 2024-12-19

**Metareview:**

The paper presents a novel ADMM-based optimization method, tailored for structured fractional minimization problems, which seem not to be well addressed in the literature. The main techniques are based on smoothing methods (via the Moreau envelope) and two well-established approaches for fractional minimization problems. The authors provide corresponding theoretical analysis and numerical results. A Lyapunov function demonstrates convergence within an oracle complexity of $O(1/\epsilon^3)$, and empirical evaluations confirm applicability across domains like sparse Fisher discriminant analysis and robust sparse recovery. Overall, the topic and results are interesting. The reviewers are broadly positive about the submission.

**Additional Comments On Reviewer Discussion:**

Some concerns were raised during the initial review phase that were adequately addressed in the rebuttal. The reviewers note that the authors have adequately addressed the concerns raised during the rebuttal process, and the work demonstrates both technical soundness and potential impact.

---

### Decision · Program_Chairs · 2025-01-22

Accept (Poster)